# The structure, function and evolution of a complete human chromosome 8

Glennis A. Logsdon[1], Mitchell R. Vollger[1], PingHsun Hsieh[1], Yafei Mao[1], Mikhail A. Liskovykh[2], Sergey Koren[3], Sergey Nurk[3], Ludovica Mercuri[4], Philip C. Dishuck[1], Arang Rhie[3], Leonardo G. de Lima[5], Tatiana Dvorkina[6], David Porubsky[1], William T. Harvey[1], Alla Mikheenko[6], Andrey V. Bzikadze[7], Milinn Kremitzki[8], Tina A. Graves-Lindsay[8], Chirag Jain[3], Kendra Hoekzema[1], Shwetha C. Murali[1,9], Katherine M. Munson[1], Carl Baker[1], Melanie Sorensen[1], Alexandra M. Lewis[1], Urvashi Surti[10], Jennifer L. Gerton[5], Vladimir Larionov[2], Mario Ventura[4], Karen H. Miga[11], Adam M. Phillippy[3] & Evan E. Eichler[1,9]✉

The complete assembly of each human chromosome is essential for understanding human biology and evolution[1,2]. Here we use complementary long-read sequencing technologies to complete the linear assembly of human chromosome 8. Our assembly resolves the sequence of five previously long-standing gaps, including a 2.08-Mb centromeric α-satellite array, a 644-kb copy number polymorphism in the β-defensin gene cluster that is important for disease risk, and an 863-kb variable number tandem repeat at chromosome 8q21.2 that can function as a neocentromere. We show that the centromeric α-satellite array is generally methylated except for a 73-kb hypomethylated region of diverse higher-order α-satellites enriched with CENP-A nucleosomes, consistent with the location of the kinetochore. In addition, we confirm the overall organization and methylation pattern of the centromere in a diploid human genome. Using a dual long-read sequencing approach, we complete high-quality draft assemblies of the orthologous centromere from chromosome 8 in chimpanzee, orangutan and macaque to reconstruct its evolutionary history. Comparative and phylogenetic analyses show that the higher-order α-satellite structure evolved in the great ape ancestor with a layered symmetry, in which more ancient higher-order repeats locate peripherally to monomeric α-satellites. We estimate that the mutation rate of centromeric satellite DNA is accelerated by more than 2.2-fold compared to the unique portions of the genome, and this acceleration extends into the flanking sequence.

Since the announcement of the sequencing of the human genome 20 years ago[1,2], human chromosomes have remained unfinished owing to large regions of highly identical repeats clustered within centromeres, regions of segmental duplication, and the acrocentric short arms of chromosomes. The presence of large swaths (more than 100 kb) of highly identical repeats that are themselves copy number polymorphic has meant that such regions have persisted as gaps, which limits our understanding of human genetic variation and evolution[3,4]. The advent of long-read sequencing technologies and the use of DNA from complete hydatidiform moles, however, have now made it possible to assemble these regions from native DNA for the first time[5–7]. Here we present the first, to our knowledge, complete linear assembly of human chromosome 8. We chose to assemble chromosome 8 because it carries a modestly sized centromere (approximately 1.5–2.2 Mb)[8,9],

in which AT-rich, 171-base-pair (bp) α-satellite repeats are organized into a well-defined higher-order repeat (HOR) array. The chromosome, however, also contains one of the most structurally dynamic regions in the human genome—the β-defensin gene cluster at 8p23.1 (refs. [10–12])—as well as a recurrent polymorphic neocentromere at 8q21.2, which have been largely unresolved for the past 20 years.

## Telomere-to-telomere assembly of chromosome 8

Unlike the assembly of the human X chromosome[13], we took advantage of both ultra-long Oxford Nanopore Technologies (ONT) and Pacific Biosciences (PacBio) high-fidelity (HiFi) data to resolve the gaps in human chromosome 8 (Fig. 1a, b, Methods). We first generated 20-fold sequence coverage of ultra-long ONT data and 32.4-fold coverage of

[1]Department of Genome Sciences, University of Washington School of Medicine, Seattle, WA, USA. [2]Developmental Therapeutics Branch, National Cancer Institute, Bethesda, MD, USA. [3]Genome Informatics Section, Computational and Statistical Genomics Branch, National Human Genome Research Institute, National Institutes of Health, Bethesda, MD, USA. [4]Department of Biology, University of Bari, Aldo Moro, Bari, Italy. [5]Stowers Institute for Medical Research, Kansas City, MO, USA. [6]Center for Algorithmic Biotechnology, Institute of Translational Biomedicine, Saint Petersburg State University, Saint Petersburg, Russia. [7]Graduate Program in Bioinformatics and Systems Biology, University of California, San Diego, San Diego, CA, USA. [8]McDonnell Genome Institute, Department of Genetics, Washington University School of Medicine, St Louis, MO, USA. [9]Howard Hughes Medical Institute, University of Washington, Seattle, WA, USA. [10]Department of Pathology, University of Pittsburgh, Pittsburgh, PA, USA. [11]Center for Biomolecular Science and Engineering, University of California, Santa Cruz, Santa Cruz, CA, USA. ✉e-mail: eee@gs.washington.edu

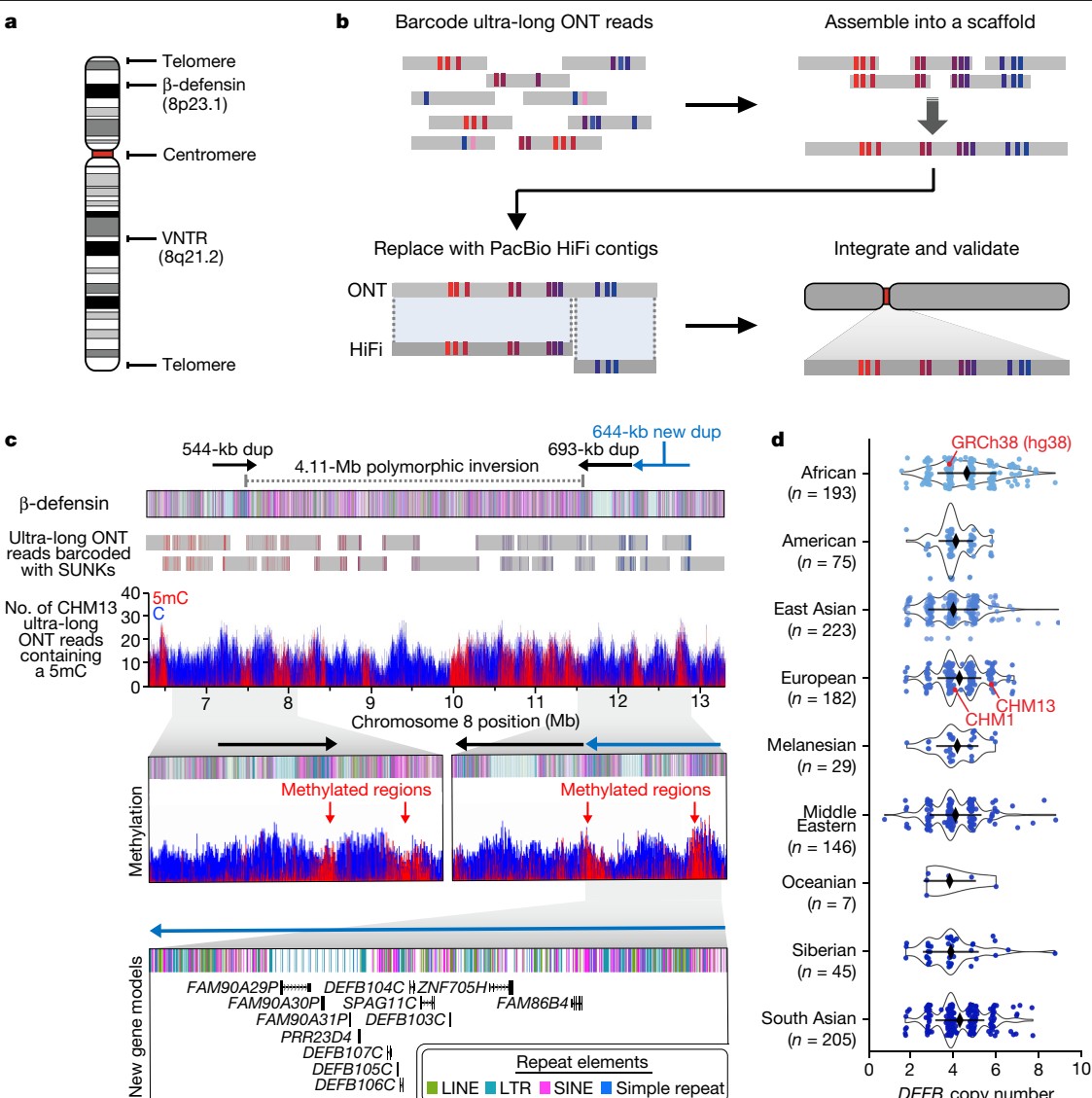

**Fig. 1 | Telomere-to-telomere assembly of human chromosome 8. a**, Gaps in the GRCh38 chromosome 8 reference sequence. **b**, Targeted assembly method to resolve complex repeat regions in the human genome. Ultra-long ONT reads (grey) are barcoded with SUNKs (coloured bars) and assembled into a sequence scaffold. Regions within the scaffold sharing high sequence identity with PacBio HiFi contigs (dark grey) are replaced, improving the base accuracy to greater than 99.99%. The PacBio HiFi assembly is integrated into an assembly of CHM13 chromosome 8 (ref. [5]) and validated. **c**, Sequence, structure, methylation status and genetic composition of the CHM13 β-defensin locus. The locus contains three segmental duplications (dups) at chr8:7098892–7643091, chr8:11528114–12220905 and chr8:12233870–12878079. A 4,110,038-bp inversion (chr8:7500325–11610363) separates the first and second duplications. Iso-Seq data reveal that the third duplication (light blue) contains 12 new protein-coding genes, five of which are *DEFB* genes (Extended Data Fig. 3g). **d**, Copy number of the *DEFB* genes (chr8:7783837–7929198 in GRCh38) throughout the human population, determined from a collection of 1,105 high-coverage genomes (Methods). Data are median ± s.d.

PacBio HiFi data from a complete hydatidiform mole (CHM13hTERT, hereafter referred to as CHM13) (Supplementary Fig. 1). Then, we assembled complex regions in chromosome 8 by creating a library of singly unique nucleotide *k*-mers (SUNKs)[14], or sequences of length *k* that occur approximately once per haploid genome (here, *k* = 20), from CHM13 PacBio HiFi data. We validated the SUNKs with Illumina data from the same genome and used them to barcode ultra-long ONT reads (Fig. 1b). Ultra-long ONT reads that share highly similar barcodes were assembled into an initial sequence scaffold that traverses each chromosome 8 gap (Fig. 1b). We improved the base-pair accuracy of the sequence scaffolds by replacing the raw ONT sequence with concordant PacBio HiFi contigs and integrating them into a previously generated[5] linear assembly of human chromosome 8 (Fig. 1b, Methods).

The complete telomere-to-telomere sequence of human chromosome 8 is 146,259,671 bases long and includes 3,334,256 bases that are

missing from the current reference genome (GRCh38). Most of the additions reside within distinct chromosomal regions: a 644-kb copy number polymorphic β-defensin gene cluster that maps to chromosome 8p23.1 (Fig. 1c, d); the complete centromere corresponding to 2.08 Mb of α-satellite HORs (Fig. 2); an 863-kb 8q21.2 variable number tandem repeat (VNTR) (Extended Data Fig. 1); and both telomeric regions that end with the canonical TTAGGG repeat sequence (Extended Data Fig. 2). We validated the assembly with optical maps (Bionano Genomics), single-cell DNA template strand sequencing (Strand-seq)[15,16], and comparisons to finished bacterial artificial chromosome (BAC) sequences as well as Illumina whole-genome sequencing data derived from the same source genome (Supplementary Fig. 2, Methods). We estimate the overall base accuracy of our chromosome 8 assembly to be between 99.9915% and 99.9999% (quality value score between 40.70 and 63.19, as determined from sequenced BACs and

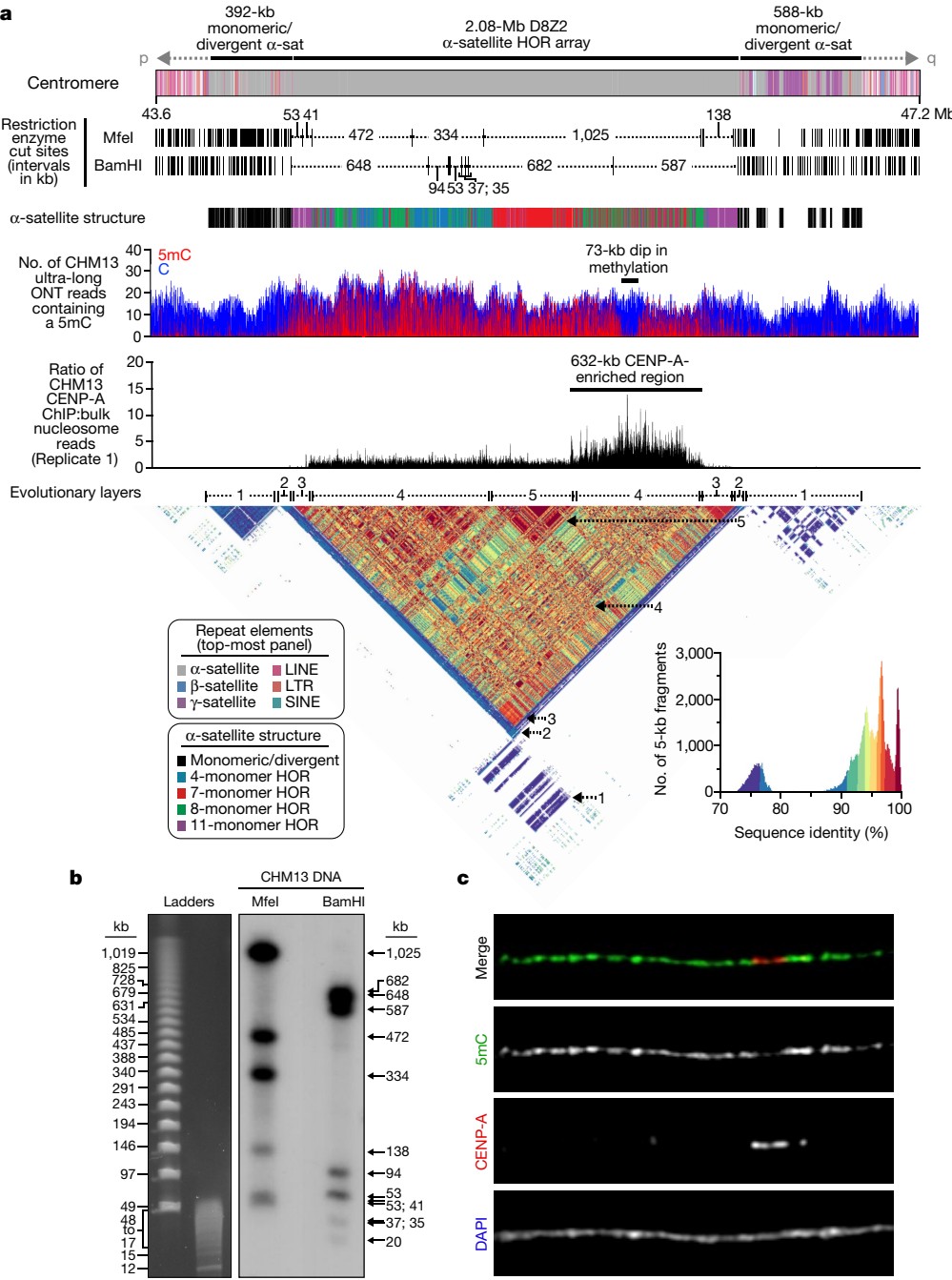

**Fig. 2 | Sequence, structure and epigenetic map of the chromosome 8 centromeric region. a**, Schematic showing the composition of the CHM13 chromosome 8 centromere. The centromeric region consists of a 2.08-Mb D8Z2 α-satellite HOR array flanked by regions of monomeric and/or divergent α-satellite interspersed with retrotransposons, β-satellite and γ-satellite. The predicted restriction digest pattern is shown. The D8Z2 α-satellite HOR array is heavily methylated except for a 73-kb hypomethylated region, which is contained within a 632-kb CENP-A chromatin domain (Extended Data Fig. 9,

Supplementary Fig. 8). A pairwise sequence identity heat map indicates that the centromere is composed of five distinct evolutionary layers (dashed arrows). **b**, Pulsed-field gel Southern blot of CHM13 DNA confirms the structure and organization of the chromosome 8 centromeric HOR array. Left, ethidium bromide (EtBr) staining; right, [32]P-labelled chromosome 8 α-satellite-specific probe. *n* = 2. See Supplementary Fig. 9a, b for gel source data. **c**, Representative images of a CHM13 chromatin fibre showing CENP-A enrichment in an unmethylated region. *n* = 3. Scale bar, 1 μm.

mapped *k*-mers[17], respectively). An analysis of 24 million human full-length transcripts generated from isoform sequencing (Iso-Seq) data identifies 61 protein-coding and 33 noncoding loci that map to this finished chromosome 8 sequence better than to GRCh38 (Extended Data Fig. 3a–f, Supplementary Table 1), including the discovery of new genes mapping to copy number polymorphic regions (Fig. 1c, d, Extended Data Fig. 3g).

Our targeted assembly method successfully resolved the β-defensin gene cluster[10] into a single 7.06-Mb locus, eliminating two 50-kb gaps in GRCh38 (Fig. 1c, Extended Data Fig. 4). We estimate the base accuracy of this locus to be 99.9911% (quality value score 40.48; based on mapped BACs) (Extended Data Fig. 5a). Our analysis reveals CHM13 has a more structurally complex haplotype than GRCh38 (Fig. 1d, Extended Data Fig. 4), consistent with previously published reports[10,12]. We resolve

the breakpoints of one of the largest common inversion polymorphisms in the human genome (4.11 Mb) and show that the breakpoints map within large, highly identical duplications that are copy number polymorphic (Fig. 1c, d, Extended Data Fig. 5b). In contrast to the human reference, which carries two such segmental duplications, there are three segmental duplications in CHM13: a 544-kb segmental duplication on the distal end and two 693- and 644-kb segmental duplications on the proximal end (Fig. 1c). Each segmental duplication cassette carries at least five β-defensin genes and, as a result, we identify five additional β-defensin genes that are almost identical at the amino acid level to the reference (Fig. 1c, Supplementary Table 2). Because ONT data allow methylation signals to be assessed[18], we determined the methylation status of cytosine residues across the entire β-defensin locus. All three segmental duplications contain a 151–163-kb methylated region that resides in the long-terminal repeat (LTR)-rich region of the duplication, whereas the remainder of the duplication, including the β-defensin gene cluster, is largely unmethylated (Fig. 1c). Complete sequence resolution of this alternative haplotype is important because the inverted haplotype preferentially predisposes to recurrent microdeletions associated with developmental delay, microcephaly and congenital heart defects[19,20]. Copy number polymorphism of the five β-defensin genes has been associated with immune-related phenotypes, such as psoriasis and Crohn's disease[11,21].

## Sequence resolution of the chromosome 8 centromere

Previous studies estimate the length of the chromosome 8 centromere to be between 1.5 and 2.2 Mb, on the basis of analysis of the HOR α-satellite array[8,9]. Although α-satellite HORs of different lengths are thought to comprise the centromere, the predominant species has a unit length of 11 monomers (1,881 bp)[8,9]. During assembly, we spanned the chromosome 8 centromere with 11 ultra-long ONT reads (mean length 389.4 kb), which were replaced with PacBio HiFi contigs based on SUNK barcoding. Our chromosome 8 centromere assembly consists of a 2.08-Mb D8Z2 α-satellite HOR array flanked by blocks of monomeric α-satellite on the p-arm (392 kb) and q-arm (588 kb) (Fig. 2a). Both monomeric α-satellite blocks are interspersed with long and short interspersed nuclear elements (LINEs and SINEs, respectively), LTRs and β-satellites, with tracts of γ-satellite specific to the q-arm. Several methods were used to validate its organization. First, long-read sequence read-depth analysis from two orthogonal native DNA sequencing platforms shows uniform coverage, which suggests that the assembly is free from large structural errors (Extended Data Fig. 6a). Fluorescent in situ hybridization (FISH) on metaphase chromosomes confirms the long-range organization of the centromere (Extended Data Fig. 6a–c). Droplet digital PCR shows that there are 1,344 ± 142 (mean ± s.d.) D8Z2 HORs within the α-satellite array, consistent with our estimates (Extended Data Fig. 6d, Methods). Pulsed-field gel electrophoresis Southern blots on CHM13 DNA digested with two different restriction enzymes supports the banding pattern predicted from the assembly (Fig. 2a, b). Finally, applying our assembly approach to ONT and HiFi data available for a diploid human genome (HG00733) (Supplementary Table 3, Methods) generates two additional chromosome 8 centromere haplotypes, replicating the overall organization with only subtle differences in the overall length of HOR arrays (Extended Data Fig. 7, Supplementary Table 4).

We find that the chromosome 8 centromeric HOR array is primarily composed of four distinct HOR types represented by 4, 7, 8 or 11 α-satellite monomer cassettes (Fig. 2a, Extended Data Fig. 8). Although the 11-monomer HOR predominates (36%), the other HORs are also abundant (19–23%) and are all derivatives of the 11-monomer HOR (Extended Data Fig. 8b, c). Notably, we find that the HORs are differentially distributed regionally across the centromere. Although most regions show a mixture of different HOR types, we also identify regions of homogeneity, such as clusters of 11-monomer HORs mapping to the periphery of the HOR array (92 and 158 kb in length) and a 177-kb region in the centre composed solely of 7-monomer HORs. To investigate the epigenetic organization, we inferred methylated cytosine residues along the centromeric region and find that most of the α-satellite HOR array is methylated, except for a small, 73-kb hypomethylated region (Fig. 2a). To determine whether this hypomethylated region is the site of the epigenetic centromere (marked by the presence of nucleosomes that contain the histone H3 variant CENP-A), we performed CENP-A chromatin immunoprecipitation with high-throughput sequencing (ChIP–seq) on CHM13 cells and found that CENP-A is primarily located within a 632-kb stretch that encompasses the hypomethylated region (Fig. 2a, Extended Data Fig. 9). Subsequent chromatin fibre FISH revealed that CENP-A maps to the hypomethylated region within the α-satellite HOR array (Fig. 2c). Notably, the hypomethylated region shows some of the greatest HOR admixture, which suggests a potential optimization of HOR subtypes associated with the active kinetochore (mean entropy over the 73-kb region = 1.91) (Extended Data Fig. 8a, Methods).

To understand the long-range organization and evolution of the centromere, we generated a pairwise sequence identity heat map, which compares the sequence identity of 5-kb fragments along the length of the centromere (Fig. 2a, Supplementary Fig. 3). We find that the centromere consists of five major evolutionary layers that show mirror symmetry. The outermost layer resides in the monomeric α-satellite, where sequences are highly divergent from the rest of the centromere but are more similar to each other (Fig. 2a, arrow 1). The second layer defines the monomeric-to-HOR transition and is a short (57–60 kb) region. The p and q regions are 87–92% identical with each other but only 78% or less with other centromeric satellites (Fig. 2a, arrow 2). The third layer is completely composed of HORs. The p and q regions are 92 and 149 kb in length, respectively, and share more than 96% sequence identity with each other (Fig. 2a, arrow 3) but less than that with the rest of the centromere. This layer consists largely of homogenous 11-monomer HORs and defines the transition from unmethylated to methylated DNA. The fourth layer is the largest and defines the bulk of the α-satellite HORs (1.42 Mb in total). It shows the greatest variety of HOR subtypes and, once again, the p and q blocks share identity with each other but are more divergent from the remaining layers (Fig. 2a, arrow 4). Finally, the fifth layer encompasses the centre-most 416 kb of the HOR array—a region of near-perfect sequence identity that is divergent from the rest of the centromere (Fig. 2a, arrow 5).

## Sequence resolution of the chromosome 8q21.2 VNTR

The layered and mirrored nature of the chromosome 8 centromere is reminiscent of another GRCh38 gap region located at chromosome 8q21.2 (Extended Data Fig. 1). This region is a cytogenetically recognizable euchromatic variant[22] that contains one of the largest VNTRs in the human genome[22]. The 12.192-kb repeating unit carries the REXO1L1 (also known as GOR) pseudogene and is highly copy number polymorphic among humans[22,23]. This VNTR is of biological interest because it is the site of a recurrent neocentromere, in which a functional centromere devoid of α-satellite has been observed in several unrelated individuals[24,25]. Using our approach, we successfully assembled the VNTR into an 863.5-kb sequence composed of approximately 71 repeating units (67 complete and 7 partial units) (Extended Data Fig. 1a). A pulsed-field gel Southern blot confirms the VNTR length and structure (Extended Data Fig. 1a, b), and chromatin fibre FISH estimates 67 ± 5.2 (mean ± s.d.) repeat units, consistent with the assembly (Extended Data Fig. 10, Methods). Among humans, the repeat unit varies from 53 to 326 copies, creating tandem repeat arrays ranging from 652 kb to 3.97 Mb (Extended Data Fig. 1c). The higher-order structure of the VNTR consists of five distinct domains that alternate in orientation (Extended Data Fig. 1a), in which each domain contains 5 to 23 complete

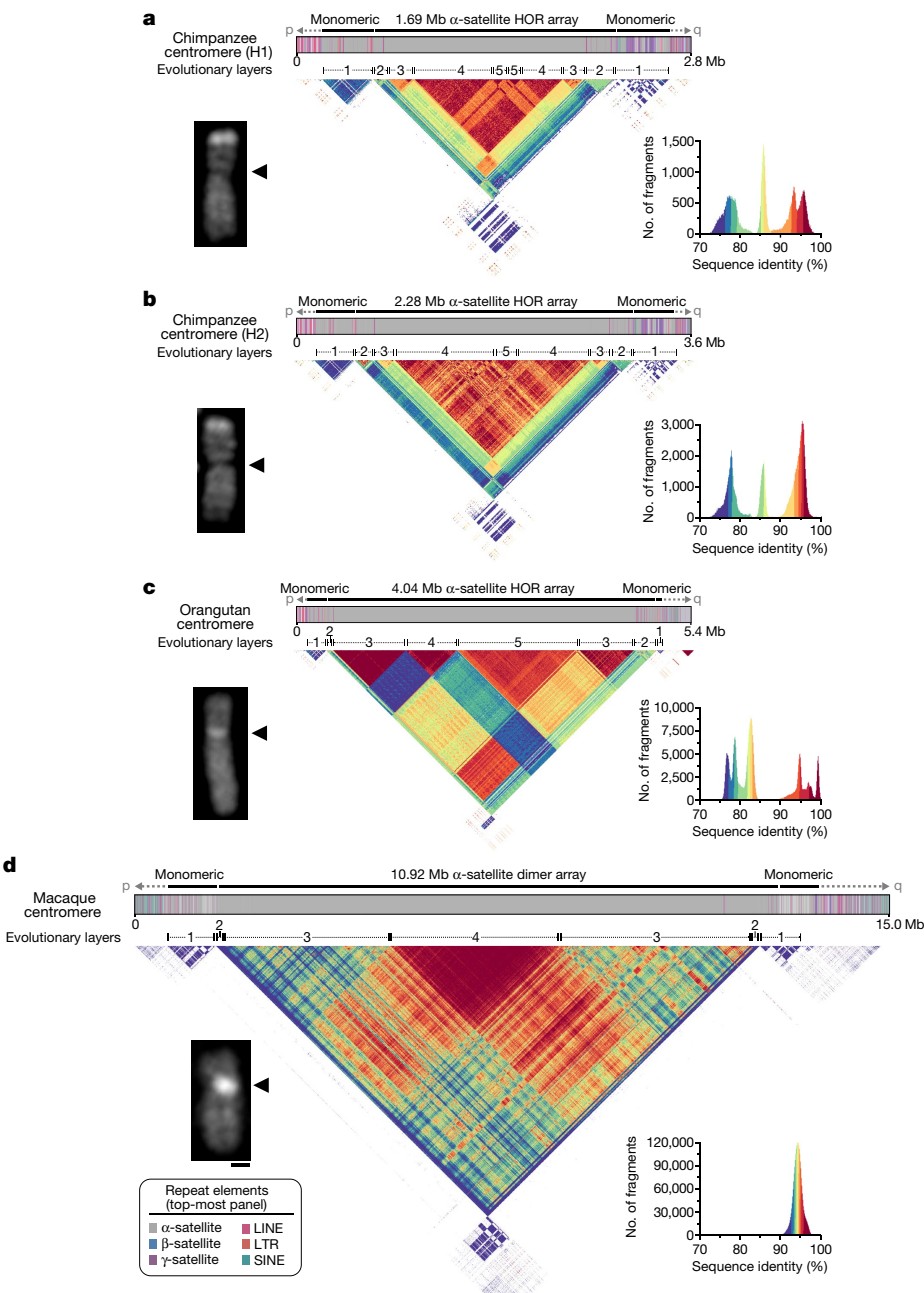

**Fig. 3 | Sequence and structure of the chimpanzee, orangutan, and macaque chromosome 8 centromeres. a–d**, Structure and sequence identity of the chimpanzee (H1) (**a**), chimpanzee (H2) (**b**), orangutan (**c**) and macaque (**d**) chromosome 8 centromeres. Each centromere has a mirrored organization consisting of four or five distinct evolutionary layers. The size of each centromeric region is consistent with microscopic analyses, showing increasingly bright DAPI staining with increasing centromere size. See Supplementary Figs. 10 and 11 for sequence identity heat maps plotted on the same colour scale. H1, haplotype 1; H2, haplotype 2. Scale bar, 1 μm.

repeat units that are more than 98.5% identical to each other (Extended Data Fig. 1a). Detection of methylated cytosine residues[18] shows that each 12.192-kb repeat is primarily methylated in the 3-kb region that corresponds to *REXO1L1* (also known as *GOR1*), whereas the rest of the repeat unit is hypomethylated (Extended Data Fig. 1a). Mapping of centromeric chromatin from a cell line that contains an 8q21.2 neocentromere[25] shows that approximately 98% of CENP-A nucleosomes map to the hypomethylated region of the repeat unit in the CHM13 assembly (Extended Data Fig. 1a). Although this is consistent with the VNTR being the potential site of the functional kinetochore of the neocentromere, sequence and assembly of this and other neocentromere-containing cell lines is vitally important.

## Centromere evolutionary reconstruction

In an effort to fully reconstruct the evolutionary history of the chromosome 8 centromere over the past 25 million years, we applied the same approach to reconstruct the orthologous centromeres in chimpanzee, orangutan and macaque. We first generated 40- to 56-fold ONT data and 25- to 40-fold PacBio HiFi data of each nonhuman primate (NHP) genome (Supplementary Table 5). Using this data, we generated two contiguous draft assemblies of the chimpanzee chromosome 8 centromere (one for each haplotype) and one haplotype assembly from the orangutan and macaque chromosome 8 centromeres (Fig. 3). Mapping of long-read data to each assembly shows uniform coverage,

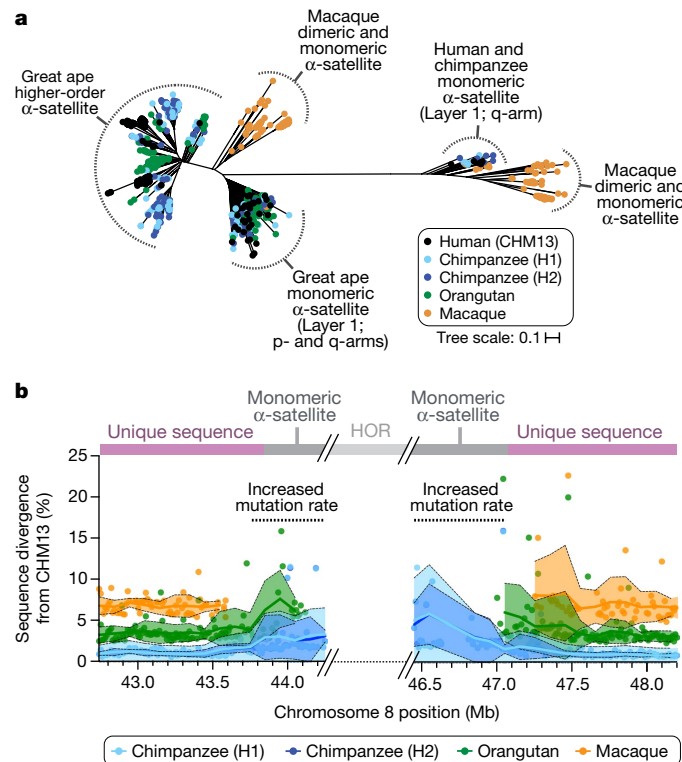

**Fig. 4 | Evolution of the chromosome 8 centromere. a**, Phylogenetic tree of human, chimpanzee, orangutan and macaque α-satellites from the chromosome 8 centromeric regions (Supplementary Fig. 6a, b). **b**, Plot showing the sequence divergence between CHM13 and nonhuman primates in the regions flanking the chromosome 8 α-satellite HOR array. See Supplementary Fig. 6d for a model of centromere evolution.

indicating a lack of large structural errors (Supplementary Figs. 4, 5). Assessment of base accuracy indicates that the assemblies are 99.9988–100% accurate (quality value score > 49.3) (Methods). Analysis of each NHP chromosome 8 centromere reveals distinct HOR arrays ranging in size from 1.69 Mb in chimpanzee to 10.92 Mb in macaque, consistent with estimates from short-read sequence data and cytogenetic analyses[26,27] (Fig. 3). Our data, once again, reveal a mirrored and layered organization, with the chimpanzee organization being most similar to human (Figs. 2a, 3). Each NHP chromosome 8 centromere is composed of four or five distinct layers, with the outermost layer showing the lowest degree of sequence identity (73–78% in chimpanzee and orangutan; 90–92% in macaque) and the innermost layer showing the highest sequence identity (90–100% in chimpanzee and orangutan; 94–100% in macaque). The orangutan structure is notable in that there appears to be very little admixture of HOR units between the layers, in contrast to other apes in which the different HOR cassettes are derived from a major HOR structure. The blocks of orangutan HORs (with the exception of layer 3) show reduced sequence identity. This suggests that the orangutan centromere evolved as a mosaic of independent HOR units. In contrast to all apes, the macaque lacks HORs and, instead, contains a basic dimeric repeat structure[26], which is much more homogenous and highly identical (>90%) across the nearly 11 Mb of assembled centromeric array.

Phylogenetically, we find that all great ape higher-order α-satellite sequences (corresponding to layers 2–5) cluster into a single clade, and the monomeric α-satellite (layer 1) split into two clades separated by tens of millions of years (Fig. 4a). The proximal clade contains monomeric α-satellite from both the p- and q-arms, whereas the more divergent clade shares monomeric α-satellite solely from the q-arm, and specifically, the α-satellite nestled between clusters

of γ-satellite (Supplementary Fig. 6a, b). Unlike great apes, both monomeric and dimeric repeat structures from the macaque group together and are sister clades to the monomeric ape clades, which suggests a common ancient origin restricted to these flanking pericentromeric regions. We used the orthology of flanking primate sequences to understand how rapidly sequences decay over the course of evolution. We assessed divergence based on 10-kb windows of pairwise alignments in the approximately 2-Mb flanking the α-satellite HOR array (Fig. 4b). We find that the mean allelic divergence increases more than threefold as the sequence transitions from unique to monomeric α-satellite. Such increases are rare in the human genome, in which only 1.27–1.99% of nearly 20,000 random loci show comparable levels of divergence (Supplementary Fig. 6c). Using evolutionary models (Methods), we estimate a minimal mutation rate of the chromosome 8 centromeric region to be approximately $4.8 \times 10^{-8}$ and $8.4 \times 10^{-8}$ mutations per base pair per generation on the p- and q-arms, respectively, which is 2.2- to 3.8-fold higher than the basal mean mutation rate (approximately $2.2 \times 10^{-8}$) (Supplementary Table 6). These analyses provide a complete comparative sequence analysis of a primate centromere for an orthologous chromosome and a framework for future studies of genetic variation and evolution of these regions across the genome.

## Discussion

Chromosome 8 is the first human autosome to be sequenced and assembled from telomere to telomere and contains only the third completed human centromere[13,28], to our knowledge. Both chromosome 8 and X centromeres (Supplementary Fig. 7) contain a pocket of hypomethylation (approximately 61–73 kb in length), and we show that this region is enriched for the centromeric histone CENP-A, consistent with the functional kinetochore-binding site[29,30]. Notably, enrichment of CENP-A extends over a broader swath of sequence (632 kb), with its peak centred over the hypomethylated region composed of diverse HORs. The layered and mirrored organization of the chromosome 8 centromere supports a model of evolution[31–33], in which highly identical repeats expand, pushing older, more divergent repeats to the edges in an assembly-line fashion (Supplementary Fig. 6d). The chromosome 8 centromere reveals five such layers, and this organization is generally identified in other NHP centromeres. We confirm that HOR structures evolved after apes diverged from Old World monkeys (less than 25 million years ago)[26,34,35] but also distinguish different classes of monomeric repeats that share an ancient origin with the Old World monkeys. One ape monomeric clade (present only in the q-arm) groups with the clade of the macaques (Supplementary Fig. 6a, b). We hypothesize that this approximately 70-kb segment present in chimpanzee and human, but absent in orangutan, represents the remnants of the ancestral centromere. Sequence comparisons show that mutation rates increase by at least two to fourfold in proximity to the HOR array, probably owing to the action of concerted evolution, unequal crossing-over, and saltatory amplification[33,36,37]. Among three human centromere 8 haplotypes, we identify regions of excess allelic variation and structural divergence (Extended Data Fig. 7), and these locations differ among haplotypes. Nevertheless, the first sequence of a complete human genome is imminent, and the next challenge will be applying the methods to fully phase and assemble diploid genomes[38–40].

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

## Methods

### Data reporting
No statistical methods were used to predetermine sample size. The experiments were not randomized, and investigators were not blinded to allocation during experiments and outcome assessment.

### Cell line sources
CHM13hTERT (CHM13) cells were originally isolated from a hydatidiform mole at Magee-Womens Hospital as part of a research study (IRB MWH-20-054). Cryogenically frozen cells from this culture were grown and transformed with the human telomerase reverse transcriptase (TERT) gene to immortalize the cell line. This cell line has been authenticated by STR analysis, tested negative for mycoplasma contamination, and karyotyped to show a 46,XX karyotype[13]. Human HG00733 lymphoblastoid cells were originally obtained from a female Puerto Rican child, immortalized with the Epstein-Barr virus (EBV), and stored at the Coriell Institute for Medical Research. Chimpanzee (*Pan troglodytes*; Clint; S006007) fibroblast cells were originally obtained from a male western chimpanzee named Clint (now deceased) at the Yerkes National Primate Research Center and immortalized with EBV. Orangutan (*Pongo abelii*; Susie; PR01109) fibroblast cells were originally obtained from a female Sumatran orangutan named Susie (now deceased) at the Gladys Porter Zoo, immortalized with EBV, and stored at the Coriell Institute for Medical Research. Macaque (*Macaca mulatta*; AG07107) fibroblast cells were originally obtained from a female rhesus macaque of Indian origin and stored at the Coriell Institute for Medical Research. The HG00733, chimpanzee, orangutan and macaque cell lines have not yet been authenticated or assessed for mycoplasma contamination, to our knowledge.

### Cell culture
CHM13 cells were cultured in complete AmnioMax C-100 Basal Medium (Thermo Fisher Scientific, 17001082) supplemented with 15% AmnioMax C-100 Supplement (Thermo Fisher Scientific, 12556015) and 1% penicillin-streptomycin (Thermo Fisher Scientific, 15140122). HG00733 cells were cultured in RPMI 1640 with L-glutamine (Thermo Fisher Scientific, 11875093) supplemented with 15% FBS (Thermo Fisher Scientific, 16000-044) and 1% penicillin-streptomycin (Thermo Fisher Scientific, 15140122). Chimpanzee (*P. troglodytes*; S006007) and macaque (*M. mulatta*; AG07107) cells were cultured in MEMα containing ribonucleosides, deoxyribonucleosides and L-glutamine (Thermo Fisher Scientific, 12571063) supplemented with 12% FBS (Thermo Fisher Scientific, 16000-044) and 1% penicillin-streptomycin (Thermo Fisher Scientific, 15140122). Orangutan (*P. abelii*; PR01109) cells were cultured in MEMα containing ribonucleosides, deoxyribonucleosides and L-glutamine (Thermo Fisher Scientific, 12571063) supplemented with 15% FBS (Thermo Fisher Scientific, 16000-044) and 1% penicillin-streptomycin (Thermo Fisher Scientific, 15140122). All cells were cultured in a humidity-controlled environment at 37 °C with 5% $CO_2$.

### DNA extraction, library preparation and sequencing
PacBio HiFi data were generated from the HG00733, chimpanzee, orangutan and macaque genomes as previously described[36] with modifications. In brief, high-molecular-weight (HMW) DNA was extracted from cells using a modified Qiagen Gentra Puregene Cell Kit protocol[37]. HMW DNA was used to generate HiFi libraries via the SMRTbell Express Template Prep Kit v2 and SMRTbell Enzyme Clean Up kits (PacBio). Size selection was performed with SageELF (Sage Science), and fractions sized 11, 14, 18, 22, or 25 kb (as determined by FEMTO Pulse (Agilent)) were chosen for sequencing. Libraries were sequenced on the Sequel II platform (Instrument Control SW v7.1 or v8.0) with three to seven SMRT Cells 8M (PacBio) using either Sequel II Sequencing Chemistry 1.0 and 12-h pre-extension or Sequel II Sequencing Chemistry 2.0 and 3- or 4-h pre-extension, both with 30-h movies, aiming for a minimum estimated coverage of 25× in HiFi reads (assuming a genome size of 3.2 Gb). Raw data were processed using the CCS algorithm (v.3.4.1 or v.4.0.0) with the following parameters: –minPasses 3 –minPredictedAccuracy 0.99 –maxLength 21000 or 50000.

Ultra-long ONT data were generated from the CHM13, HG00733, chimpanzee and orangutan genomes according to a previously published protocol[41]. In brief, $5 \times 10^7$ cells were lysed in a buffer containing 10 mM Tris-Cl (pH 8.0), 0.1 M EDTA (pH 8.0), 0.5% (w/v) SDS, and 20 µg ml$^{-1}$ RNase A for 1 h at 37 °C. Proteinase K (200 µg ml$^{-1}$) was added, and the solution was incubated at 50 °C for 2 h. DNA was purified via two rounds of 25:24:1 phenol-chloroform-isoamyl alcohol extraction followed by ethanol precipitation. Precipitated DNA was solubilized in 10 mM Tris (pH 8) containing 0.02% Triton X-100 at 4 °C for two days. Libraries were constructed using the Rapid Sequencing Kit (SQK-RAD004) from ONT with modifications to the manufacturer's protocol. Specifically, 2–3 µg of DNA was resuspended in a total volume of 18 µl with 16.6% FRA buffer. FRA enzyme was diluted 2- to 12-fold into FRA buffer, and 1.5 µl of diluted FRA was added to the DNA solution. The DNA solution was incubated at 30 °C for 1.5 min, followed by 8 °C for 1 min to inactivate the enzyme. RAP enzyme was diluted 2- to 12-fold into RAP buffer, and 0.5 µl of diluted RAP was added to the DNA solution. The DNA solution was incubated at room temperature for 2 h before loading onto a primed FLO-MIN106 R9.4.1 flow cell for sequencing on a GridION using MinKNOW (v.2.0 - v1.9.12).

Additional ONT data were generated from the CHM13, HG00733, chimpanzee, orangutan, and macaque genomes. In brief, HMW DNA was extracted from cells using a modified Qiagen Gentra Puregene Cell Kit protocol[37]. HMW DNA was prepared into libraries with the Ligation Sequencing kit (SQK-LSK109) from ONT and loaded onto primed FLO-MIN106 or FLO-PRO002 R9.4.1 flow cells for sequencing on a GridION or PromethION, respectively, using MinKNOW (v.2.0–v.19.12). All ONT data were base called with Guppy 3.6.0 or 4.0.11 with the HAC model.

### PacBio HiFi whole-genome assembly
The CHM13 genome was assembled from PacBio HiFi data using HiCanu[5] as previously described[5]. The HG00733 genome was assembled from PacBio HiFi data (Supplementary Table 3) using hifiasm[6] (v.0.7). The chimpanzee, orangutan and macaque genomes were assembled from PacBio HiFi data (Supplementary Table 5) using HiCanu[5] (v.2.0). Contigs from each assembly were used to replace the ONT-based sequence scaffolds in targeted regions (described below).

### Targeted sequence assembly
Gapped regions within human chromosome 8 were targeted for assembly via a SUNK-based method that combines both PacBio HiFi and ONT data. Specifically, CHM13 PacBio HiFi data were used to generate a library of SUNKs ($k = 20$; total = 2,062,629,432) via Jellyfish (v.2.2.4) on the basis of the sequencing coverage of the HiFi dataset. In total, 99.88% (2,060,229,331) of the CHM13 PacBio HiFi SUNKs were validated with CHM13 Illumina data (SRR3189741). A subset of CHM13 ultra-long ONT reads aligning to the CHM1 β-defensin patch (GenBank: KZ208915.1) or select regions within the GRCh38 chromosome 8 reference sequence (chr8:42,881,543–47,029,467 for the centromere and chr8:85,562,829–85,848,463 for the 8q21.2 locus) were barcoded with Illumina-validated SUNKs. Reads sharing at least 50 SUNKs were selected for inspection to determine whether their SUNK barcodes overlapped. SUNK barcodes can be composed of 'valid' and 'invalid' SUNKs. Valid SUNKs are those that occur once in the genome and are located at the exact position on the read. By contrast, invalid SUNKs are those that occur once in the genome but are falsely located at the position on the read, and this may be due to a sequencing or base-calling error, for example. Valid SUNKs were identified within the barcode as those that share pairwise distances with at least ten other SUNKs on the same read. Reads that shared a SUNK barcode containing at least three valid SUNKs

and their corresponding pairwise distances (±1% of the read length) were assembled into a tile. The process was repeated using the tile and subsetted ultra-long ONT reads several times until a sequence scaffold spanning the gapped region was generated. Validation of the scaffold organization was carried out via three independent methods. First, the sequence scaffold and underlying ONT reads were subjected to RepeatMasker (v.3.3.0) to ensure that read overlaps were concordant in repeat structure. Second, the centromeric scaffold and underlying ONT reads were subjected to StringDecomposer[42] to validate the HOR organization in overlapping reads. Finally, the sequence scaffold for each target region was incorporated into the CHM13 chromosome 8 assembly previously generated[5], thereby filling the gaps in the chromosome 8 assembly. CHM13 PacBio HiFi and ONT data were aligned to the entire chromosome 8 assembly via pbmm2 (v.1.1.0) (for PacBio data; https://github.com/PacificBiosciences/pbmm2) or Winnowmap[43] (v.1.0) (for ONT data) to identify large collapses or misassemblies. Although the ONT-based scaffolds are structurally accurate, they are only 87–98% accurate at the base level owing to base-calling errors in the raw ONT reads[7]. Therefore, we sought to improve the base accuracy of the sequence scaffolds by replacing the ONT sequences with PacBio HiFi contigs assembled from the CHM13 genome[5], which have a consensus accuracy greater than 99.99%[5]. Therefore, we aligned CHM13 PacBio HiFi contigs generated via HiCanu[5] to the chromosome 8 assembly via minimap2[44] (v2.17-r941; parameters: minimap2 -t 8 -I 8G -a --eqx -x asm20 -s 5000) to identify contigs that share high sequence identity with the ONT-based sequence scaffolds. A typical scaffold had multiple PacBio HiFi contigs that aligned to regions within it. Therefore, the scaffold was used to order and orient the PacBio HiFi contigs and bridge gaps between them when necessary. PacBio HiFi contigs with high sequence identity replaced almost all regions of the ONT-based scaffolds: ultimately, the chromosome 8 assembly consists of 146,254,195 bp of PacBio HiFi contigs and only 5,490 bp of ONT sequence scaffolds (99.9963% PacBio HiFi contigs and 0.0037% ONT scaffold). The chromosome 8 assembly was incorporated into a whole-genome assembly of CHM13 previously generated[5] for validation via orthogonal methods (detailed below). The HG00733, chimpanzee, orangutan and macaque chromosome 8 centromeres were assembled via the same SUNK-based method.

## Accuracy estimation

The accuracy of the CHM13 chromosome 8 assembly was estimated from mapped $k$-mers using Merqury[17]. In brief, Merqury (v.1.1) was run on the chromosome 8 assembly with the following command: eval/qv.sh CHM13.k21.meryl chr8.fasta chr8_v9.

CHM13 Illumina data (SRR1997411, SRR3189741, SRR3189742 and SRR3189743) were used to identify $k$-mers with $k = 21$. In Merqury, every $k$-mer in the assembly is evaluated for its presence in the Illumina $k$-mer database, with any $k$-mer missing in the Illumina set counted as base-level 'error'. We detected 1,474 $k$-mers found only in the assembly out of 146,259,650, resulting in a quality value score of 63.19, estimated as follows: $-10 \times \log(1 - (1 - 1{,}474/146{,}259{,}650)^{(1/21)}) = 63.19$.

The accuracy percentage for chromosome 8 was estimated from this quality value score as: $100 - (10^{(63.19/-10)}) \times 100 = 99.999952$.

The accuracy of the CHM13 chromosome 8 assembly and β-defensin locus were also estimated from sequenced BACs. In brief, 66 BACs from the CHM13 chromosome 8 (BAC library VMRC59) were aligned to the chromosome 8 assembly via minimap2[44] (v2.17-r941) with the following parameters: -I 8G -2K 1500m --secondary = no -a --eqx -Y -x asm20 -s 200000 -z 10000,1000 -r 50000 -O 5,56 -E 4,1 -B 5. The quality value was then estimated using the CIGAR string in the resulting BAM, counting alignment differences as errors according to the following formula:

$$\text{Quality value} = -10 \times \log_{10}[1 - (\text{matches}/$$
$$(\text{mismatches} + \text{matches} + \text{insertions} + \text{deletions}))]$$

The median quality value was 40.6988 for the entire chromosome 8 assembly and 40.4769 for the β-defensin locus (chr8:6300000–13300000; estimated from 47 individual BACs) (see Extended Data Fig. 5 for more details), which falls within the 95% confidence interval for the whole chromosome. This quality value score was used to estimate the base accuracy[36] as follows:

$$100 - (10^{(40.6988/-10)}) \times 100 = 99.9915$$

$$100 - (10^{(40.4769/-10)}) \times 100 = 99.9910$$

The BAC quality value estimation should be considered a lower bound, because differences between the BACs and the assembly may originate from errors in the BAC sequences themselves. BACs were previously shown to occasionally contain sequencing errors that are not supported by the underlying PacBio HiFi reads[36]. In addition, the upper bound for the estimated BAC quality value is limited to approximately 53, because BACs are typically 200 kb and, as a result, the maximum calculable quality value is 1 error in 200 kb (quality value 53). We also note that the quality value of the centromeric region could not be estimated from BACs owing to biases in BAC library preparation, which preclude centromeric sequences in BAC clones.

The accuracy of the HG00733, chimpanzee, orangutan and macaque chromosome 8 centromere assemblies was estimated with Merqury[17]. In brief, Merqury (v.1.1) was run on the centromere assemblies as described above for the CHM13 chromosome 8 assembly. Ultimately, we detected 248 $k$-mers found only in the HG00733 maternal assembly out of 3,877,376 bp (estimated quality value score of 55.16; base accuracy of 99.9997%); 10,562 $k$-mers found only in the HG00733 paternal assembly out of 3,597,645 bp (estimated quality value score of 38.54; base accuracy of 99.986%); 0 $k$-mers found only in the chimpanzee H1 assembly out of 2,803,083 bp (estimated quality value score of infinity; base accuracy of 100%); 20 $k$-mers found only in the chimpanzee H2 assembly out of 3,603,864 bp (estimated quality value score of 65.7796; base accuracy of 99.9999%); 1,302 $k$-mers found only in the orangutan assembly out of 5,372,621 bp (estimated quality value score of 49.3774; accuracy of 99.9988%); and 104 $k$-mers found only in the macaque assembly out of 14,999,980 bp (estimated quality value score of 64.8128; accuracy of 99.9999%). We note that Merqury detects the presence of erroneous $k$-mers in the assembly that have no support within the raw reads, but it cannot detect the absence of true $k$-mers (variants) within the assembled repeat copies. Thus, within these highly repetitive arrays, Merqury is useful for comparative analyses but may overestimate the overall accuracy of the consensus.

## Strand-seq analysis

We evaluated the directional and structural contiguity of CHM13 chromosome 8 assembly, including the centromere, using Strand-seq data. First, all Strand-seq libraries produced from the CHM13 genome[36] were aligned to the CHM13 assembly, including chromosome 8 using BWA-MEM[45] (v.0.7.17-r1188) with default parameters for paired-end mapping. Next, duplicate reads were marked by sambamba[46] (v.0.6.8) and removed before subsequent analyses. We used SAMtools[47] (v.1.9) to sort and index the final BAM file for each Strand-seq library. To detect putative misassembly breakpoints in the chromosome 8 assembly, we ran breakpointR[48] on all BAM files to detect strand-state breakpoints. Misassemblies are visible as recurrent changes in strand state across multiple Strand-seq libraries[39]. To increase our sensitivity of misassembly detection, we created a 'composite file' that groups directional reads across all available Strand-seq libraries[49,50]. Next, we ran breakpointR on the 'composite reads file' using the function 'runBreakpointr' to detect regions that are homozygous ('ww'; 'HOM' - all reads mapped in minus orientation) or heterozygous inverted ('wc', 'HET' - approximately equal number of reads mapped in minus and plus orientation). To further detect any putative chimaerism in the chromosome 8 assembly, we

applied Strand-seq to assign 200-kb long chunks of the chromosome 8 assembly to unique groups corresponding to individual chromosomal homologues using SaaRclust[39,51]. For this, we used the SaaRclust function 'scaffoldDenovoAssembly' on all BAM files.

### Bionano analysis

Bionano Genomics data were generated from the CHM13 genome[13]. Long DNA molecules labelled with Bionano's Direct Labelling Enzyme were collected on a Bionano Saphyr Instrument to a coverage of 130×. The molecules were assembled with the Bionano assembly pipeline Solve (v.3.4), using the nonhaplotype-aware parameters and GRCh38 as the reference. The resulting data produced 261 genome maps with a total length of 2,921.6 Mb and a genome map N50 of 69.02 Mb.

The molecule set and the nonhaplotype-aware map were aligned to the CHM13 draft assembly and the GRCh38 assembly, and discrepancies were identified between the Bionano maps and the sequence references using scripts in the Bionano Solve software package—runCharacterize. py, runSV.py, and align_bnx_to_cmap.py.

A second version of the map was assembled using the haplotype-aware parameters. This map was also aligned to GRCh38 and the final CHM13 assembly to verify heterozygous locations. These regions were then examined further.

Analysis of Bionano alignments revealed three heterozygous sites within chromosome 8 located at approximately chr8:21,025,201, chr8:80,044,843 and chr8:121,388,618 (Supplementary Table 7). The structure with the greatest ONT read support was selected for inclusion in the chromosome 8 assembly (Supplementary Table 7).

### TandemMapper and TandemQUAST analysis of the centromeric HOR array

We assessed the structure of the CHM13 and NHP centromeric HOR arrays by applying TandemMapper and TandemQUAST[52] (https://github.com/ablab/TandemTools; version from 20 March 2020), which can detect large structural assembly errors in repeat arrays. For the CHM13 centromere, we first aligned ONT reads longer than 50 kb to the CHM13 assembly containing the contiguous chromosome 8 with Winnowmap[43] (v.1.0) and extracted reads aligning to the centromeric HOR array (chr8:44243868–46323885). We then inputted these reads in the following TandemQUAST command: tandemquast.py -t 24 --nano {ont_reads.fa} -o {out_dir} chr8.fa. For the NHP centromeres, we aligned ONT reads to the whole-genome assemblies containing the contiguous chromosome 8 centromeres with Winnowmap[43] (v.1.0) and extracted reads aligning to the centromeric HOR arrays. We then inputted these reads in the following TandemQUAST command: tandemquast.py -t 24 --nano {ont_reads.fa} -o {out_dir} chr8.fa.

### Methylation analysis

Nanopolish[18] (v.0.12.5) was used to measure the frequency of CpG methylation from raw ONT reads (>50 kb in length for CHM13) aligned to whole-genome assemblies via Winnowmap[43] (v.1.0). Nanopolish distinguishes 5-methylcytosine from unmethylated cytosine via a Hidden Markov model (HMM) on the raw nanopore current signal. The methylation caller generates a log-likelihood value for the ratio of probability of methylated to unmethylated CpGs at a specific $k$-mer. We filtered methylation calls using the nanopore_methylation_utilities tool (https://github.com/isaclee/nanopore-methylation-utilities)[53], which uses a log-likelihood ratio of 2.5 as a threshold for calling methylation. CpG sites with log-likelihood ratios greater than 2.5 (methylated) or less than −2.5 (unmethylated) are considered high quality and included in the analysis. Reads that do not have any high-quality CpG sites are filtered from the BAM for subsequent methylation analysis. Nanopore_methylation_utilities integrates methylation information into the BAM file for viewing in IGV[54] bisulfite mode, which was used to visualize CpG methylation.

### Iso-Seq data generation and sequence analyses

RNA was purified from approximately $1 \times 10^7$ CHM13 cells using an RNeasy kit (Qiagen; 74104) and prepared into Iso-Seq libraries following a standard protocol[55]. Libraries were loaded on two SMRT Cells 8M and sequenced on the Sequel II. The data were processed via isoseq3 (v.8.0), ultimately generating 3,576,198 full-length non-chimeric reads. Poly-A trimmed transcripts were aligned to this CHM13 chr8 assembly and to GRCh38 with minimap2[44] (v.2.17-r941) with the following parameters: -ax splice -f1000 --sam-hit-only --secondary = no --eqx. Transcripts were assigned to genes using featureCounts[56] with GENCODE[57] (v.34) annotations, supplemented with CHESS v.2.2[58] for any transcripts unannotated in GENCODE. Each transcript was scored for the percentage identity of its alignment to each assembly, requiring 90% of the length of each transcript to align to the assembly for it to count as aligned. For each gene, the percentage identity of non-CHM13 transcripts to GRCh38 was compared to the CHM13 chromosome 8 assembly. Genes with an improved representation in the CHM13 assembly were identified with a cut-off value of 20 improved reads per gene, with at least 0.2% average improvement in percentage identity. GENCODE (v.34) transcripts were lifted over to the CHM13 chr8 assembly using Liftoff[59] to compare the GRCh38 annotations to this assembly and Iso-Seq transcripts.

We combined the 3.6 million full-length transcript data (above) with 20,937,742 full-length non-chimeric publicly available human Iso-Seq data (Supplementary Table 8). In total, we compared the alignment of 24,513,940 full-length non-chimeric reads from 13 tissue and cell line sources to both the completed CHM13 chromosome 8 assemblies and the current human reference genome, GRCh38. Of the 848,048 non-CHM13 cell line transcripts that align to chromosome 8, 93,495 (11.02%) align with at least 0.1% greater percentage identity to the CHM13 assembly, and 52,821 (6.23%) to GRCh38. This metric suggests that the chromosome 8 reference improves human gene annotation by approximately 4.79% even though most of those changes are subtle in nature. Overall, 61 protein-coding and 33 noncoding loci have improved alignments to the CHM13 assembly compared to GRCh38, with >0.2% average percentage identity improvement, and at least 20 supporting transcripts (Extended Data Fig. 3a–c, Supplementary Table 1). As an example, *WDYHV1* (also known as *NTAQ1*) has four amino acid replacements, with 13 transcripts sharing the identical open reading frame to CHM13 (Extended Data Fig. 3d).

### Pairwise sequence identity heat maps

To generate pairwise sequence identity heat maps, we fragmented the centromere assemblies into 5-kb fragments (for example, 1–5,000, 5,001–10,000, and so on) and made all possible pairwise alignments between the fragments using the following minimap2[44] (v.2.17-r941) command: minimap2 -f 0.0001 -t 32 -X --eqx -ax ava-ont. The sequence identity was determined from the CIGAR string of the alignments and then visualized using ggplot2 (geom_raster) in R (v.1.1.383)[60]. The colour of each segment was determined by sorting the data by identity and then creating 10 equally sized bins, each of which received a distinct colour from the spectral pallet. The choice of a 5-kb window came after testing a variety of window sizes. Ultimately, we found 5 kb to be a good balance between resolution of the figure (because each 5 kb fragment is plotted as a pixel) and sensitivity of minimap2 (fragments less than 5 kb often missed alignments with the ava-ont preset). A schematic illustrating this process is shown in Supplementary Fig. 3.

### Miropeats analysis

To compare the organization and orientation of the CHM13 and GRCh38 β-defensin loci, we aligned the two β-defensin regions (CHM13 chr8:6300000–13300000; GRCh38 chr8:6545299–13033398) to each other using the following minimap2[44] parameters: minimap2 -x asm20 -s 200000 -p 0.01 -N 1000 --cs {GRCh38_defensin.fasta} {CHM13_defensin.fasta}. Then, we applied a version of Miropeats[61] that is modified to

use minimap2[44] alignments (https://github.com/mrvollger/minimiro) to produce the figure showing homology between the two sequences.

## Analysis of α-satellite organization

To determine the organization of the CHM13 chromosome 8 centromeric region, we used two independent approaches. First, we subjected the CHM13 centromere assembly to an in silico restriction enzyme digestion in which a set of restriction enzyme recognition sites were identified within the assembly. In agreement with previous findings that XbaI digestion can generate a pattern of HORs within the chromosome 8 HOR array[9], we found that each α-satellite HOR could be extracted via XbaI digestion. The in silico digestion analysis indicates that the chromosome 8 centromeric HOR array consists of 1,462 HOR units: 283 4-monomer HORs, 4 5-monomer HORs, 13 6-monomer HORs, 356 7-monomer HORs, 295 8-monomer HORs, and 511 11-monomer HORs. As an alternative approach, we subjected the centromere assembly to StringDecomposer[42] (https://github.com/ablab/stringdecomposer; version from 28 February 2020) using a set of 11 α-satellite monomers derived from a chromosome 8 11-mer HOR unit. The sequence of the α-satellite monomers used are as follows: A: AGCATTCTCAGAAACACCTTCGTGATGTTTGCAATCAAGTCACAGAGTTGAACCTTCCGTTTCATAGAGCAGGTTGGAAACA CTCTTATTGTAGTATCTGGAAGTGGACATTTGGAGCGCTTTCAGGCCTATGGTGAAAAAGGAAATATCTTCCCATAAAAACGACATAGA; B: AGCTATCTCAGGAACTTGTTTATGATGCATCTAATCAACTAACAGTGTTGAACCTTTGTACTGACAGAGCACTTTGAAACACTCTTTTTTGGAATCTGCAAGTGGATATTTGGATCGCTTTGAGGATTTCGTTGGAAACGGGATGCAATATAAAACGTACACAGC; C: AGCATACTCAGAAAATACTTTGCCATATTTCCATTCAAGTCACAGAGTGGAACATTCCCATTCATAGAGCAGGTTGGAAACACTCTTTTTGGAGTATCTGGAAGTGGACATTTGGAGCGCTTTCTGAACTATGGTGAAAAAGGAAATATCTTCCAATGAAAACAAGACAGA; D: AGCATTCTGAGAAACTTATTTGTGATGTGTGTCCTCAACAAACGGACTTGAACCTTTCGTTTCATGCAGTACTTCTGGAACACTCTTTTTGAAGATTCTGCATGCGGATATTTGGATAGCTTTGAGGATTTCGTTGGAAACGGCTTACATGTAAAAATTAGACAGC; E: AGCATTCTCAGAAACTTCTTTGTGGTGTCTGCATTCAAGTCACAGAATTGAACTTCTCCTCACATAGAGCAGTTGTGCAGCACTCTATTTGTAGTATCTGGAAGTGGACATTTGGAGGGCTTTGTAGCCTATCTGGAAAAAGGAAATATCTTCCCATGAATGCGAGATAGA; F: AGTAATCTCAGAAACATGTTTATGCTGTATCTACTCAACTAACTGTGCTGAACATTTCTATTGATAGAGCAGTTTTGAGACCCTCTTCTTTTGGAATCTGCAAGTGGATATTTGGATAGATTTGAGGATTTCGTTGGAAACGGGATTATATATAAAAAGTAGACAGC; G: AGCATTCTCAGAAACTTCTTTGTGATGTTTGCATCCAGCTCTCAGAGTTGAACATTCCCTTTCATAGAGTAGGTTTGAAACCCTCTTTTTATAGTGTCTGGAAGCGGGCATTTGGAGCGCTTTCAGGCCTATGCTGAAAAAGGAAATATCTACATATAGAAACTAGACAGA; H: AGCATTCTGAGAATCAAGTTTGTGATGTGGGTACTCAACTAACAGTGTTGATCCATTCTTTTGATACAGCAGTTTTGAACCACACTTTTTGTAGAATCTGCAAGTGGATATTTGGATAGCTGTGAGGATTTCGTTGGAAACGGGAATGTCTTCATAGAAAATTTAGACAGA; I: AGCATTCTCAGAACCTTGATTGTGATGTGTGTTCTCCACTAACAGAGTTGAACCTTTCTTTTGACAGAACTGTTCTGAAACATTCTTTTTATAGAATCTGGAAGTGGATATTTGGAAAGCTTTGAGGATTTCGTTGGAAACGGGAATATCTTCAAATAAAATCTAGCCAGA; J: AGCATTCTAAGAAACATCTTAGGGATGTTTACATTCAAGTCACAGAGTTGAACATTCCCTTTCACAGAGCAGGTTTGAAACAATCTTCTCGTACTATCTGGCAGTGGACATTTTGAGCTCTTTGGGGCCTATGCTGAAAAAGGAAATATCTTCCGACAAAAACTAGTCAGA; K: AGCATTCGCAGAATCCCGTTTGTGATGTGTGCACTCAACTGTCAGAATTGAACCTTGGTTTGGAGAGAGCACTTTTGAAACACACTTTTGTAGAATCTGCAGGTGGATATTTGGCTAGCTTTGAGGATTTCGTTGGAAACGGTAATGTCTTCAAAGAAAATCTAGACAGA.

This analysis indicated that the CHM13 chromosome 8 centromeric HOR array consists of 1,515 HOR units: 286 4-monomer HORs, 12 6-monomer HORs, 366 7-monomer HORs, 303 8-monomer HORs, 3 10-monomer HORs, 539 11-monomer HORs, 2 12-monomer HORs, 2 13-monomer HORs, 1 17-monomer HOR, and 1 18-monomer HOR, which

is concordant with the in silico restriction enzyme digestion results. The predominant HOR types from StringDecomposer[42] are presented in Extended Data Fig. 8.

## Copy number estimation

To estimate the copy number for the 8q21.2 VNTR and *DEFB* loci in human lineages, we applied a read-depth based copy number geno-typer[14] to a collection of 1,105 published high-coverage genomes[62–67]. In brief, sequencing reads were divided into multiples of 36-monomer HORs, which were then mapped to a repeat-masked human reference genome (GRCh38) using mrsFAST[68] (v.3.4.1). To increase the mapping sensitivity, we allowed up to two mismatches per 36-monomer HOR. The read depth of mappable sequences across the genome was corrected for underlying GC content, and copy number estimate for the locus of interest was computed by summarizing over all mappable bases for each sample.

## Entropy calculation

To define regions of increased admixture within the centromeric HOR array, we calculated the entropy using the frequencies of the different HOR units in 10-unit windows (1 unit slide) over the entire array. The following formula was used to determine entropy:

$$\text{Entropy} = -\Sigma(\text{frequency}_i \times \log_2(\text{frequency}_i))$$

in which frequency is: (no. of HORs)/(total no. of HORs) in a 10-unit window. The analysis is analogous to that previously performed[69].

## Droplet digital PCR

Droplet digital PCR was performed on CHM13 genomic DNA to estimate the number of D8Z2 α-satellite HORs, as was previously done for the DXZ1 α-satellite HORs[13]. In brief, genomic DNA was isolated from CHM13 cells using the DNeasy Blood & Tissue Kit (Qiagen). DNA was quantified using a Qubit Fluorometer and the Qubit dsDNA HS Assay (Invitrogen). Reactions (20 μl) were prepared with 0.1 ng of gDNA for the D8Z2 assay or 1 ng of gDNA for the *MTUS1* single-copy gene (as a control). EvaGreen droplet digital PCR (Bio-Rad) master mixes were simultaneously prepared for the D8Z2 and *MTUS1* reactions, which were then incubated for 15 min to allow for restriction digest, according to the manufacturer's protocol.

## Pulsed-field gel electrophoresis and Southern blot

CHM13 genomic DNA was prepared in agarose plugs and digested with either BamHI or MfeI (to characterize the chromosome 8 centromeric region) or BmgBI (to characterize the chromosome 8q21.2 region) in the buffer recommended by the manufacturer. The digested DNA was separated with the CHEF Mapper system (Bio-Rad; autoprogram, 5–850-kb range, 16 h run), transferred to a membrane (Amersham Hybond-N+) and blot-hybridized with a 156 bp probe specific to the chromosome 8 centromeric α-satellite or 8q21.2 region. The probe was labelled with [32]P by PCR-amplifying a synthetic DNA template 233: 5′-TTTGTGGAAGTGGACATTTCGCTTTGTAGCCTATCTGGAAAAAGGAAATATCTTCCCATGAATGCGAGATAGAAGTAATCTCAGAAACATGTTTATGCTGTATCTACTCAACTAACTGTGCTGAACATTTCTATTGTAAAAATAGACAGAAGCATT-3′ (for the centromere of chromosome 8); 264: 5′-TTTGTGGAAGTGGACATTTCG CCCGAGGGGCCGCGGCAGGGATTCCGGGGGACCGGGAGTGGGGGGGTTGGGGTTACTCTTGGCTTTTTGCCCTCTCCTGCCGCCGGCTGCTCCAGTTTCTTTCGCTTTGCGGCGAGGTGGTAAAAATAGACAGAAGCATT-3′ (for the organization of the chromosome 8q21.2 locus) with PCR primers 129: 5′-TTTGTGGAAGTGGACATTTC-3′ and 130: 5′-AATGCTTCTGTCTAT TTTTA-3′. The blot was incubated for 2 h at 65 °C for pre-hybridization in Church's buffer (0.5 M Na-phosphate buffer containing 7% SDS and 100 μg ml⁻¹ of unlabelled salmon sperm carrier DNA). The labelled probe was heat denatured in a boiling water bath for 5 min and snap-cooled

on ice. The probe was added to the hybridization Church's buffer and allowed to hybridize for 48 h at 65 °C. The blot was washed twice in 2× SSC (300 mM NaCl, 30 mM sodium citrate, pH 7.0), 0.05% SDS for 10 min at room temperature, twice in 2× SSC, 0.05% SDS for 5 min at 60 °C, twice in 0.5× SSC, 0.05% SDS for 5 min at 60 °C, and twice in 0.25× SSC, 0.05% SDS for 5 min at 60 °C. The blot was exposed to X-ray film for 16 h at −80 °C. Uncropped, unprocessed images of all gels and blots are shown in Supplementary Fig. 9.

## FISH and immunofluorescence

To validate the organization of the chromosome 8 centromere, we performed FISH on metaphase chromosome spreads as previously described[70] with slight modifications. In brief, CHM13 cells were treated with colcemid and resuspended in HCM buffer (10 mM HEPES pH7.3, 30 mM glycerol, 1 mM CaCl$_2$, 0.8 mM MgCl$_2$). After 10 min, cells were fixed with methanol:acetic acid (3:1), dropped onto previously clean slides, and soaked in 1× PBS. Slides were incubated overnight in cold methanol, hybridized with labelled FISH probes at 68 °C for 2 min, and incubated overnight at 37 °C. Slides were washed three times in 0.1× SSC at 65 °C for 5 min each before mounting in Vectashield containing 5 μg ml$^{-1}$ DAPI. Slides were imaged on a fluorescence microscope (Leica DM RXA2) equipped with a charge-coupled device camera (CoolSNAP HQ2) and a 100× 1.6–0.6 NA objective lens. Images were collected using Leica Application Suite X (v.3.7).

The probes used to validate the organization of the chromosome 8 centromere were picked from the human large-insert clone fosmid library ABC10. ABC10 end sequences were mapped using MEGABLAST (similarity = 0.99, parameters: -D 2 -v 7 -b 7 -e 1e-40 -p 80 -s 90 -W 12 -t 21 -F F) to a repeat-masked CHM13 genome assembly containing the complete chromosome 8 (parameters: -e wublast -xsmall -no_is -s -species Homo sapiens). Expected insert size for fosmids was set to (min) 32 kb and (max) 48 kb. Resulting clone alignments were grouped into the following categories based on uniqueness of the alignment for a given pair of clones, alignment orientation and the inferred insert size from the assembly. (1) Concordant best: unique alignment for clone pair, insert size within expected fosmid range, expected orientation. (2) Concordant tied: non-unique alignment for clone pair, insert size within expected fosmid range, expected orientation. (3) Discordant best: unique alignment of clone pair, insert size too small, too large or in opposite expected orientation of expected fosmid clone. (4) Discordant tied: non unique alignment for clone pair, insert size too small, too large or in opposite expected orientation of expected fosmid clone. (5) Discordant trans: clone pair has ends mapping to different contigs.

Clones aligning to regions within the chromosome 8 centromeric region were selected for FISH validation. The fosmid clones used for validation of the chromosome 8 centromeric region are: 174552_ABC10_2_1_000046302400_C7 for the p-arm monomeric α-satellite region (Cy5; blue), 174222_ABC10_2_1_000044375100_H13 for the p-arm portion of the D8Z2 HOR array (FluorX; green), 171417_ABC10_2_1_000045531400_M19 for the central portion of the D8Z2 HOR array (Cy3; red), 173650_ABC10_2_1_000044508400_J14 for the q-arm portion of the D8Z2 HOR array (FluorX; green), and 173650_ABC10_2_1_000044091500_K11 for the q-arm monomeric α-satellite region (Cy5; blue).

To determine the location of CENP-A relative to methylated DNA (specifically, 5-methylcytosines), we performed immunofluorescence on stretched CHM13 chromatin fibres as previously described[71,72] with modifications. In brief, CHM13 cells were swollen in a hypotonic buffer consisting of a 1:1:1 ratio of 75 mM KCl, 0.8% sodium citrate, and dH$_2$O for 5 min. Then, $3.5 \times 10^4$ cells were cytospun onto an ethanol-washed glass slide with a Shandon Cytospin 4 at 55g for 4 min with high acceleration and allowed to adhere for 1 min before immersing in a salt-detergent-urea lysis buffer (25 mM Tris pH 7.5, 0.5 M NaCl, 1% Triton X-100 and 0.3 M urea) for 15 min at room temperature. The slide was slowly removed from the lysis buffer over a time period of 38 s and

subsequently washed in PBS, incubated in 4% formaldehyde in PBS for 10 min, and washed with PBS and 0.1% Triton X-100. The slide was rinsed in PBS and 0.05% Tween-20 (PBST) for 3 min, blocked for 30 min with immunofluorescence block (2% FBS, 2% BSA, 0.1% Tween-20 and 0.02% NaN$_2$), and then incubated with a mouse monoclonal anti-CENP-A antibody (1:200, Enzo, ADI-KAM-CC006-E) and rabbit monoclonal anti-5-methylcytosine antibody (1:200, RevMAb, RM231) for 3 h at room temperature. Cells were washed three times for 5 min each in PBST and then incubated with Alexa Fluor 488 goat anti-rabbit (1:200, Thermo Fisher Scientific, A-11034) and Alexa Fluor 594 conjugated to goat anti-mouse (1:200, Thermo Fisher Scientific, A-11005) for 1.5 h. Cells were washed three times for 5 min each in PBST, fixed for 10 min in 4% formaldehyde, and washed three times for 1 min each in dH$_2$O before mounting in Vectashield containing 5 μg ml$^{-1}$ DAPI. Slides were imaged on an inverted fluorescence microscope (Leica DMI6000) equipped with a charge-coupled device camera (Leica DFC365 FX) and a 40× 1.4 NA objective lens.

To assess the repeat organization of the 8q21 neocentromere, we performed FISH[73] on CHM13 chromatin fibres. DNA fibres were obtained following Henry H. Q. Heng's protocol with minor modifications[74]. In brief, chromosomes were fixed with methanol:acetic acid (3:1), dropped onto previously clean slides, and soaked in 1× PBS. Manual elongation was performed by coverslip in NaOH:ethanol (5:2) solution. Slides were mounted in Vectashield containing 5 μg ml$^{-1}$ DAPI and imaged on a fluorescence microscope (Leica DM RXA2) equipped with a charge-coupled device camera (CoolSNAP HQ2) and a 100× 1.6–0.6 NA objective lens. The probes used for validation of the 8q21.2 locus were picked from the same ABC10 fosmid library described above and include 174552_ABC10_2_1_000044787700_O7 for Probe 1 (Cy3; red) and 173650_ABC10_2_1_000044086000_F24 for Probe 2 (FluorX; green). Several CHM13 8q21.2 chromatin fibres were imaged. We quantified the number and intensity of the probe signals on a set of CHM13 chromatin fibres using ImageJ's Gel Analysis tool (v.1.51) and found that there were 63 ± 7.55 green signals and 67 ± 5.20 red signals ($n = 3$ independent experiments), consistent with the 67 full and 7 partial repeats in the CHM13 8q21.2 VNTR.

## Native CENP-A ChIP–seq and analysis

We performed two independent replicates of native CENP-A ChIP–seq on CHM13 cells as previously described[25,72] with some modifications. In brief, $3 \times 10^7 – 4 \times 10^7$ cells were collected and resuspended in 2 ml of ice-cold buffer I (0.32 M sucrose, 15 mM Tris, pH 7.5, 15 mM NaCl, 5 mM MgCl$_2$, 0.1 mM EGTA, and 2× Halt Protease Inhibitor Cocktail (Thermo Fisher 78429)). Ice-cold buffer II (2 ml; 0.32 M sucrose, 15 mM Tris, pH 7.5, 15 mM NaCl, 5 mM MgCl$_2$, 0.1 mM EGTA, 0.1% IGEPAL, and 2× Halt Protease Inhibitor Cocktail) was added, and samples were placed on ice for 10 min. The resulting 4 ml of nuclei were gently layered on top of 8 ml of ice-cold buffer III (1.2 M sucrose, 60 mM KCl, 15 mM, Tris pH 7.5, 15 mM NaCl, 5 mM MgCl$_2$, 0.1 mM EGTA, and 2× Halt Protease Inhibitor Cocktail (Thermo Fisher 78429)) and centrifuged at 10,000g for 20 min at 4 °C. Pelleted nuclei were resuspended in buffer A (0.34 M sucrose, 15 mM HEPES, pH 7.4, 15 mM NaCl, 60 mM KCl, 4 mM MgCl$_2$ and 2× Halt Protease Inhibitor Cocktail) to 400 ng ml$^{-1}$. Nuclei were frozen on dry ice and stored at 80 °C. MNase digestion reactions were carried out on 200–300 μg chromatin, using 0.2–0.3 U μg$^{-1}$ MNase (Thermo Fisher 88216) in buffer A supplemented with 3 mM CaCl$_2$ for 10 min at 37 °C. The reaction was quenched with 10 mM EGTA on ice and centrifuged at 500g for 7 min at 4 °C. The chromatin was resuspended in 10 mM EDTA and rotated at 4 °C for 2 h. The mixture was adjusted to 500 mM NaCl, rotated for another 45 min at 4 °C and then centrifuged at maximum speed (21,100g) for 5 min at 4 °C, yielding digested chromatin in the supernatant. Chromatin was diluted to 100 ng ml$^{-1}$ with buffer B (20 mM Tris, pH 8.0, 5 mM EDTA, 500 mM NaCl and 0.2% Tween 20) and pre-cleared with 100 μl 50% protein G Sepharose bead (GE Healthcare) slurry for 20 min at 4 °C, rotating. Precleared supernatant (10–20 μg bulk nucleosomes) was saved for further processing. To the remaining

supernatant, 20 μg mouse monoclonal anti-CENP-A antibody (Enzo ADI-KAM-CC006-E) was added and rotated overnight at 4 °C. Immunocomplexes were recovered by the addition of 200 ml 50% protein G Sepharose bead slurry followed by rotation at 4 °C for 3 h. The beads were washed three times with buffer B and once with buffer B without Tween. For the input fraction, an equal volume of input recovery buffer (0.6 M NaCl, 20 mM EDTA, 20 mM Tris, pH 7.5 and 1% SDS) and 1 ml of RNase A (10 mg ml$^{-1}$) was added, followed by incubation for 1 h at 37 °C. Proteinase K (100 mg ml$^{-1}$, Roche) was then added, and samples were incubated for another 3 h at 37 °C. For the ChIP fraction, 300 μl of ChIP recovery buffer (20 mM Tris, pH 7.5, 20 mM EDTA, 0.5% SDS and 500 mg ml$^{-1}$ proteinase K) was added directly to the beads and incubated for 3–4 h at 56 °C. The resulting proteinase K-treated samples were subjected to a phenol–chloroform extraction followed by purification with a QIAGEN MinElute PCR purification column. Unamplified bulk nucleosomal and ChIP DNA were analysed using an Agilent Bioanalyzer instrument and a 2100 High Sensitivity Kit.

Sequencing libraries were generated using the TruSeq ChIP Library Preparation Kit Set A (Illumina IP-202-1012) according to the manufacturer's instructions, with some modifications. In brief, 5–10 ng bulk nucleosomal or ChIP DNA was end-repaired and A-tailed. Illumina TruSeq adaptors were ligated, libraries were size-selected to exclude polynucleosomes using an E-Gel SizeSelect II agarose gel, and the libraries were PCR-amplified using the PCR polymerase and primer cocktail provided in the kit. The resulting libraries were submitted for 150 bp, paired-end Illumina sequencing using a NextSeq 500/550 High Output Kit v2.5 (300 cycles). The resulting reads were assessed for quality using FastQC (https://github.com/s-andrews/FastQC), trimmed with Sickle (https://github.com/najoshi/sickle; v1.33) to remove low-quality 5′ and 3′ end bases, and trimmed with Cutadapt[75] (v1.18) to remove adapters.

Processed CENP-A ChIP and bulk nucleosomal reads were aligned to the CHM13 whole-genome assembly[5] using two different approaches: (1) BWA-MEM[76] (v.0.7.17) and (2) a $k$-mer-based mapping approach we developed (described below).

For BWA-MEM mapping, data were aligned with the following parameters: bwa mem -k 50 -c 1000000 {index} {read1.fastq.gz} for single-end data, and bwa mem -k 50 -c 1000000 {index} {read1.fastq.gz} {read2.fastq.gz} for paired-end data. The resulting SAM files were filtered using SAMtools[47] with FLAG score 2308 to prevent multi-mapping of reads. With this filter, reads mapping to more than one location are randomly assigned a single mapping location, thereby preventing mapping biases in highly identical regions. Alignments to the chromosome 8 centromere were downsampled to the same coverage and normalized with deepTools[77] (v.3.4.3) bamCompare with the following parameters: bamCompare -b1 {ChIP.bam} -b2 {Bulk_nucleosomal.bam} --operation ratio --binSize 1000 -o {out.bw}. The resulting bigWig file was visualized on the UCSC Genome Browser using the CHM13 chromosome 8 assembly as an assembly hub.

For the $k$-mer-based mapping, the initial BWA-MEM alignment was used to identify reads specific to the chromosome 8 centromeric region (chr8:43600000–47200000). The $k$-mers ($k = 50$) were identified from each chromosome 8 centromere-specific data set using Jellyfish (v.2.3.0) and mapped back onto reads and chromosome 8 centromere assembly allowing for no mismatches. Approximately 93–98% of all $k$-mers identified in the reads were also found within the D8Z2 HOR array. Each $k$-mer from the read data was then placed once at random between all sites in the HOR array that had a perfect match to that $k$-mer. These data were then visualized using a histogram with 1-kb bins in R (R core team, 2020).

## Mappability of short reads within the chromosome 8 centromeric region

To determine the mappability of short reads within the chromosome 8 centromeric HOR array, we performed a simulation where we generated 300,000 random 150-bp fragments from five equally sized (416 kb) regions across the CHM13 D8Z2 HOR array. We mapped these fragments back to the CHM13 chromosome 8 centromeric region using BWA-MEM (v0.7.17) or the $k$-mer-based approach, as described above. For BWA-MEM mapping, the 150-bp fragments were aligned with the following parameters: bwa mem -k 50 -c 1000000 {index} {fragments.fasta}. The resulting SAM files were filtered using SAMtools[47] with FLAG score 2308 to prevent multi-mapping of reads and then converted to a BAM file. BAM files were visualized in IGV[54]. For the $k$-mer-based mapping, $k$-mers ($k = 50$) were identified from each set of 150-bp fragments using Jellyfish (v.2.3.0) and mapped back onto the fragments and the chromosome 8 centromere assembly allowing for no mismatches. $k$-mers with perfect matches to multiple sites within the centromeric region were assigned to one of the sites at random. These data were visualized using a histogram with 1-kb bins in R (R core team, 2020).

## Phylogenetic analysis

To assess the phylogenetic relationship between α-satellite repeats, we first masked every non-α-satellite repeat in the human and NHP centromere assemblies using RepeatMasker[78] (v.4.1.0). Then, we subjected the masked assemblies to StringDecomposer[42] (version available 28 February 2020) using a set of 11 α-satellite monomers derived from a chromosome 8 11-monomer HOR unit (described in the 'Analysis of α-satellite organization' section above). This tool identifies the location of α-satellite monomers in the assemblies, and we used this to extract the α-satellite monomers from the HOR/dimeric array and monomeric regions into multi-FASTA files. We ultimately extracted 12,989, 8,132, 12,224, 25,334 and 63,527 α-satellite monomers from the HOR/dimeric array in human, chimpanzee (H1), chimpanzee (H2), orangutan and macaque, respectively, and 2,879, 3,781, 3,351, 1,573 and 8,127 monomers from the monomeric regions in human, chimpanzee (H1), chimpanzee (H2), orangutan and macaque, respectively. We randomly selected 100 and 50 α-satellite monomers from the HOR/dimeric array and monomeric regions and aligned them with MAFFT[79,80] (v.7.453). We used IQ-TREE[81] to reconstruct the maximum-likelihood phylogeny with model selection and 1000 bootstraps. The resulting tree file was visualized in iTOL[82].

To estimate sequence divergence along the pericentromeric regions, we first mapped each NHP centromere assembly to the CHM13 centromere assembly using minimap2[44] (v.2.17-r941) with the following parameters: -ax asm20 --eqx -Y -t 8 -r 500000. Then, we generated a BED file of 10 kb windows located within the CHM13 centromere assembly. We used the BED file to subset the BAM file, which was subsequently converted into a set of FASTA files. FASTA files contained at least 5 kb of orthologous sequences from one or more NHP centromere assemblies. Pairs of human and NHP orthologous sequences were realigned using MAFFT (v.7.453) and the following command: mafft --maxiterate 1000 --localpair. Sequence divergence was estimated using the Tamura-Nei substitution model[83], which accounts for recurrent mutations and differences between transversions and transitions as well as within transitions. Mutation rate per segment was estimated using Kimura's model of neutral evolution[84]. In brief, we modelled the estimated divergence ($D$) is a result of between-species substitutions and within-species polymorphisms; that is, $D = 2\mu t + 4Ne\mu$, in which Ne is the ancestral human effective population size, $t$ is the divergence time for a given human–NHP pair, and $\mu$ is the mutation rate. We assumed a generation time of [20, 29] years and the following divergence times: human–macaque = [$23 \times 10^6$, $25 \times 10^6$] years, human–orangutan = [$12 \times 10^6$, $14 \times 10^6$] years, human–chimpanzee = [$4 \times 10^6$, $6 \times 10^6$] years. To convert the genetic unit to a physical unit, our computation also assumes Ne = 10,000 and uniformly drawn values for the generation and divergence times.

## Reporting summary

Further information on research design is available in the Nature Research Reporting Summary linked to this paper.

## Data availability

The complete CHM13 chromosome 8 sequence and all data generated and/or used in this study are publicly available and listed in Supplementary Table 9 with their BioProject, accession numbers and/or URL. For convenience, we also list their BioProjects and/or URLs here: complete CHM13 chromosome 8 sequence (PRJNA686384); CHM13 ONT, Iso-Seq, and CENP-A ChIP-seq data (PRJNA559484); CHM13 Strand-Seq alignments (https://zenodo.org/record/3998125); HG00733 ONT data (PRJNA686388); HG00733 PacBio HiFi data (PRJEB36100); testis and fetal brain Iso-Seq data (PRJNA659539); and NHPs (chimpanzee (Clint; S006007), orangutan (Susie; PR01109), and macaque (AG07107)) ONT and PacBio HiFi data (PRJNA659034). All CHM13 BACs used in this study are listed in Supplementary Table 10 with their accession numbers.

## Code availability

Custom code for the SUNK-based assembly method is available at https://github.com/glogsdon1/sunk-based_assembly. All other code is publicly available.

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

**Acknowledgements** We thank S. Goodwin for sequence data generation; M. Jain and D. Miller for re-base-calling sequence data; R. Tindell, H. Visse, A. Tornabene, and G. Ellis for technical assistance; Z. Zhao for computational assistance; F. F. Dastvan for instrumentation; D. Gordon for accessioning BACs; G. Bouffard for accessioning ONT FAST5 data; J. G. Underwood for discussions; and T. Brown for assistance in editing this manuscript. We acknowledge experimental support from the W. M. Keck Microscopy Center (UW) and the computational resources of the NIH HPC Biowulf cluster (https://hpc.nih.gov). This research was supported, in part, by funding from the National Institutes of Health (NIH), HG002385 and HG010169 (E.E.E.); National Institute of General Medical Sciences (NIGMS), F32 GM134558 (G.A.L.); Intramural Research Program of the National Human Genome Research Institute at NIH (S.K., A.M.P., A.R.); National Library of Medicine Big Data Training Grant for Genomics and Neuroscience 5T32LM012419-04 (M.R.V.); NIH/NHGRI Pathway to Independence Award K99 HG011041 (P.H.); NIH/NHGRI R21 1R21HG010548-01 and NIH/NHGRI U01 1U01HG010971 (K.H.M.); and the Intramural Research Program of the NIH, National Cancer Institute, Center for Cancer Research, USA (V.L.). E.E.E. is an investigator of the Howard Hughes Medical Institute.

**Author contributions** G.A.L. and E.E.E. conceived the project; G.A.L., K.H., K.M.M., A.M.L., C.B. and M.S. generated long-read sequencing data; G.A.L., M.R.V., P.H., Y.M., S.K., S.N., P.C.D., A.R., T.D., D.P., W.T.H., A.M., A.V.B., M.K., T.A.G.-L., C.J., S.C.M., K.H.M. and A.M.P. analysed sequencing data, created genome assemblies, and performed quality control analyses; G.A.L., M.R.V., S.K., A.M.P. and S.N. finalized the chromosome 8 assembly; G.A.L., S.K., S.N., A.M., A.V.B. and K.H.M. assessed the assembly of the centromere; M.A.L. generated pulsed-field gel Southern blots; G.A.L., L.M. and M.V. generated microscopy data; L.G.D. generated and analysed droplet digital PCR data; U.S. provided the CHM13 cell line; J.L.G. and V.L. supervised experimental analyses; G.A.L., M.R.V., D.P. and E.E.E. developed the figures; and G.A.L. and E.E.E. drafted the manuscript.

**Competing interests** The authors declare no competing interests.

**Additional information**
**Correspondence and requests for materials** should be addressed to E.E.E.

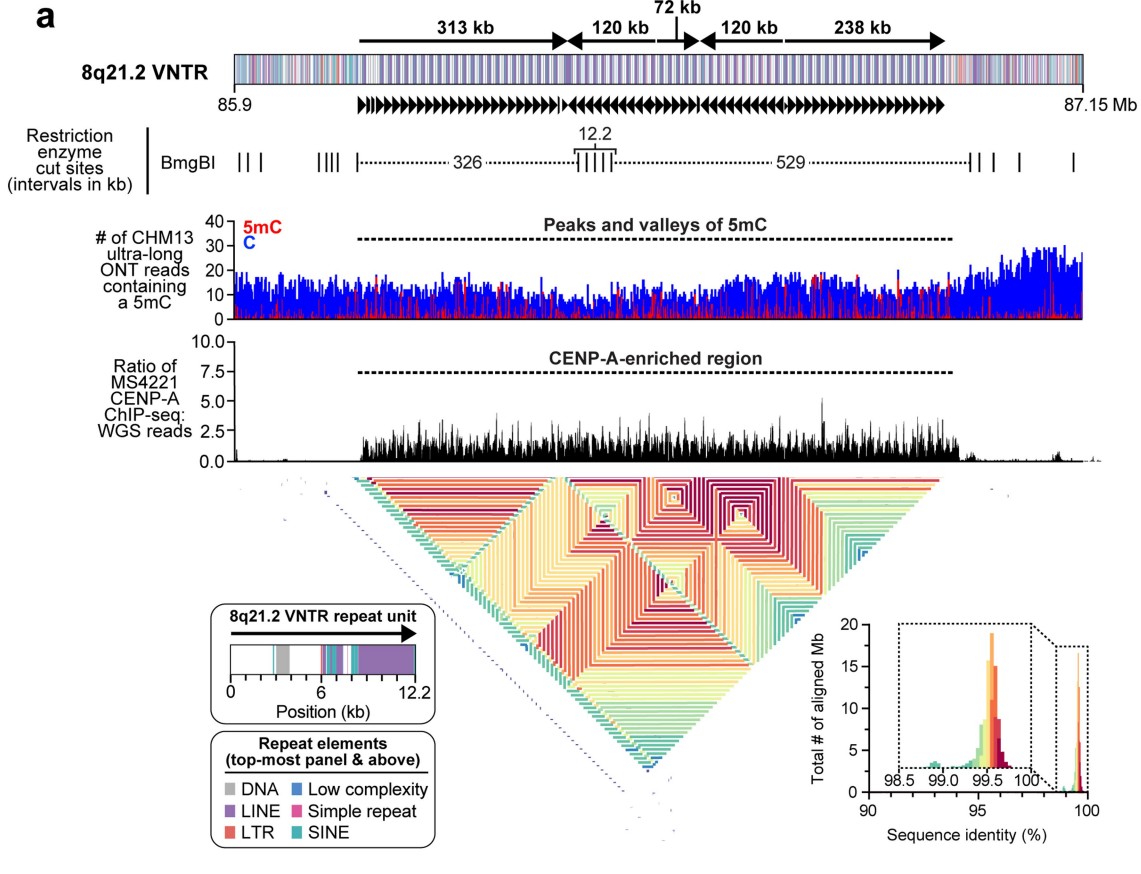

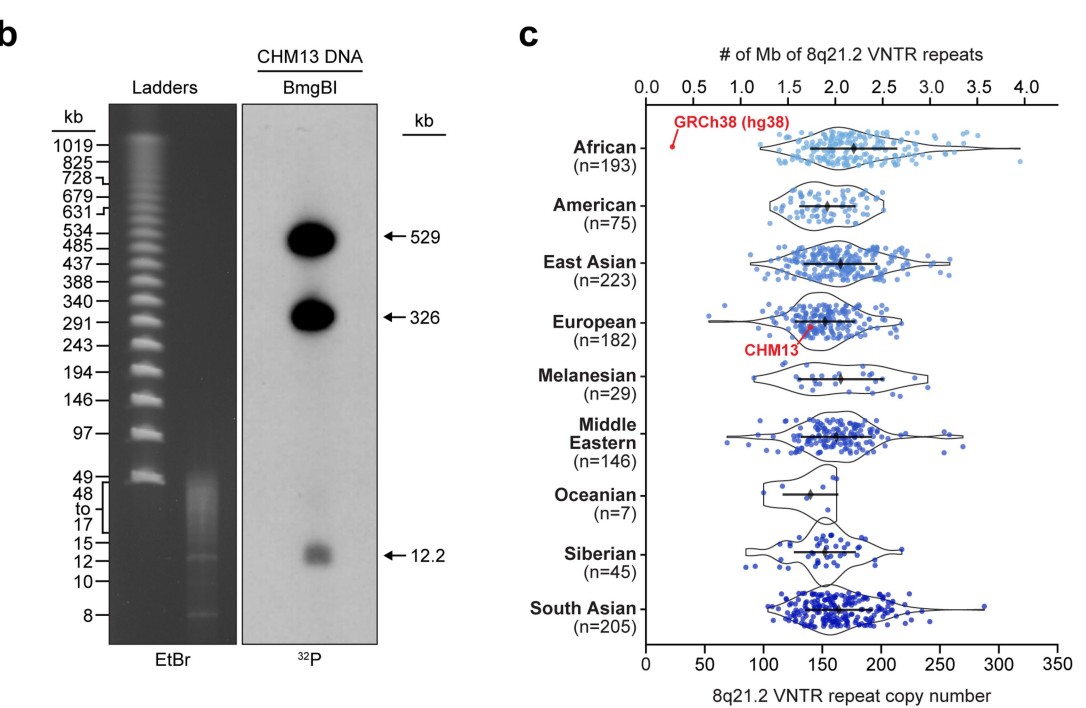

**Extended Data Fig. 1** | See next page for caption.

**Extended Data Fig. 1 | Sequence, structure and epigenetic map of the neocentromeric chromosome 8q21.2 VNTR. a**, Schematic showing the composition of the CHM13 8q21.2 VNTR. This VNTR consists of 67 full and 7 partial 12.192-kb repeats that span 863 kb in total. The predicted restriction digest pattern is indicated. Each repeat is methylated within a 3-kb region and hypomethylated within the rest of the sequence. Mapping of CENP-A ChIP–seq data from the chromosome 8 neodicentric cell line known as MS4221[24,25] (Methods) reveals that approximately 98% of CENP-A chromatin is located within the hypomethylated portion of the repeat. A pairwise sequence identity heat map across the region indicates a mirrored symmetry within a single layer, consistent with the evolutionarily young status of the tandem repeat. **b**, Pulsed-field gel Southern blot of CHM13 DNA digested with BmgBI confirms the size and organization of the chromosome 8q21.2 VNTR. Left, ethidium bromide staining; right, $^{32}$P-labelled chromosome 8q21.2-specific probe. For gel source data, see Supplementary Fig. 1c, d. **c**, Copy number of the 8q21 repeat (chr8:85792897–85805090 in GRCh38) throughout the human population. CHM13 is estimated to have 144 total copies of the 8q21 repeat, or 72 copies per haplotype, whereas GRCh38 only has 26 copies (red data points). Median ± s.d. is shown.

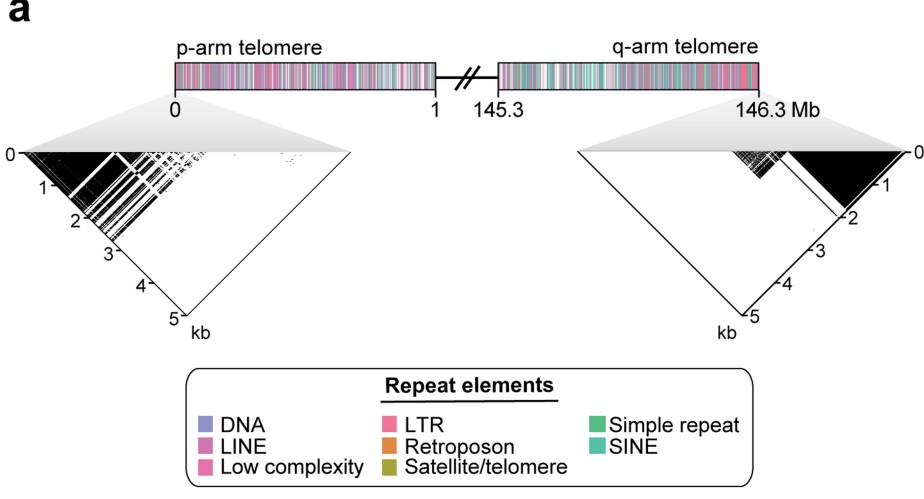

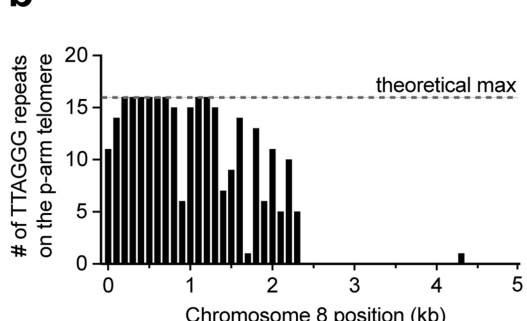

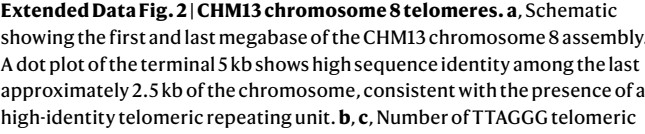

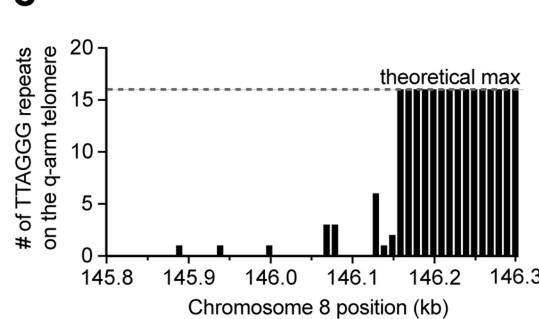

**Extended Data Fig. 2 | CHM13 chromosome 8 telomeres. a**, Schematic showing the first and last megabase of the CHM13 chromosome 8 assembly. A dot plot of the terminal 5 kb shows high sequence identity among the last approximately 2.5 kb of the chromosome, consistent with the presence of a high-identity telomeric repeating unit. **b**, **c**, Number of TTAGGG telomeric repeats in the last 5 kb of the p-arm (**b**) and q-arm (**c**) in chromosome 8. The p-arm has a gradual transition to pure TTAGGG repeats over nearly 1 kb, whereas the q-arm has a very sharp transition to pure TTAGGG repeats that occurs over nearly 300 bp.

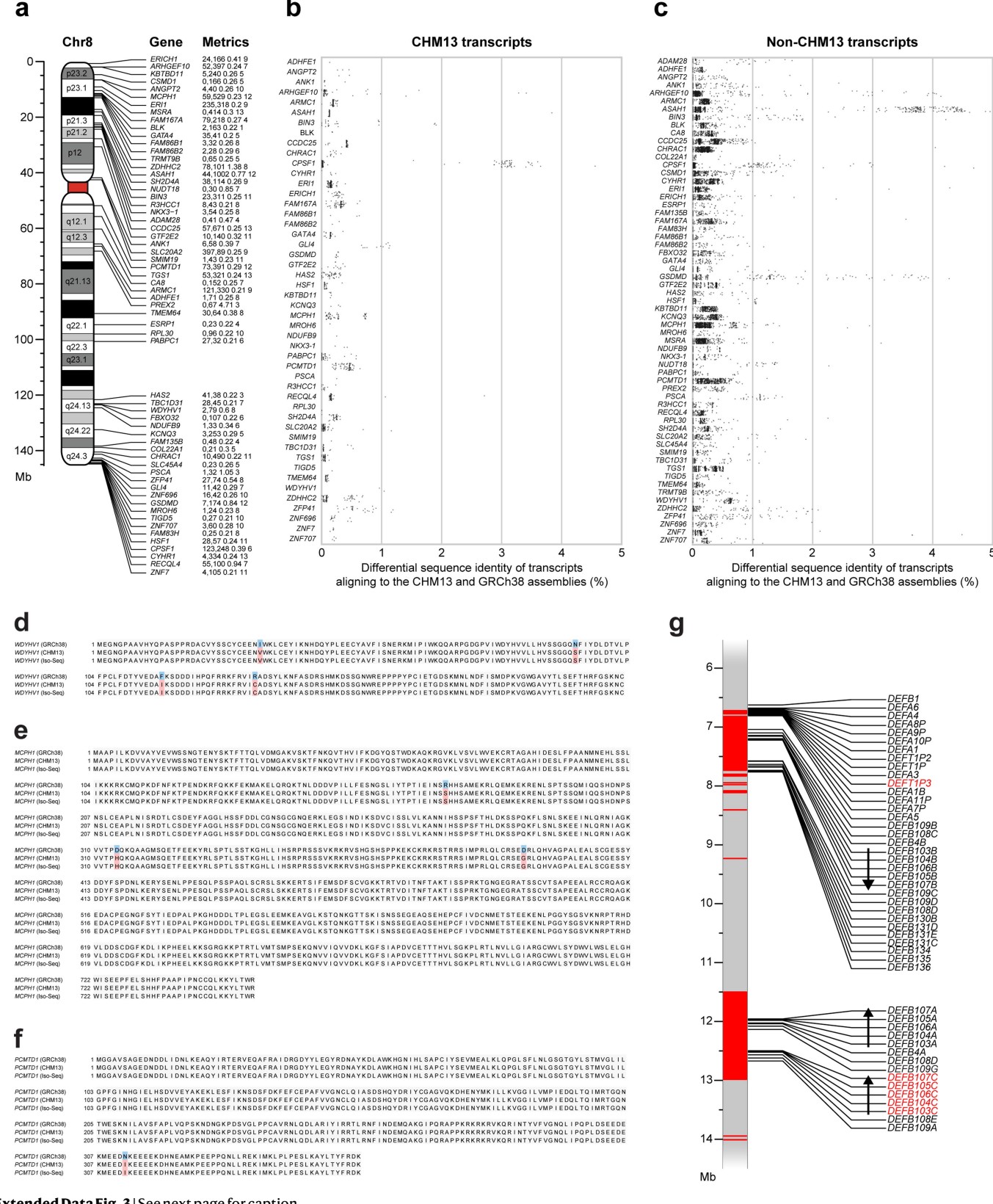

**Extended Data Fig. 3** | See next page for caption.

**Extended Data Fig. 3 | Genes with improved alignment to the CHM13 chromosome 8 assembly relative to GRCh38. a**, Ideogram of chromosome 8 showing protein-coding genes with improved transcript alignments to the CHM13 chromosome 8 assembly relative to GRCh38 (hg38). Each gene is labelled with its name, count of improved transcripts from the CHM13 cell line, count of improved transcripts from other tissues, the average percent improvement of non-CHM13 cell line alignments, and the number of tissue sources with improved transcript mappings. **b**, **c**, Differential percentage sequence identity of transcripts aligning to CHM13 or GRCh38 for CHM13 cell line transcripts (**b**) and non-CHM13 cell line transcripts (**c**). **d**–**f**, Multiple-sequence alignments for *WDYHV1* (**d**), *MCPH1* (**e**) and *PCMTD1* (**f**), all of which have at least 0.1% greater sequence identity of >20 full-length Iso-Seq transcripts to the CHM13 chromosome 8 assembly than to GRCh38 (Methods). For each gene, the GRCh38 annotation is compared to the same annotation lifted over to the CHM13 chromosome 8 assembly, and the substitutions are confirmed by translated predicted open reading frames from Iso-Seq transcripts. Matching amino acids are shaded in grey, those matching only the Iso-Seq data are in red, and those different from the Iso-Seq data are in blue. Each substitution in CHM13 relative to GRCh38 has an allele frequency of 0.36 in gnomAD (v3). **g**, Location of *DEFA* and *DEFB* genes in the CHM13 chromosome 8 β-defensin locus. Segmental duplication regions were identified by SEDEF[85], and new paralogues are shown in red. Duplication cassettes are marked with arrows indicating orientation for each copy.

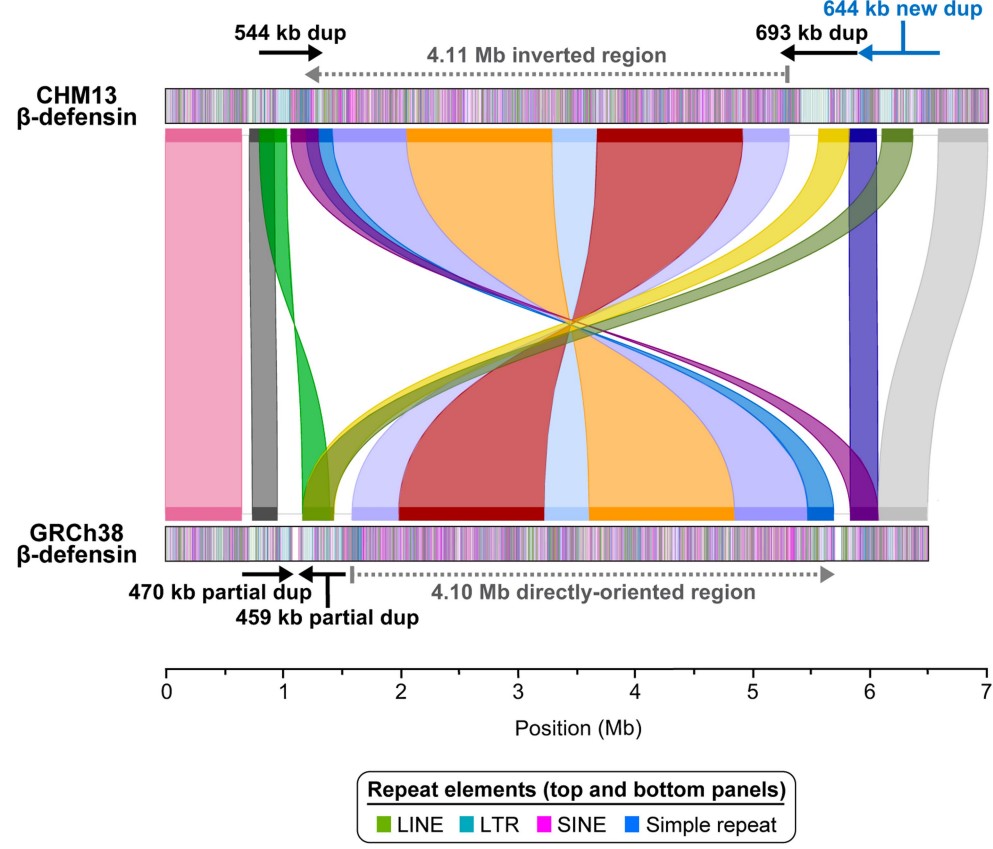

**Extended Data Fig. 4 | Comparison of the CHM13 and GRCh38 β-defensin loci.** Miropeats comparison of the CHM13 and GRCh38 β-defensin loci identifies a 4.11-Mb inverted region (dashed grey line) bracketed by proximal and distal segmental duplications (dup; black and blue arrows) in CHM13. CHM13 also has an additional segmental duplication (blue arrow) relative to the GRCh38. In total, the CHM13 haplotype adds 611.9 kb of new sequence, of which 602.6 kb is located within segmental duplications and 9.3 kb is located at the distal edge of the inverted region. Coloured segments track blocks of homology between CHM13 and GRCh38.

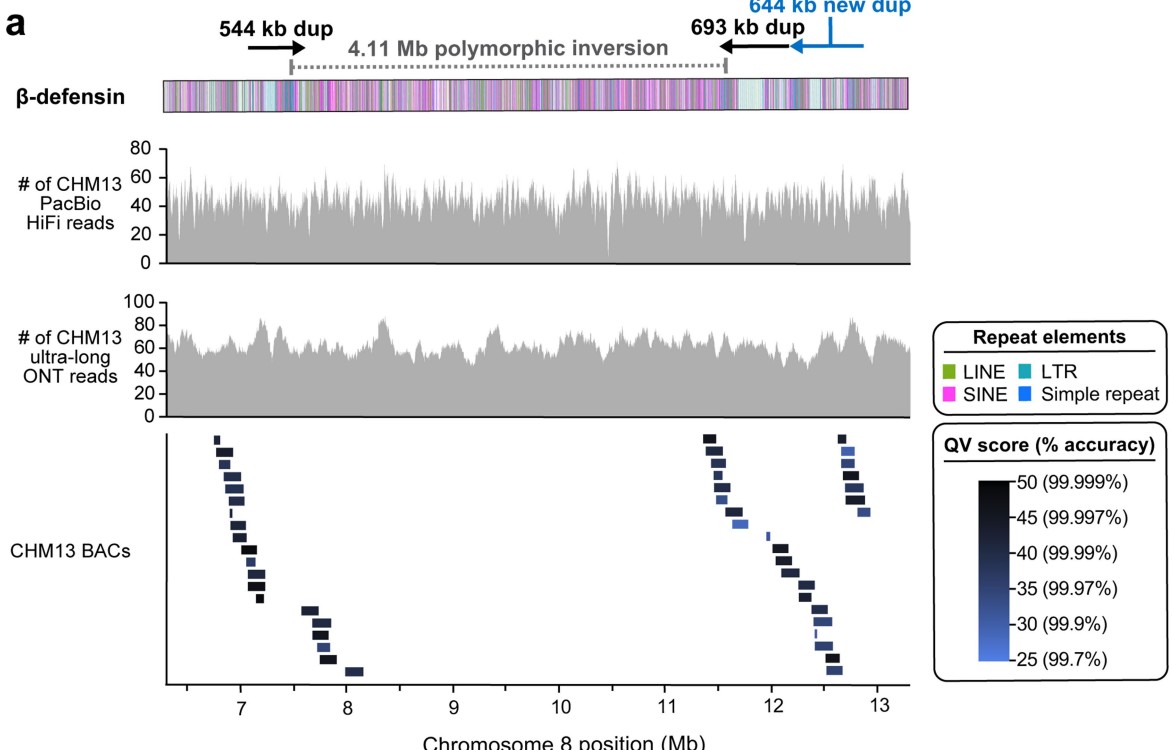

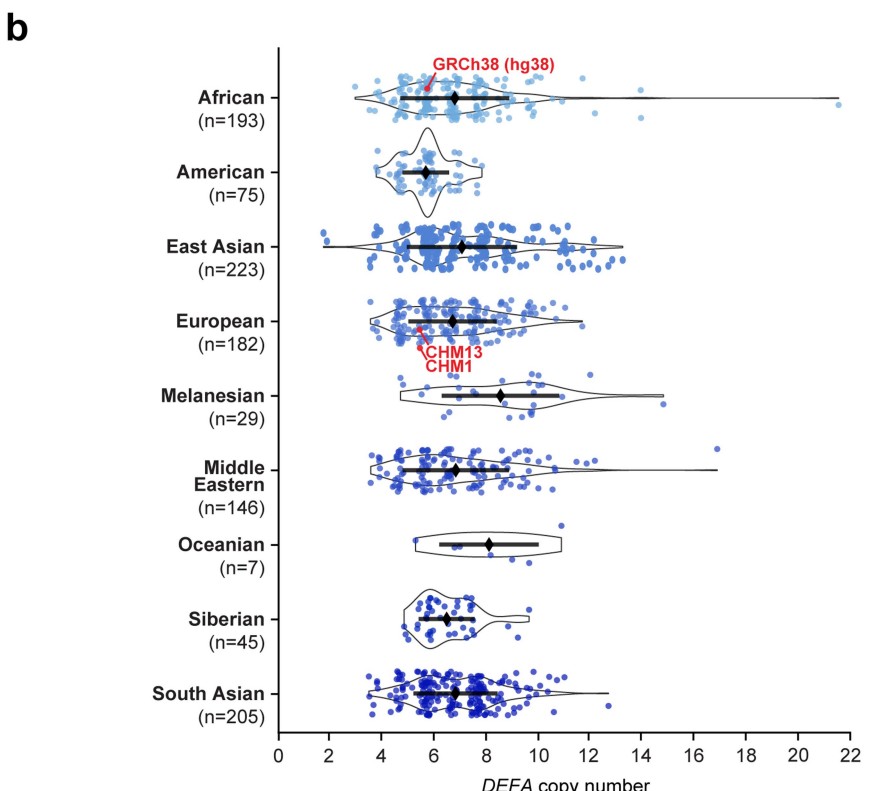

**Extended Data Fig. 5 | Validation of the CHM13 β-defensin locus, and copy number of the *DEFA* gene family. a**, Coverage of CHM13 ONT and PacBio HiFi data along the CHM13 β-defensin locus (top two panels). The ONT and PacBio data have largely uniform coverage, indicating it is free of large structural errors. The dip in HiFi coverage near position 10.46 Mb is due to a G/A bias in HiFi chemistry[5]. The alignment of 47 CHM13 BACs (bottom) reveals that those regions have an estimated quality value score >25 (>99.7% accurate). **b**, Copy number of *DEFA* (chr8:6976264−6995380 in GRCh38 (hg38)) throughout the human population. Median ± s.d. is shown.

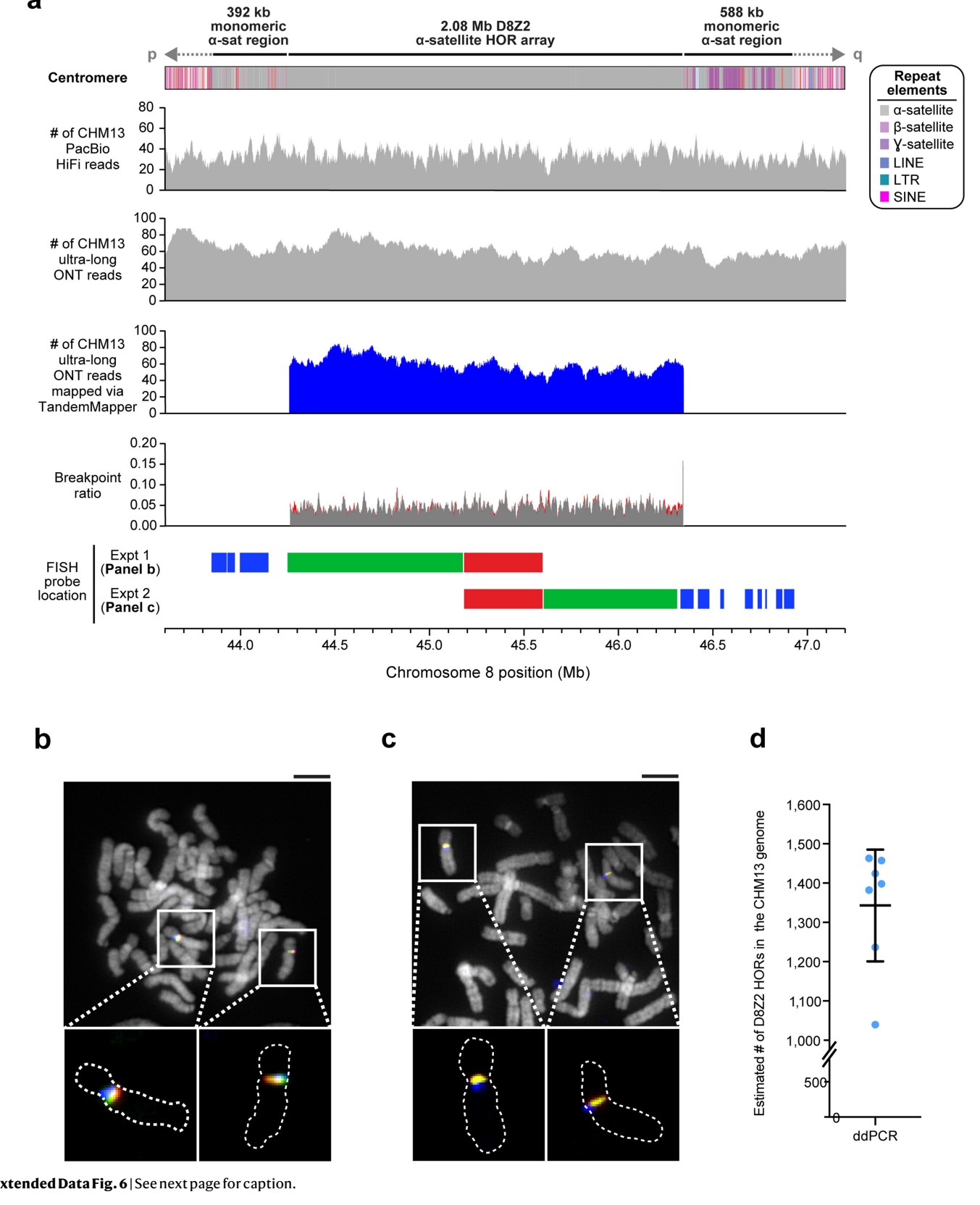

**Extended Data Fig. 6** | See next page for caption.

**Extended Data Fig. 6 | Validation of the CHM13 chromosome 8 centromeric region. a**, Coverage of CHM13 ONT and PacBio HiFi data along the CHM13 chromosome 8 centromeric region (top two panels) is largely uniform, indicating a lack of large structural errors. Analysis with TandemMapper and TandemQUAST[52], which are tools that assess repeat structure via mapped reads (third panel) and misassembly breakpoints (fourth panel; red), indicates that the chromosome 8 D8Z2 α-satellite HOR array lacks large-scale assembly errors. Five different FISH probes targeting regions in the chromosome 8 centromeric region (bottom) are used to confirm the organization of the α-satellite DNA (**b**, **c**). **b**, **c**, Representative images of metaphase chromosome spreads hybridized with FISH probes targeting regions within the chromosome 8 centromere (**a**). Insets show both chromosome 8s with the predicted organization of the centromeric region. **d**, Droplet digital PCR of the chromosome 8 D8Z2 α-satellite array indicates that there are 1,344 ± 142 D8Z2 HORs present on chromosome 8, consistent with the predictions from an in silico restriction digest and StringDecomposer[42] analysis (Methods). Mean ± s.d. is shown. Scale bar, 5 μm. Insets, 2.5× magnification.

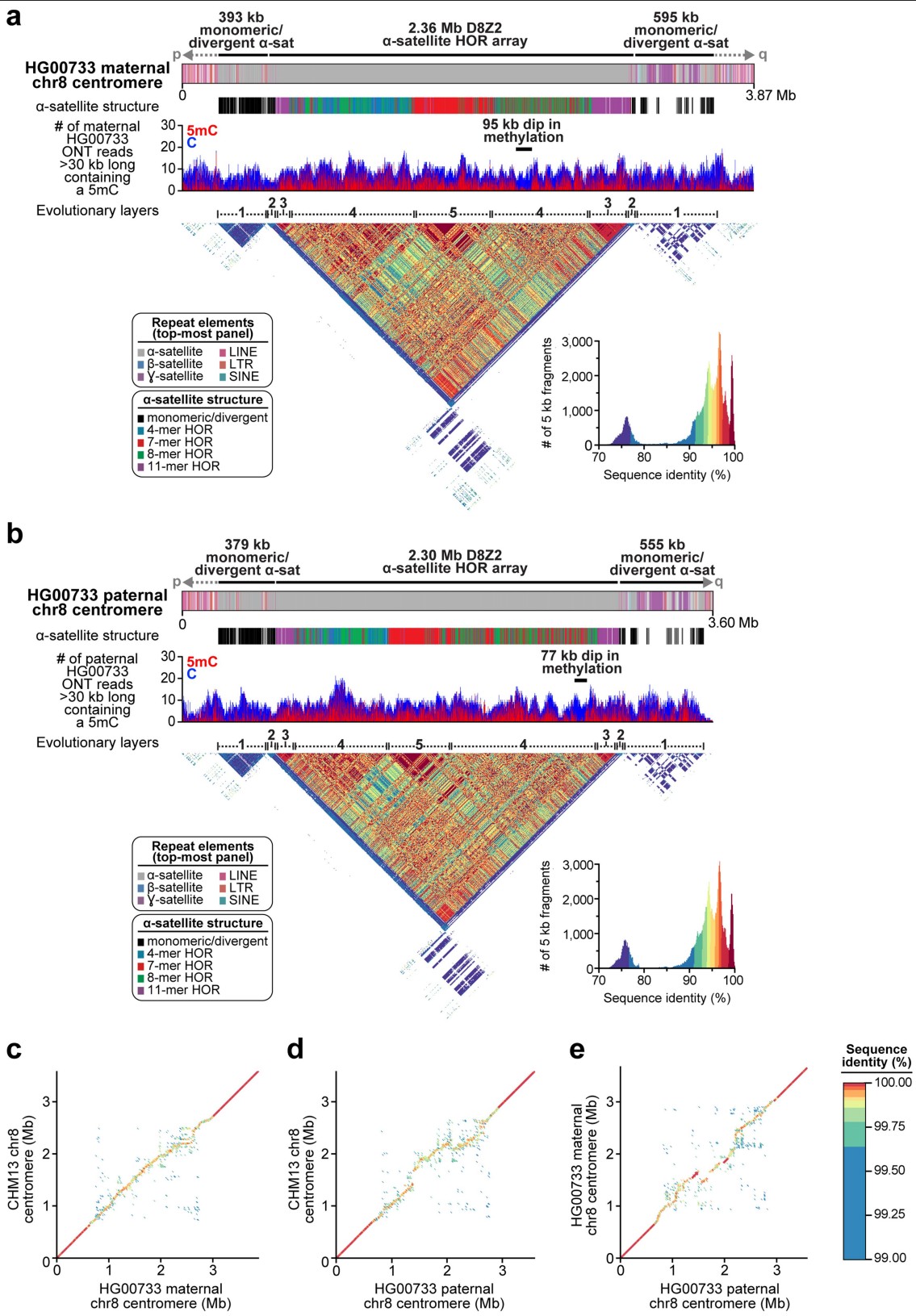

**Extended Data Fig. 7 | Sequence, structure and epigenetic map of human diploid HG00733 chromosome 8 centromeres. a**, **b**, Repeat structure, α-satellite organization, methylation status and sequence identity heat map of the maternal (**a**) and paternal (**b**) chromosome 8 centromeric regions from a diploid human genome (HG00733; Supplementary Table 2) shows structural and epigenetic similarity to the CHM13 chromosome 8 centromeric region (Fig. 2a). **c**–**e**, Dot plot comparisons between the CHM13 and maternal (**c**),

CHM13 and paternal (**d**), and maternal and paternal (**e**) chromosome 8 centromeric regions in the HG00733 genome show more than 99% sequence identity overall, with high concordance in the unique and monomeric α-satellite regions of the centromeres (dark red line) that devolves into lower sequence identity in the α-satellite HOR array, consistent with rapid evolution of this region.

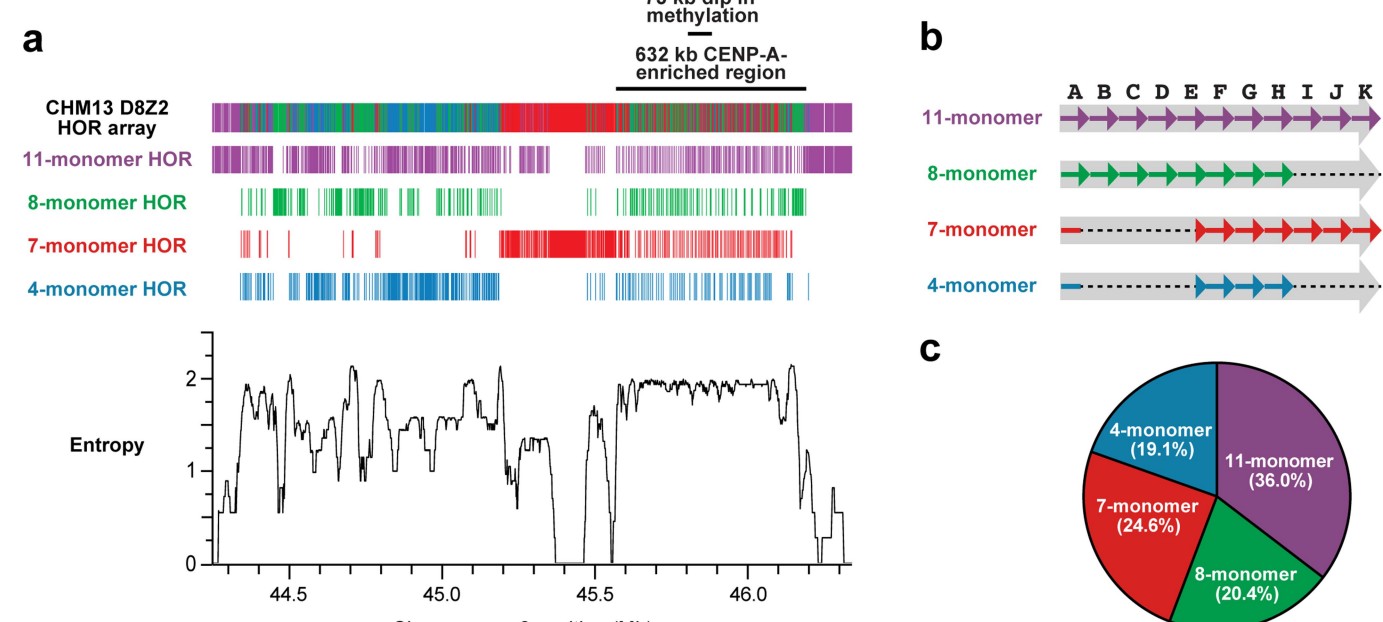

**Extended Data Fig. 8 | Composition, organization and entropy of the CHM13 D8Z2 α-satellite HOR array. a**, HOR composition and organization of the chromosome 8 α-satellite array as determined via StringDecomposer[42]. The predominant HOR subtypes (4-, 7-, 8- and 11-monomer HORs) are shown, whereas those occurring less than 15 times are not (see Methods for absolute quantification). The entropy of the D8Z2 HOR array is plotted in the bottom panel and reveals that the hypomethylated and CENP-A-enriched regions have the highest consistent entropy in the entire array. **b**, Organization of α-satellite monomers within each HOR. The initial monomer of the 4- and 7-monomer HORs is a hybrid of the A and E monomers, with the first 87 bp the A monomer and the subsequent 84 bp the E monomer. **c**, Abundance of the predominant HOR types within the D8Z2 HOR array as determined via StringDecomposer[42].

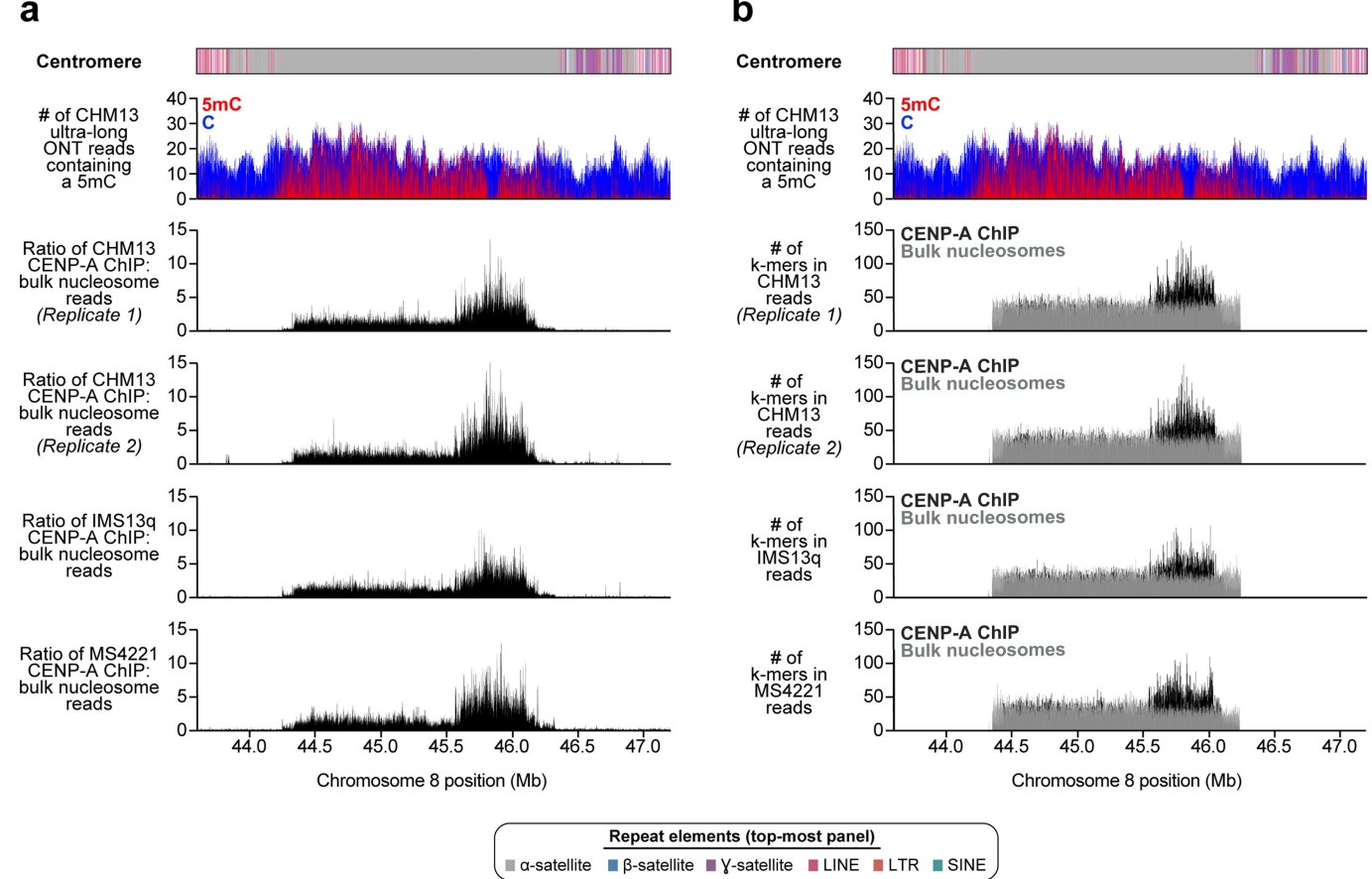

**Extended Data Fig. 9 | Location of CENP-A chromatin within the CHM13 D8Z2 α-satellite HOR array. a**, **b**, Plot of the ratio of CENP-A ChIP to bulk nucleosome reads mapped via BWA-MEM (**a**), or the number of *k*-mer-mapped CENP-A ChIPs (black) or bulk nucleosome (dark grey) reads (**b**) (Methods). Shown are two independent replicates of CENP-A ChIP–seq performed on CHM13 cells (top two panels), as well as single replicates of CENP-A ChIP–seq performed on human diploid neocentromeric cell lines (bottom two panels;

Methods). Although the neocentromeric cell lines have a neocentromere located on either chromosome 13 (IMS13q) or 8 (MS4221)[24,25], they both have at least one karyotypically normal chromosome 8 from which centromeric chromatin can be mapped. We limited our analysis to diploid cell lines rather than aneuploid ones to avoid potentially confounding results stemming from multiple chromosome 8 copies that vary in structure, such as those observed in HeLa cells[86].

## a

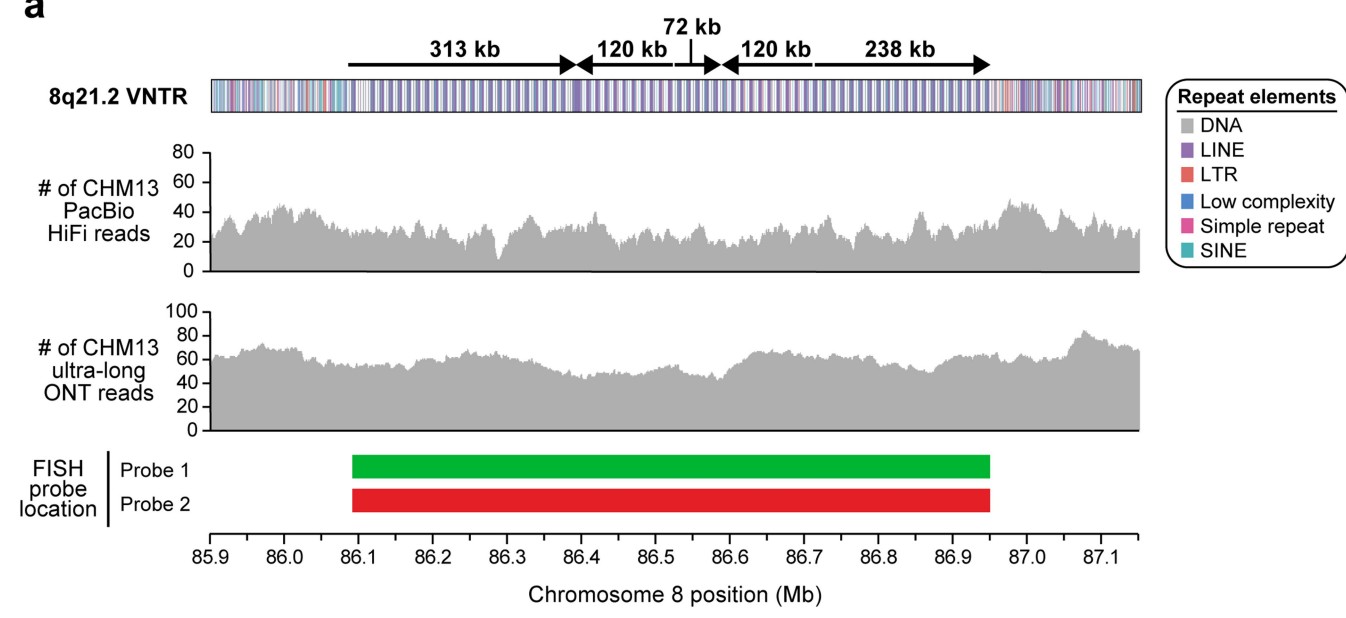

## b

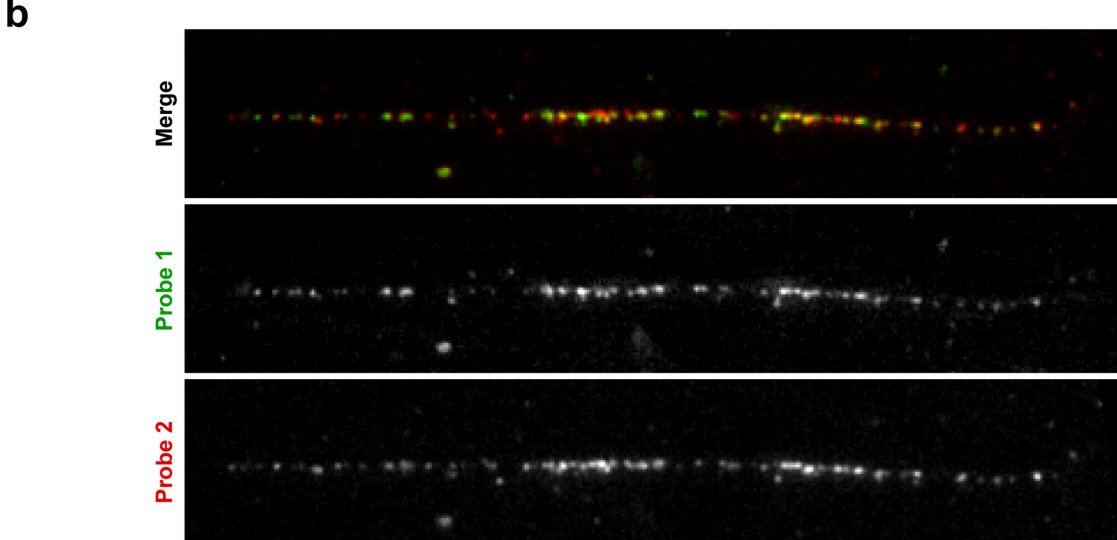

## c

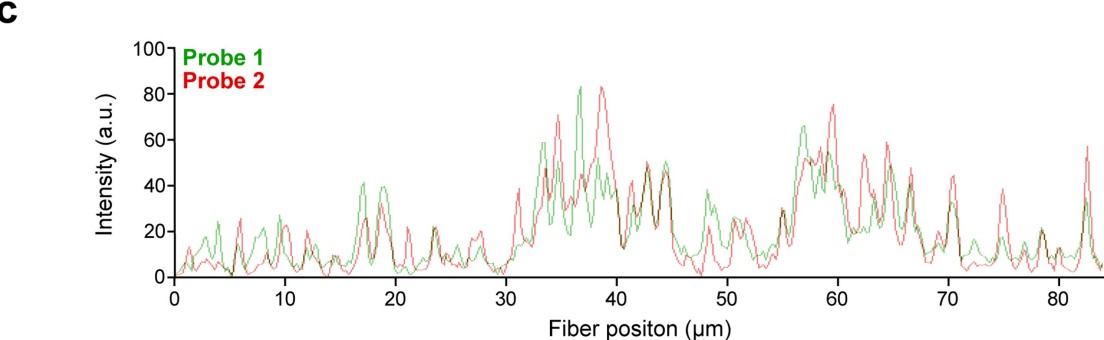

**Extended Data Fig. 10** | See next page for caption.

**Extended Data Fig. 10 | Validation of the CHM13 8q21.2 VNTR. a**, Coverage of CHM13 ONT and PacBio HiFi data along the 8q21.2 VNTR (top two panels) is largely uniform, indicating a lack of large structural errors. Two FISH probes targeting the 12.192-kb repeat in the 8q21.2 VNTR are used to estimate the number of repeats in the CHM13 genome (**b**, **c**). **b**, Representative FISH images of a CHM13 stretched chromatin fibre. Although the FISH probes were designed against the entire VNTR array, stringent washing during FISH produces a punctate probe signal pattern, which may be due to stronger hybridization of the probe to a specific region in the 12.192-kb repeat (perhaps based on GC content or a lack of secondary structures). This punctate pattern can be used to estimate the repeat copy number in the VNTR, thereby serving as a source of validation. **c**, Plot of the signal intensity on the CHM13 chromatin fibre shown in **b**. Quantification of peaks across three independent experiments reveals an average of $63 \pm 7.55$ peaks and $67 \pm 5.20$ peaks (mean ± s.d.) from the green and red probes, respectively, which is consistent with the number of repeat units in the 8q21.2 assembly (67 full and 7 partial repeats). Scale bar, 5 μm.

# Reporting Summary

Nature Research wishes to improve the reproducibility of the work that we publish. This form provides structure for consistency and transparency in reporting. For further information on Nature Research policies, see our Editorial Policies and the Editorial Policy Checklist.

## Statistics

For all statistical analyses, confirm that the following items are present in the figure legend, table legend, main text, or Methods section.

| n/a | Confirmed | |
|---|---|---|
| ☐ | ☒ | The exact sample size ($n$) for each experimental group/condition, given as a discrete number and unit of measurement |
| ☐ | ☒ | A statement on whether measurements were taken from distinct samples or whether the same sample was measured repeatedly |
| ☒ | ☐ | The statistical test(s) used AND whether they are one- or two-sided<br>*Only common tests should be described solely by name; describe more complex techniques in the Methods section.* |
| ☒ | ☐ | A description of all covariates tested |
| ☐ | ☒ | A description of any assumptions or corrections, such as tests of normality and adjustment for multiple comparisons |
| ☐ | ☒ | A full description of the statistical parameters including central tendency (e.g. means) or other basic estimates (e.g. regression coefficient) AND variation (e.g. standard deviation) or associated estimates of uncertainty (e.g. confidence intervals) |
| ☒ | ☐ | For null hypothesis testing, the test statistic (e.g. $F$, $t$, $r$) with confidence intervals, effect sizes, degrees of freedom and $P$ value noted<br>*Give P values as exact values whenever suitable.* |
| ☒ | ☐ | For Bayesian analysis, information on the choice of priors and Markov chain Monte Carlo settings |
| ☒ | ☐ | For hierarchical and complex designs, identification of the appropriate level for tests and full reporting of outcomes |
| ☒ | ☐ | Estimates of effect sizes (e.g. Cohen's $d$, Pearson's $r$), indicating how they were calculated |

*Our web collection on statistics for biologists contains articles on many of the points above.*

## Software and code

Policy information about availability of computer code

| | |
|---|---|
| Data collection | Leica Application Suite X (v3.7), Oxford Nanopore Technologies MinKNOW (v2.0 - v19.12), and Pacific Bioscience Sequel II Instrument Control SW (v7.1 or v8.0). |
| Data analysis | Custom code for the SUNK-based assembly method is available at https://github.com/glogsdon1/sunk-based_assembly. Other software used in this study are publicly available and include Pacific Biosciences CCS algorithm (v3.4.1 or v4.0.0), HiCanu (v2.0), minimap2 (v2.17), Jellyfish (v2.2.4), pbmm2 (v1.1.0), Winnowmap (v1.0), Merqury (v1.1), BWA-MEM (v0.7.17), sambamba (v0.6.8), SAMtools (v1.9), BEDtools (v2.27.1), deepTools (v3.4.3), TandemTools (version available March 20th, 2020), StringDecomposer (version available February 28th, 2020), Nanopolish (v0.12.5), CHESS (v2.2), R (v1.1.383), Solve (v3.4), RepeatMasker (v4.1.0), ImageJ (v1.51), MAFFT (v7.453), mrsFAST (v3.4.1), Sickle (v1.33), and Cutadapt (v1.18). |

For manuscripts utilizing custom algorithms or software that are central to the research but not yet described in published literature, software must be made available to editors and reviewers. We strongly encourage code deposition in a community repository (e.g. GitHub). See the Nature Research guidelines for submitting code & software for further information.

## Data

Policy information about availability of data

All manuscripts must include a data availability statement. This statement should provide the following information, where applicable:
- Accession codes, unique identifiers, or web links for publicly available datasets
- A list of figures that have associated raw data
- A description of any restrictions on data availability

The complete CHM13 chromosome 8 sequence and all data generated and/or used in this study are publicly available and listed in Supplementary Table 9 with their BioProject, accession #, and/or URL. For convenience, we list their BioProjects and/or URLs here: complete CHM13 chromosome 8 sequence (PRJNA559484);

# Field-specific reporting

Please select the one below that is the best fit for your research. If you are not sure, read the appropriate sections before making your selection.

☒ Life sciences ☐ Behavioural & social sciences ☐ Ecological, evolutionary & environmental sciences

For a reference copy of the document with all sections, see nature.com/documents/nr-reporting-summary-flat.pdf

# Life sciences study design

All studies must disclose on these points even when the disclosure is negative.

| | |
|---|---|
| Sample size | We generated a whole-chromosome assembly of human chromosome 8 and assembled the chromosome 8 centromere in a diploid human cell line and three diploid nonhuman primates in order to perform phylogenetic and comparative analyses. For phylogenetic tree reconstruction of the centromeric satellite, we used 150 data points from each genome, which resulted in a bootstrap value of 100 for all major branches of the tree (meaning, 100 out of 100 times, the same branch was observed in that clade when repeating the phylogenetic reconstruction on resampled data). For the centromeric mutation rate computation, we compared 1,002 10 kbp regions from across the chimpanzee, orangutan, and macaque genomes to the corresponding human region, which spans approximately 1.65 Mbp of sequence. This number of data points is the maximum number of points that can possibly be analyzed within this region (assuming 10 kbp windows) and is strengthened by the comparison across three different species (rather than just one). For gene copy number estimation, we analyzed 1,105 published high-coverage datasets spanning nine human superpopulations, which were all that were available for this analysis and provides a sufficiently high number of genomes to determine a median and standard deviation of gene copy number for each superpopulation with confidence. For droplet digital PCR (ddPCR), we performed seven technical replicates, which is four more than the standard three technical replicates used in such experiments. For the chromatin fiber-FISH, we generated three slides, which served as technical replicates, and identified multiple fibers showing the indicated CENP-A and methylation patterns. For the pulsed-field gel Southern blots, each experiment was performed twice with different restriction enzymes, and each result confirmed the expected banding pattern. For FISH on metaphase chromosome spreads, experiments were performed >3 times and generated several spreads with chromosome 8 FISH probes hybridized in the expected order. This number of FISH replicates meets or exceeds the standard number of experimental replication commonly accepted by the field. |
| Data exclusions | No data were excluded. |
| Replication | Computational experiments are deterministic and are, therefore, reproducible. Despite this expected reproducibility, computational experiments were run multiple times with different parameters to improve the experimental analysis. All attempts at replication were successful for both computation and wet-lab experiments. |
| Randomization | Randomization is not applicable to this study because we did not perform any experiments where there are treatment and control groups that would necessitate randomization between the subjects. |
| Blinding | Blinding is not applicable to this study because we did not perform any experiments where there are treatment and control groups that would necessitate blinding. |

# Reporting for specific materials, systems and methods

We require information from authors about some types of materials, experimental systems and methods used in many studies. Here, indicate whether each material, system or method listed is relevant to your study. If you are not sure if a list item applies to your research, read the appropriate section before selecting a response.

## Materials & experimental systems

| n/a | Involved in the study |
|---|---|
| ☐ | ☒ Antibodies |
| ☐ | ☒ Eukaryotic cell lines |
| ☒ | ☐ Palaeontology and archaeology |
| ☒ | ☐ Animals and other organisms |
| ☒ | ☐ Human research participants |
| ☒ | ☐ Clinical data |
| ☒ | ☐ Dual use research of concern |

## Methods

| n/a | Involved in the study |
|---|---|
| ☐ | ☒ ChIP-seq |
| ☒ | ☐ Flow cytometry |
| ☒ | ☐ MRI-based neuroimaging |

## Antibodies

| | |
|---|---|
| Antibodies used | Mouse monoclonal anti-CENP-A antibody (Enzo, ADI-KAM-CC006-E) |

| Antibodies used | Rabbit monoclonal anti-5-methylcytosine antibody (RevMAb, RM231)<br>Alexa Fluor 488 goat anti-rabbit (Thermo Fisher Scientific, A-11034)<br>Alexa Fluor 594 conjugated to goat anti-mouse (Thermo Fisher Scientific, A-11005) |
| --- | --- |
| Validation | The anti-CENP-A antibody was generated against a synthetic peptide consisting of aa3-19 of CENP-A, and mutation of this epitope in human cells prevents antibody binding (Logsdon et. al., JCB, 2015).<br><br>The anti-5-methylcytosine antibody was tested against 50, 5, and 0.5 ng of double stranded 5-hydroxymethylcytosine (5-hmC) DNA, 5-methylcytosine (5-mC) DNA, and unmethylated DNA on a dot blot, and it only detected the 5-mC DNA (see https://www.revmab.com/index.php/product/anti-5-methylcytosine-5-mc-rabbit-monoclonal-antibody-clone-rm231-5-mc/). |

## Eukaryotic cell lines

Policy information about cell lines

| Cell line source(s) | CHM13hTERT (abbr. CHM13) cells were originally isolated from a hydatidiform mole at Magee-Womens Hospital (Pittsburgh, PA) as part of a research study (IRB MWH-20-054). Cryogenically frozen cells from this culture were grown and transformed using human telomerase reverse transcriptase (TERT) to immortalize the cell line. This cell line retains a 46,XX karyotype and complete homozygosity. Human HG00733 lymphoblastoid cells were originally obtained from a female Puerto Rican child, immortalized with the Epstein-Barr Virus (EBV), and stored at the Coriell Institute for Medical Research (Camden, NJ). Chimpanzee (Pan troglodytes; Clint; S006007) fibroblast cells were originally obtained from a male western chimpanzee named Clint (now deceased) at the Yerkes National Primate Research Center (Atlanta, GA) and immortalized with EBV. Orangutan (Pongo abelii; Susie; PR01109) fibroblast cells were originally obtained from a female Sumatran orangutan named Susie (now deceased) at the Gladys Porter Zoo (Brownsville, TX), immortalized with EBV, and stored at the Coriell Institute for Medical Research (Camden, NJ). Macaque (Macaca mulatta; AG07107) fibroblast cells were originally obtained from a female rhesus macaque of Indian origin and stored at the Coriell Institute for Medical Research (Camden, NJ). |
| --- | --- |
| Authentication | The CHM13hTERT cell line was authenticated via STR analysis and karyotyped to show a 46,XX karyotype (Miga et al., Nature, 2020). The other cell lines used in this study have not been authenticated to our knowledge. |
| Mycoplasma contamination | The CHM13hTERT cell line is negative for mycoplasma contamination (Miga et al., Nature, 2020). The other cell lines used in this study have not been assessed for mycoplasma contamination to our knowledge. |
| Commonly misidentified lines<br>(See ICLAC register) | No commonly misidentified cell lines were used in this study. |

## ChIP-seq

### Data deposition

☒ Confirm that both raw and final processed data have been deposited in a public database such as GEO.

☒ Confirm that you have deposited or provided access to graph files (e.g. BED files) for the called peaks.

| Data access links<br>*May remain private before publication.* | https://www.ncbi.nlm.nih.gov/sra/?term=SRR13278681<br>https://www.ncbi.nlm.nih.gov/sra/?term=SRR13278682<br>https://www.ncbi.nlm.nih.gov/sra/?term=SRR13278683<br>https://www.ncbi.nlm.nih.gov/sra/?term=SRR13278684 |
| --- | --- |
| Files in database submission | CHM13_CA_ChIP_1_S3_R1_001.fastq.gz<br>CHM13_CA_ChIP_1_S3_R2_001.fastq.gz<br>CHM13_CA_ChIP_2_S4_R1_001.fastq.gz<br>CHM13_CA_ChIP_2_S4_R2_001.fastq.gz<br>CHM13_Input_1_S1_R1_001.fastq.gz<br>CHM13_Input_1_S1_R2_001.fastq.gz<br>CHM13_Input_2_S2_R1_001.fastq.gz<br>CHM13_Input_2_S2_R2_001.fastq.gz |
| Genome browser session<br>(e.g. UCSC) | Alignment of the CHM13 CENP-A ChIP-seq data to the CHM13 chromosome 8 assembly can be viewed on the UCSC Genome Browser session at the following link: https://genome.ucsc.edu/s/glogsdon1/CHM13_Chr8_CA_ChIP-seq. |

### Methodology

| Replicates | Two independent replicates of CENP-A ChIP-seq (with chromatin input as a control) were performed on CHM13 cells and were in agreement with each other. |
| --- | --- |
| Sequencing depth | All samples were sequenced with 150 bp, paired-end Illumina sequencing, generating a total of 447,609,176 reads. The number of reads associated with each sample is listed below.<br><br>CHM13 CENP-A ChIP (Replicate 1) = 114,230,840 reads<br>CHM13 CENP-A ChIP (Replicate 2) = 131,316,036 reads<br>CHM13 Input (Replicate 1) = 98,173,458 reads<br>CHM13 Input (Replicate 2) = 103,888,842 reads |

| | |
|---|---|
| Antibodies | A mouse monoclonal anti-CENP-A antibody (Enzo, ADI-KAM-CC006-E) was used for the ChIP-seq experiments. |
| Peak calling parameters | All data were aligned to the CHM13 whole-genome assembly containing the contiguous chromosome 8 with the following BWA-MEM parameters: bwa mem -k 50 -c 1000000 {index} {read1.fastq.gz} {read2.fastq.gz}. The resulting SAM files were filtered using SAMtools with FLAG score 2308 to prevent multi-mapping of reads. With this filter, reads mapping to more than one location are randomly assigned a single mapping location, thereby preventing mapping biases in highly identical regions. The ChIP-seq data were downsampled to the same coverage across all datasets and normalized with deepTools bamCompare with the following parameters: bamCompare -b1 {ChIP.bam} -b2 {WGS.bam} --operation ratio --binSize 1000 -o {out.bw}. The resulting bigWig file was visualized on the UCSC Genome Browser using the CHM13 chromosome 8 assembly as an assembly hub. |
| Data quality | Data were quality-checked using FastQC (https://github.com/s-andrews/FastQC), and low-quality end bases were trimmed with Sickle (https://github.com/najoshi/sickle). |
| Software | deepTools bamCompare was used to compare the ratio of ChIP to Input reads aligning to the chromosome 8 centromere. The following parameters were used: bamCompare -b1 {ChIP.bam} -b2 {WGS.bam} --operation ratio --binSize 1000 -o {out.bw}. |

