## [Peer Review File · Nature]

Manuscript Title: The structure, function, and evolution of a complete human chromosome 8**Editorial Notes:****Reviewer Comments & Author Rebuttals****Reviewer Reports on the Initial Version:****Ref #1**

Logsdon and colleagues finish the assembly of human chromosome 8, marking the first telomere-to-telomere assembly of a human autosome. Their approach was to build initial scaffolds with ultra-long ONT reads and then improve the assembly using the more accurate HiFi reads. For their primary assembly, they find barcoded ONT reads with "SUNKs", unique 20-mers from HiFi reads, and then assembled ONT reads based on common SUNKs. They focus on gap closure with an emphasis on three regions: the tandem duplications at the B-defensin gene cluster, the HOR alpha satellite array at the centromere, and the 8q21.2 VNTR (neo-centromere site). In addition to complete the telomere-to-telomere reference for chromosome 8, they infer methylation status and regions of CENP-A enrichment, finding that the likely kinetochore location overlaps hypomethylated regions. They include extensive validation of their assemblies in sequence, organization, and localization of CENP-A enrichment.

The alpha satellite array organization indicates that the youngest and most homogenized repeats are in the center of the array, and more divergent and older repeats occur toward the edges of the array. This is an expected result of unequal exchange within the satellites, and has been observed in other species. Finally, the authors compare the organization of human centromere 8 to homologous centromeres in chimpanzee, orangutan, and macaque to infer the evolutionary history of alpha satellite HOR. They conclude that the HOR organization of alpha satellite originated in the great ape ancestor and that there is an elevated mutation rate at centromeres and in flanking regions.

The work in this paper is not just an exciting technical feat, but offers interesting biological insights into chromosome and centromere evolution. The paper is well organized and well written and their conclusions are justified. My specific comments are below:

Page 5 line 193: I did not see a supplemental figure showing mappability at the centromere. This is important to consider for the ChIP-seq analyses, although the chromatin fiber FISH validates the large scale organization of the CENP-A domain.

Page 5 line 201: You create pairwise alignments of 5kb fragments but it is difficult to imagine the pairwise alignment of HOR regions, since it seems you may end up aligning different monomers. A supplemental figure may help you be clear about what goes into these figures.

Figure 3:

- Although it looks like the VNTR is rich in LINE elements, it isn't easy to glean the sequence composition from the figure as presented.
- The correlation between CENP-A and hypomethylation is not obvious in this figure. Please quantitate the overlap.

Page 7: It is interesting that the CENP-A-enriched domain is over the more variable HOR region rather than in the homogenized region. The correlation between hypomethylation and CENP-A is also interesting. You show that the CENP-A domain is similar in a diploid cell line suggesting that this isn't specific to CHM13 cells. However, all of the hypomethylation analyses seem to be in the CHM13 cells. Is the hypomethylation pattern similar (with

respect to HOR organization) in the diploid human sample HG00733? If there is biological significance to the organization of HOR, comparing between species could be interesting. Are there conserved patterns of methylation with respect to HOR type across species?

Page 7 line 295: "Such increases are rare in the genome..." It is not obvious what you mean here: that it is rare to find divergence increase along a short region of chromosome, or do you mean that such high divergence is rare?

Page 7 line 315: "diploid samples are genetically admixed" -- They are heterozygous for divergent alleles? The HORs are "admixed"? Elsewhere in the paper, you use "admixed" to describe the sequence divergence and organization of the alpha satellite repeats (e.g. Page 5 lines 185,212: 'admixture'; pg 8 line 328 'purity'), which was initially confusing to me.

All of the pairwise sequence comparison plots have different color scales for sequence identity, making comparison between figures difficult. I think that these need to be on the same scale.

Extended data figure 11: It looks like you have lower coverage of ONT and Hifi reads over the macaque alpha satellite dimer array compared to flanking regions. Does this suggest a possible misassembly, read composition bias/underrepresentation of reads rich in alpha satellite dimers, or mapping issue?

Page 12 line 871: "is"  "as"

Ref #2

In this manuscript by Logsdon et al, a new telomere to telomere assembly of human chromosome 8 has been generated and analyzed. The authors use a combination of long-read technologies (PacBio HiFi reads and ONT ultra long reads), based on an assembly produced earlier this year using HiCanu. In addition to this high-quality reference for chromosome 8, the authors present centromere 8 assemblies for a second human sample and three non-human primates (NHPs), macaque, chimpanzee and orangutan. Finally, the authors attempt to demonstrate the utility of this assembly by contextualizing centromere epigenetics and comparative evolution of centromeric satellite arrays. This manuscript is generally well written and presented; however, there are many instances of language that overstates the data (or over simplifies it). There is a lack of clarifying details in the main text for some of the major points in the paper (centromere structure and evolution) that, when filtered out through the supplemental data and referenced works, render some conclusions overstated with the data in hand. A key point in the manuscript is the observation of a hypomethylation dip in the centromere of chromosome 8, recapitulating the observation made in the X chromosome array DXZ1 presented in Nature earlier this year by Miga et al. The authors try to establish a direct link between CENP-A and this hypomethylation, but limitations in the data used herein make this link inferential.

I highlight below some major comments with respect to comments above, as well as minor comments that would improve the manuscript. Overall, this reviewer is left to wonder why the focus of this paper centers on the centromere, with limited data to support some of the claims made, rather than highlighting other parts of the chromosome in more detail.

Line 44 – In the abstract, the authors state that they “complete the orthologous chromosome 8 centromeric regions in chimpanzee, orangutan, and macaque for the first time” but later in the paper they highlight that these are “draft assemblies” (line 368). Again, in the middle of the paper (lines 256-8), the authors state they generated “two contiguous assemblies of the chimpanzee chromosome 8 centromere (one for each haplotype), one haplotype assembly from the orangutan chromosome 8 centromere, and one complete haplotype from the macaque chromosome 8 centromere”. This may be an issue of semantics, but the point of the CHM13 chromosome 8 assembly is to highlight that it is “complete”, but that is a different (and notable!) level of “completeness” than the others, particularly when the boundaries of the actual centromeric regions of the NHPs are being inferred rather than directly annotated (e.g. with CENP-A).

lines 502-3: "The chimpanzee, orangutan, and macaque chromosome 8 centromeres were assembled via the same SUNK-based method." Was the same accuracy method (from Merqury) employed?

Lines 93-94 – The sentence wording here makes it unclear to the reader why the ONT reads harbor more sequence variation (i.e. they are longer).

Line 109 – The wording in the main text is not abundantly clear on whether the primary assembly from Nurk and colleagues was the foundation for this improvement or if a complete reassembly occurred. If this is an improvement, the authors should describe how this improved the Nurk et al assembly (contig reduction, correction, etc) with respect to this previous chr8 assembly.

Lines 139-140 and 151-154 – These statements as worded are confusing. In this region of the new assembly, the authors were able to "resolve one of the largest common inversion polymorphisms in the human genome (3.89Mbp in length)...", which is indicated in Fig 1c. Where is this in relation to the 4.56 Mbp region in the GRCh38 assembly? It is implied that the additional sequence added to this locus to build a 7.06Mbp locus that includes this 3.89Mbp region but where in this sequence with respect to GRCh38 is not clear – a comparison to GRCh38 in the figure would help. How much of the new assembly included this 3.89 inversion or does the new assembly add more information to the surrounding regions allowing for better resolution of the breakpoints? The wording of this sentence is vague in terms of whether CHM13 actually carries the inversion with respect to GRCh38 or if this description refers to the fact there is a polymorphic inversion that can be found in some human genomes at this location. In the last sentence of this paragraph the authors state that resolution of the alternate haplotype is important since the inverted haplotype predisposes to various human disorders, but again, is the inversion found in CHM13? The wording of this section is not explicitly clear. In Figure 1d, the data that was used to determine copy number polymorphisms in the human population should be referenced.

Line 171 – And Ext Data fig 6a, b – It is highly unusual to put just a single chromosome in a representative image – a full cell should be shown with an inset to each of the two chromosome 8's. How were the probes designed? The phrase "stretched chromosome" is not really very scientific. What exactly do the authors mean? The methods state that a cytofuge was used, but how does this equate to a "stretched chromosome"?

Lines 174-5 – The authors state the PFGE "recapitulates" the size predicted, but with what margin of specificity? This should be clearly stated as the PFGE is not as high a resolution as the sequencing. (in other words, "recapitulates" is an overstatement.... "supports" might be a better term.)

Line 176-178 – This is a red flag for the paper - what is this diploid genome? You have to dig through the supplements to find that it is HG00733, but nowhere are genome assembly statistics, methods, etc provided for this sample. Thus, it makes rendering how statistically meaningful the comparisons to this CHM13 assembly really are.

Line 190-195 – This is another red flag for the paper – it appears the authors are trying to make a very specific statement about functional annotations of the centromere of chromosome 8 in CHM13 that are only inferential since the ChIP-seq data employed herein is not actually from CHM13 but different data sets from earlier studies. The authors cannot state that they determined the epigenetic centromere for this specific chromosome – they can only infer where it is. This is an important point, particularly given the samples that were used for the ChIP-seq are neither normal diploid lines (contrary to the statement in the supplemental figure legend), but rather are two abnormal cell lines with neocentromeres. This limitation should be made clearer in the language used throughout on this point. Moreover, the recent observations that CENP-A positionally shifts in the genomic space of a centromere among different individual plants of the same species (see Gent et al 2017) make this limitation, and the links to methylation, a crucial point to clarify.

Ext data figure 9 – The authors show ChIP-seq peaks for both IMS13q and MS4221, and the scale and heights are virtually identical. It is not clear how this can be the case when IMS13q has two active centromeres at the same location on each homolog of chromosome 8, but MS4221 only has one active centromere at this location. How are the methylation reads and these variable ChIP-seq reads being normalized to afford a realistic comparison among these different centromere haplotypes?

Lines 230-231 and Ext data figure 10 – The rationale behind the comment that fiber FISH allows copy resolution is tenuous. How can the authors distinguish copy numbers versus the signal that occurs on fibers due to chromatin breakage during the FISH technique?

Ext Data Table 4 – The quality of NHPs assemblies are low that that of the presented CHM13 - this should be explicitly stated in the main text since direct comparisons are being made to a complete assembly of human 8.

Lines 297-301, 386-7 and 861-880, Do the authors consider the lower recombination rates at centromeres compared to chromosome arms? This would affect rate models for neutral evolution.

Lines 320-323. This is another overstatement of the data. The hypomethylated region is not shown herein to be enriched for CENP-A in this genome – rather it is inferred from orthogonal data from different individuals.

Lines 395-7 –In the following statement, the authors link multiple statements to a small number of references that are, taken in the context of the entire sentence, incorrect. Molecular drive, including gene conversion, was first described in the early '80s (Dover et al 1982). Refs 48-49 present a model for centromere drive that links observed rapid rates of satellite sequences to rapid rates of centromere binding sequences, but do not present direct evidence for “rapid centromere sequence evolution as a driving force in speciation” nor “hybrid incompatibility”. Other studies have made the link between rapid satellite evolution and hybrid incompatibility that do not necessarily involve centromeres (for example Ferree and Barbash 2009), so this overstatement does not reflect these other satellite observations. Moreover, HOR divergence, specifically, has not been shown to cause speciation nor hybrid incompatibility. Finally, reference 50 is for a study in monkeyflowers that links a meiotic driver to an allele (D) adjacent to a centromere, but the sequence of that allele is not directly defined as an HOR array. The sentence is awkward as stated (“rapid evolution ...due to the accumulation of sequence differences” is circular). As worded, this sentence makes a grand and defining statement about speciation that both simplifies and overstates the current state of the field and does not consider the many other factors involved, such as inversions, repressed recombination, meiotic drive, etc.

Lines 756 – The title of the methods section should be changed to reflect the content; the order is reversed with respect to the order data is presented in the main text.

Lines 506-508 – What are the mapped kmers being used? This is awkward wording, making the kmers referred to unclear. Why are the authors using Jellyfish instead of Merqury for the validation?

Ref #3

The manuscript entitled “The structure, function, and evolution of a complete human chromosome 8” presents first linear assembly of a complete human autosome, including the centromeres and complex repetitive regions, as well as a in depth description of the function and evolution of complex centromeric regions. Overall it is clear that this is a significant achievement that required a large amount of effort and ingenuity. I think the paper could be acceptable for publication as is, but I do have a few comments that might improve the presentation to a broader audience.

- 1) In the opening sentence it is not completely clear if the authors are referring to complex repeats within segmental duplications or the segmental duplications themselves. I suspect it is actually both, but it may be worth rewording for clarity.
- 2) On line 84 where they mention “neocentromere” for the first time it might be useful to call it a “recurrent polymorphic neocentromere” or add some additional description, as read it sounds like they are saying that the primary human centromere is a neocentromere, although the rest of the manuscript makes it clear that this is not the case.
- 3) Although this assembly is clearly a significant feat, I think readers may ask why all chromosomes have not been assembled in the same way given that they are all (all but X) present in the ONT and a PacBio data that were generated for this study. Chromosome 8 is a reasonable first target, but why not another or all other

chromosomes? A bit more discussion of the challenges that are likely to be met in putting together the other chromosomes could be useful to address this.

4) For Pac Bio data it seems a bit strange to use the term “mapped” in relation to methylated cytosines (e.g. line 188, 237). Would it be better to say something like - we inferred the location of methylated cytosines based on PacBio polymerase kinetics – or something to that effect?

5) “While this is consistent with the VNTR being the potential site of the functional kinetochore of the neocentromere, sequence and assembly of the neocentromere containing cell line will be critically important” - Given that it is stated earlier in the paragraph that there are multiple unrelated instances of neocentromere formation, should this be extended to mention a need to sample other individuals/cell lines?

6) On Line 344 the “Smith model” is mentioned. This seems to refer to a reference earlier in the paragraph, but it would be useful to tie this to a specific reference in the sentence where it is mentioned.

Sincerely,

Jeremiah Smith

Ref #4

Manuscript: 2020-09-16437

Title: The structure, function, and evolution of a complete human chromosome 8

Authors: Logsdon et al.

The authors present the first telomere-to-telomere assembly of a human autosome, chromosome 8. A combination of ONT and HIFI data allowed to bridge three previously remaining gaps, all composed of (multi-)megabase sized near-identical repeat clusters not amenable to sequence assembly before the introduction of ultra-long ONT and Pacbio Sequel II Hifi technology. Besides presenting a blueprint for a method to produce finished human chromosome assemblies the authors demonstrate the importance of gap-free sequences. They reveal the complexity and structural variability of the β -defensin cluster which is important, as SV and CNV at this gene cluster is associated with important diseases. Similarly, the structure of the 8q21.2 VNTR region is fully resolved which is a prerequisite for testing structural features at this locus for their importance in neocentromere formation which has been reported for this locus.

Sequencing and assembly of entire centromeres of human chromosomes was shown before (Jain et al. 2018, Miga et al. 2020). The exciting key example of this study though for demonstrating the importance of a gap free reference sequence as a basis for understanding basic questions in biology is the presentation of the entire Chr8 centromere including methylation status of the alpha-satellite HOR arrays, postulating the most likely position of the functional kinetochore. Chromosome 8 centromere organization was then analysed in an evolutionary context: orthologous chr8 centromeres were produced for chimpanzee, orangutan and macaque. Shared and distinct repeat organization, overall structural organization and mutation frequencies were studied, providing the basis for new hypothesis to be tested broadly after more autosome centromeres are sequenced in human and apes.

The study is well written and illustrated with intuitive figures. Abstract, introduction and discussion are clear and rather straightforward also for people outside of the human genome research, however, some nomenclature could make reading for non-biologists harder. One example: human chromosome banding nomenclature doesn't need to be explained necessarily, however, 8p23.1 for the β -defensins should probably also be shown in Fig 1a for making tracking of content easier, as 8q21.2 is shown. Major conclusions based on the structural genomic findings were typically validated by independent technical methods. I could not find any indication of flaws.

Initially, I was worried whether we have to expect now gapfree sequence papers for all 22 human autosomes as recently Miga et al. already reported a gapfree x-chromosome as part of a whole genome de novo assembly of the same cell line CHM13. I think the biology and story presented here justifies a chromosome 8 specific publication, however, I think the authors must be more transparent about what datasets were newly developed here and what data was recycled from Miga et al. – a table, listing previously published datasets and newly generated datasets, including all accession codes (not only project IDs) would be very useful and desired, especially as Miga et al. did not fully disclose all their data as datasets submitted to NCBI.

Miga et al. report already the chromosome 8 centromere and methylation status (Miga et al. extended data fig. 10) – please comment!

Miga et al. used the kmer based ONT ultra-long read barcoding approach for centromere and other repeat-based gap regions. Here Logsdon et al. give the impression of a newly established approach. Since the papers are from the same group of authors, still a transparent referencing should be expected, or – Logsdon should try to explain better what exactly was improved in comparison to the previous Miga et al. paper or clearly state that the same approach was used (maybe by including more de novo generated data).

All sequencing data was generated generically from whole genomic DNA. The technical advance for centromere assembly was featured here, however for a single autosome similarly as previously done for the human X chromosome. If the authors are not following a strategy for publishing now 20 more chromosome papers, I think it would be very instructive, not only to human genome researchers, to get a better understanding of the challenges remaining for full assembly of the other autosome centromeres. Miga et al. have briefly mentioned this problem and indicated that some autosomal centromeres will be easier assembled than others. I suppose, at least the Logsdon et al datasets and established computational pipelines should be similarly applicable to the whole genome assembly of CHM13. Please give a better understanding of what are the expected limitations for the assembly of other chromosomes? Or are the other centromeres assembled in fact already to similar quality? Why is this not a process for parallelization? For example, I am impressed by the 7 Mbp alpha-satellite repeat assembled for Macaque - this result is of general importance for genome assembly also outside human research. Especially crop genomes can have centromeres much larger than those of human autosomes, so more detail about the true potential and/or limitation of the presented approach beyond human would make this study more attractive for a broader audience.

Author Rebuttals to Initial Comments:

Referee comments are in black; our responses are in blue

Referee #1 (Remarks to the Author):

Logsdon and colleagues finish the assembly of human chromosome 8, marking the first telomere-to-telomere assembly of a human autosome. Their approach was to build initial scaffolds with ultra-long ONT reads and then improve the assembly using the more accurate HiFi reads. For their primary assembly, they find barcoded ONT reads with "SUNKs", unique 20-mers from HiFi reads, and then assembled ONT reads based on common SUNKs. They focus on gap closure with an emphasis on three regions: the tandem duplications at the B-defensin gene cluster, the HOR alpha satellite array at the centromere, and the 8q21.2 VNTR (neo-centromere site). In addition to complete the telomere-to-telomere reference for chromosome 8, they infer methylation status and regions of CENP-A enrichment, finding that the likely kinetochore location overlaps hypomethylated regions. They include extensive validation of their assemblies in sequence, organization, and localization of CENP-A enrichment.

The alpha satellite array organization indicates that the youngest and most homogenized repeats are in the center of the array, and more divergent and older repeats occur toward the edges of the array. This is an expected result of unequal exchange within the satellites, and has been observed in other species. Finally, the authors compare the organization of human centromere 8 to homologous centromeres in chimpanzee, orangutan, and macaque to infer the evolutionary history of alpha satellite HOR. They conclude that the HOR organization of alpha satellite originated in the great ape ancestor and that there is an elevated mutation rate at centromeres and in flanking regions.

The work in this paper is not just an exciting technical feat, but offers interesting biological insights into chromosome and centromere evolution. The paper is well organized and well written and their

conclusions are justified. My specific comments are below:

We thank the referee for their kind comments and insightful suggestions to improve this paper. Based on the referee's suggestions, we 1) assessed the mappability of the chromosome 8 centromere (**new Extended Data Fig. 19**), 2) analyzed the HOR organization and methylation status of the HG00733 chromosome 8 centromeres (**updated Extended Data Fig. 8**), 3) quantified the overlap between CENP-A and the hypomethylated regions within the 8q21.2 VNTR (main text), and 4) presented new schematics/data to help the reader understand our methods and results (**new Extended Data Figs. 12, 20, and 21**).

Page 5 line 193: I did not see a supplemental figure showing mappability at the centromere. This is important to consider for the ChIP-seq analyses, although the chromatin fiber FISH validates the large scale organization of the CENP-A domain.

To assess the mappability of short reads within centromeric α -satellite, we performed a simulation where we generated 300,000 random 150 bp fragments from five equally sized regions across the CHM13 chromosome 8 D8Z2 HOR array. We, then, mapped the fragments from each region back to the assembly using BWA-MEM or the k-mer-based mapping approach we developed in order to assess their specificity. As expected, there is some crosstalk among α -satellite repeats; however, overall, we find that the reads preferentially map to the same regions from which they originated. Importantly, both approaches show that the reads do not preferentially map to the hypomethylated region and, thus, mapping biases do not account for our observations. We have included this analysis as the **new Extended Data Fig. 19** (below).

Extended Data Figure 19. Mappability of short reads within the D8Z2 α -satellite HOR array. To determine the mappability of short reads within the chromosome 8 centromeric HOR array, we performed a simulation where we generated 150 bp fragments from five 416 kbp regions across the HOR array and mapped them back to the D8Z2 α -satellite HOR array using **a)** BWA-MEM or **b)** our k-mer-based mapping approach (**Methods**). We find that the fragments mapped preferentially to the regions from which they originated. Importantly, we find that these fragments do not preferentially map to the hypomethylated region, where CENP-A is predicted to be located (**Fig. 2a**, **Extended Data Fig. 11**), indicating that mapping biases are not at an appreciable level in this region.

Page 5 line 201: You create pairwise alignments of 5kb fragments but it is difficult to imagine the pairwise alignment of HOR regions, since it seems you may end up aligning different monomers. A supplemental figure may help you be clear about what goes into these figures.

We agree that the details of this alignment procedure would benefit from a schematic. To address this point, we generated a **new Extended Data Fig. 12** (below) to illustrate how 5 kbp fragments are aligned, scored, and plotted to form a pairwise sequence identity heat map. To specifically address the question of how α -satellite HORs are aligned to each other, we included an example of a 5 kbp fragment that contains multiple α -satellite HORs to show how suboptimal alignments are eliminated and only the best alignment between two fragments is plotted. We include this figure in the Extended Data and reference it in the main text.

Example alignment between two 5 kbp α-satellite-containing fragments:

Calculate the sequence identity from the best alignment between each pair, and store it in an $N \times N$ matrix

		j	$j+1$	$j+2$	$j+3$...	N
j		95%	73%	82%	87%		76%
$j+1$		73%	84%	99%	99%		99%
$j+2$		82%	95%	95%	95%		95%
$j+3$		87%	99%	99%	99%		99%
							95%
							99%
							99%

Plot the sequence identity between each pair as a single data point in an $N \times N$ matrix

Extended Data Figure 12. Process to generate a pairwise sequence identity heat map. To generate a pairwise sequence identity heat map, we first fragment the region of interest into 5 kbp non-overlapping fragments. Then, we align every 5 kbp fragment to every other 5 kbp fragment using minimap²¹, retaining only the best local alignment between each pair (**Methods**). As an example, we illustrate the alignment between two 5 kbp

fragments originating from the D8Z2 centromeric HOR array. Here, a target fragment and a query fragment are aligned to each other, generating multiple potential alignments. The highest-scoring alignment occurs between those with the most similar sequence and structure, and this guarantees registry among HORs wherever possible. Sequence identity is calculated from the highest-scoring alignment between each pair (**Methods**) and stored in an $N \times N$ matrix, while the others are discarded. Sequence identities between each pair are plotted as data points in R using ggplot2 such that the highest identities are dark red and the lowest identities are dark purple. Since the $N \times N$ matrix has identical information on each side of the diagonal, only one half of the matrix is presented.

Figure 3:

- Although it looks like the VNTR is rich in LINE elements, it isn't easy to glean the sequence composition from the figure as presented.
- The correlation between CENP-A and hypomethylation is not obvious in this figure. Please quantitate the overlap.

We modified **Fig. 3a** (below) to include an inset that shows the repeat content of the 8q21.2 VNTR repeat unit. Approximately 37% of the repeat unit is composed of LINE repeat elements, and another ~15% is composed of SINEs, LTRs, low-complexity repeats, DNA repeats, and simple repeats.

To quantitate the overlap between CENP-A and the hypomethylated portion of the 8q21.2 VNTR array, we first used CHM13 ONT data to determine that approximately 13% of all available CpG dinucleotides within the array are methylated. If we restrict this to the ~9 kbp portion of each repeat unit lacking the GOR1/REXO1L1 gene, this percentage drops to ~10%. If we measure the percentage of all CENP-A ChIP reads that reside within this hypomethylated ~9 kbp portion, we find that approximately 98% of the ChIP signal is located within the region. Therefore, we conclude that nearly all of the CENP-A signal is located within the hypomethylated ~9 kbp portion of the 8q21.2 VNTR repeat unit. We have added this quantitation to the main text and figure legend.

Page 7: It is interesting that the CENP-A-enriched domain is over the more variable HOR region rather than in the homogenized region. The correlation between hypomethylation and CENP-A is also interesting. You show that the CENP-A domain is similar in a diploid cell line suggesting that this isn't specific to CHM13 cells. However, all of the hypomethylation analyses seem to be in the CHM13 cells. Is the hypomethylation pattern similar (with respect to HOR organization) in the diploid human sample HG00733? If there is biological significance to the organization of HOR, comparing between species could be interesting. Are there conserved patterns of methylation with respect to HOR type across species?

As requested by the referee, we compared the methylation and HOR organization patterns of the HG00733 and CHM13 chromosome 8 centromeres. To do this, we generated an additional 51.7 Gbp (16.7-fold coverage) of HG00733 ONT data (**new Extended Data Table 2**, below), which allowed us to obtain the coverage needed to unambiguously detect methylated cytosines on both HG00733 chromosome 8 haplotypes. As shown in the updated **Extended Data Fig. 8** below, we find that both HG00733 chromosome 8 centromeres are heavily methylated across the alpha-satellite HOR array, except for a 77 or 95 kbp hypomethylated region located within a highly variable region on the two different alleles. This is consistent with our initial observation for the chromosome 8 centromere in CHM13 (**Fig. 2a**), where the alpha-satellite HOR array is methylated except for a 73 kbp region located within a region of diverse HORs. Because the methylation and HOR organization patterns of the CHM13 chromosome 8 centromere are consistent with those in the diploid HG00733 chromosome 8 centromeres, we conclude that these are general features of human chromosome 8 centromere organization relevant to kinetochore function.

Extended Data Table 2. Sequence and assembly of the HG00733 genome.

Species	Assembly*			PacBio HiFi data		ONT data	
	Size (Gbp)	No. of contigs	N50 (Mbp)	Sequencing depth [†]	Read N50 (kbp)	Sequencing depth [†]	Read N50 (kbp)
Human (HG00733)	6.08	1,592	34.89	33.48	13.5	94.0	32.7

*Assembled from PacBio HiFi data with hifiasm (Cheng et al., *arXiv*, 2020)

[†]Assumes a 3.1 Gbp genome

Extended Data Figure 8. Sequence, structure, and epigenetic map of human diploid HG00733 chromosome 8 centromeres. a,b) Repeat structure, alpha-satellite organization, methylation status, and sequence identity heat map of the **a)** maternal and **b)** paternal chromosome 8 centromeric regions from a diploid human genome (HG00733; **Extended Data Table 2**) shows structural, evolutionary, and epigenetic similarity to the CHM13 chromosome 8 centromeric region (**Fig. 2a**).

We note that the generation of ONT data with a suitable read length distribution is a laborious and time-consuming process, requiring 1-2 months per sample. Given the time constraints and more limited access to the wet-bench, we thought it most relevant to focus on assessing the human diploid sample rather than focusing on the nonhuman primates, which would have required us to generate hundreds of Gbp of additional ONT data for each genome in order to perform this analysis.

Page 7 line 295: "Such increases are rare in the genome..." It is not obvious what you mean here: that it is rare to find divergence increase along a short region of chromosome, or do you mean that such high divergence is rare?

We meant to say that high divergence is rare. We reworded this sentence to state the following (changes in **bold**), which we hope has improved clarity:

"We find that the mean **allelic** divergence increases more than threefold as the sequence transitions from unique to monomeric α -satellite. Such increases **in divergence** are rare in the **human** genome based on sampling of at least 19,926 random loci, where only 1.27-1.99% of loci show comparable levels of divergence (**Fig. 5c**)."

Page 7 line 315: "diploid samples are genetically admixed" -- They are heterozygous for divergent alleles? The HORs are "admixed"? Elsewhere in the paper, you use "admixed" to describe the sequence divergence and organization of the alpha satellite repeats (e.g. Page 5 lines 185,212: 'admixture'; pg 8 line 328 'purity'), which was initially confusing to me.

The referee is correct. We use the term "admixed" or "admixture" in different contexts. We revised the following sentences accordingly to avoid any confusion (changes in **bold**, below):

Sentence originally on line 315: "It should be noted that both the human and chimpanzee diploid samples **carry diverse haplotypes**, and it is possible that this **allelic** heterogeneity facilitated the partitioning of reads and the reconstruction of both haplotypes from these samples.

Sentence originally on line 185: "Interestingly, we find that HORs are differentially distributed regionally across the centromere. While most regions **show a mixture of** different HOR types..."

Sentence originally on line 212: "The fourth layer is the largest and defines the bulk of the HOR α -satellite (1.42 Mbp in total). It shows the greatest **variety of** different HOR subtypes...."

Sentence originally on line 328: "Although the HOR units are derived from the original 11-mer repeat, the degree of **HOR diversity** varies considerably across **different regions of** the centromere..."

All of the pairwise sequence comparison plots have different color scales for sequence identity, making comparison between figures difficult. I think that these need to be on the same scale.

For the purpose of comparison, we now show all primate centromere sequence identity plots on the same scale in the following two figures: **Extended Data Figs. 20** and **21**. Using this uniform scale, however, we note that the evolutionary layers become more difficult to see and interpret. Because of this, we chose to keep the plots with different color scales as main figures but point the reader to the Extended Data Figures in the legends for those who would like to visualize them all on identical scales.

Extended Data Figure 20. Sequence identity maps of human chromosome 8 centromeric regions, presented on the same color scale. a-c) Pairwise sequence identity map of the a) CHM13, b) HG00733 maternal, and c) HG00733 paternal chromosome 8 centromeric regions. All maps are shown on the same color scale and are consistent with those shown in Extended Data Fig. 21.

Extended Data Figure 21. Sequence identity maps of nonhuman primate chromosome 8 centromeric regions, presented on the same color scale. a-d) Pairwise sequence identity maps of the a) chimpanzee (H1), b) chimpanzee (H2), c) orangutan, and d) macaque chromosome 8 centromeric regions. All maps are shown on the same color scale and are consistent with those shown in Extended Data Fig. 20. H1, haplotype 1; H2, haplotype 2.

Extended data figure 11: It looks like you have lower coverage of ONT and Hifi reads over the macaque alpha satellite dimer array compared to flanking regions. Does this suggest a possible misassembly, read composition bias/underrepresentation of reads rich in alpha satellite dimers, or mapping issue?

The coverage within the macaque alpha-satellite dimer array is actually consistent with the expected level for a properly resolved single haplotype. However, the coverage on the flanking regions is double the expectation. This suggests that the sequence flanking the centromere is so identical that it is indistinguishable and the flanking pericentromeric haplotypes are more difficult to resolve. To make this clearer, we included an additional explanation in the figure legend: **“The increase in coverage on the edges of the macaque centromeric region is due to the inability to resolve the two haplotypes flanking the centromeric satellite array. Our results suggest that there are too few allelic differences to distinguish the flanking haplotypes. The macaque α -satellite dimer array, however, is fully resolved and does not show any signs of sequence collapse.”**

Page 12 line 871: "is"  "as"

Thank you for pointing this typo out. We fixed this sentence.

Referee #2 (Remarks to the Author):

In this manuscript by Logsdon et al, a new telomere to telomere assembly of human chromosome 8 has been generated and analyzed. The authors use a combination of long-read technologies (PacBio HiFi reads and ONT ultra long reads), based on an assembly produced earlier this year using HiCanu. In addition to this high-quality reference for chromosome 8, the authors present centromere 8 assemblies for a second human sample and three non-human primates (NHPs), macaque, chimpanzee and orangutan. Finally, the authors attempt to demonstrate the utility of this assembly by contextualizing centromere epigenetics and comparative evolution of centromeric satellite arrays. This manuscript is generally well written and presented; however, there are many instances of language that overstates the data (or over simplifies it). There is a lack of clarifying details in the main text for some of the major points in the paper (centromere structure and evolution) that, when filtered out through the supplemental data and referenced works, render some conclusions overstated with the data in hand. A key point in the manuscript is the observation of a hypomethylation dip in the centromere of chromosome 8, recapitulating the observation made in the X chromosome array DXZ1 presented in Nature earlier this year by Miga et al. The authors try to establish a direct link between CENP-A and this hypomethylation, but limitations in the data used herein make this link inferential.

I highlight below some major comments with respect to comments above, as well as minor comments that would improve the manuscript. Overall, this reviewer is left to wonder why the focus of this paper centers on the centromere, with limited data to support some of the claims made, rather than highlighting other parts of the chromosome in more detail.

For the main text, we chose to focus on the centromere because this is where most of the novel insights (e.g., evolutionary reconstruction, hypomethylated regions, and the potential similarities with the neocentromere structure on 8q21.2) were revealed. We could have gone into much more detail of many aspects with respect to finishing of the chromosome and other regions (e.g., β -defensin structural

variation, copy number differences in the population, new gene models, etc.). We opted to elaborate more extensively on these regions in the Extended Data.

With that said, we thank the referee for their review of our manuscript. We spent the last two months generating more extensive data to support the major findings of the paper. Addressing the comments and criticisms made by the referee improved the manuscript as a whole, including the addition of 1) two new independent replicates of CENP-A ChIP-seq data from the CHM13 cell line, which confirms the location of the centromeric histone CENP-A within a 632 kbp stretch encompassing the hypomethylated dip of the centromeric α -satellite HOR array (**Fig. 2a, Extended Data Fig. 11**); 2) new analyses and schematic to show the inverted orientation of the β -defensin locus relative to GRCh38 (**Extended Data Fig. 6**); 3) new images showing a full metaphase chromosome spread with both chromosome 8s shown as insets (**Extended Data Fig. 7b,c**); 4) a new table listing the sequencing data and assembly statistics of the HG00733 genome (**Extended Data Table 2**); 5) new evolutionary analyses to show that lower recombination rates at centromeres are unlikely to affect sequence divergence estimates (**Rebuttal Figs. 1,2**); and 6) additional details in the Results, Discussion, Methods, and Figure Legends to clarify the data and methods presented.

Line 44 – In the abstract, the authors state that they “complete the orthologous chromosome 8 centromeric regions in chimpanzee, orangutan, and macaque for the first time” but later in the paper they highlight that these are “draft assemblies” (line 368). Again, in the middle of the paper (lines 256- 8), the authors state they generated “two contiguous assemblies of the chimpanzee chromosome 8 centromere (one for each haplotype), one haplotype assembly from the orangutan chromosome 8 centromere, and one complete haplotype from the macaque chromosome 8 centromere”. This may be an issue of semantics, but the point of the CHM13 chromosome 8 assembly is to highlight that it is “complete”, but that is a different (and notable!) level of “completeness” than the others, particularly when the boundaries of the actual centromeric regions of the NHPs are being inferred rather than directly annotated (e.g. with CENP-A).

We regard both the CHM13 and nonhuman primate (NHP) chromosome 8 centromere assemblies as complete. Indeed, numerous bioinformatic analyses, including long-read data ONT and HiFi comparisons, short-read data read-depth analysis, and TandemQUAST, support the structure and sequence of all of the assemblies. Merqury estimates that the NHP centromeres are 99.9988-100% accurate (QV score >49.3). The distinction we wish to make is, unlike the CHM13 centromere assembly, the NHP centromere assemblies are not yet validated with wet-lab experimental techniques such as Southern blot or ddPCR. These two wet-lab techniques are currently very difficult to perform on diploid genomes in repeat regions that are undergoing accelerated evolution. We attempted a pulsed-field gel Southern blot on the chimpanzee chromosome 8 centromeres and were unable to achieve a satisfactory and interpretable result that would validate the assemblies. As such, we refer to the NHP assemblies as “**high-quality** draft assemblies”, and we revised the abstract and the main text to keep this designation consistent.

lines 502-3: “The chimpanzee, orangutan, and macaque chromosome 8 centromeres were assembled via the same SUNK-based method.” Was the same accuracy method (from Merqury) employed?

Merqury estimates that all NHP centromere assemblies are 99.9988-100% accurate (QV score 49.3- Infinity), as mentioned above. We now state these accuracy estimations in the main text (lines 273-274) and include the details of this analysis in the Methods.

Lines 93-94 – The sentence wording here makes it unclear to the reader why the ONT reads harbor

more sequence variation (i.e. they are longer).

Thank you for pointing this out. We revised this sentence as follows (changes in **bold**):

“We reasoned that ultra-long (>100 kbp) ONT reads **are long enough to** harbor sequence variation **that could** permit the assembly of complex regions, generating an initial sequence scaffold that could be replaced with highly accurate PacBio high-fidelity (HiFi) contigs to improve the overall base accuracy.”

Line 109 – The wording in the main text is not abundantly clear on whether the primary assembly from Nurk and colleagues was the foundation for this improvement or if a complete reassembly occurred. If this is an improvement, the authors should describe how this improved the Nurk et al assembly (contig reduction, correction, etc) with respect to this previous chr8 assembly.

Nurk and colleagues generated the primary chromosome 8 assembly, which had five gaps that we resolved with our targeted assembly method (**Fig. 1a**). The rest of the chromosome 8 assembly was correctly assembled, as determined by numerous orthogonal technologies, including long- and short- read mapping, Bionano Genomics data, and Merqury, and did not need further correction. Therefore, the Nurk et al. chromosome 8 assembly served as the sequence backbone in which we placed our assemblies. To reflect this, we updated the main text and Methods as follows (changes in **bold**):

Main text:

“We improved the base-pair accuracy of the sequence scaffolds by replacing the raw ONT sequence with several concordant PacBio HiFi contigs and integrating them into **the same** linear assembly of human chromosome 8 **generated by Nurk and colleagues**¹¹ (**Fig. 1b; Methods**).”

Methods:

“...the sequence scaffold for each target region was incorporated into the CHM13 chromosome 8 assembly **generated by Nurk and colleagues**¹¹, thereby filling the gaps in the chromosome 8 assembly.”

Lines 139-140 and 151-154 –These statements as worded are confusing. In this region of the new assembly, the authors were able to “resolve one of the largest common inversion polymorphisms in the human genome (3.89Mbp in length)...”, which is indicated in Fig 1c. Where is this in relation to the 4.56 Mbp region in the GRCh38 assembly? It is implied that the additional sequence added to this locus to build a 7.06Mbp locus that includes this 3.89Mbp region but where in this sequence with respect to GRCh38 is not clear – a comparison to GRCh38 in the figure would help. How much of the new assembly included this 3.89 inversion or does the new assembly add more information to the surrounding regions allowing for better resolution of the breakpoints? The wording of this sentence is vague in terms of whether CHM13 actually carries the inversion with respect to GRCh38 or if this description refers to the fact there is a polymorphic inversion that can be found in some human genomes at this location. In the last sentence of this paragraph the authors state that resolution of the alternate haplotype is important since the inverted haplotype predisposes to various human disorders, but again, is the inversion found in CHM13? The wording of this section is not explicitly clear. In Figure 1d, the data that was used to determine copy number polymorphisms in the human population should be referenced.

CHM13 does, in fact, carry the fully-sequenced inverted allele, in contrast to GRCh38, which represents the directly-oriented allele. To make this clearer, we generated a new figure (**Extended Data Figure 6**; below), which shows the organization and orientation of the CHM13 β -defensin locus relative to GRCh38. Importantly, this figure shows that the region is *inverted* in CHM13 relative to GRCh38. We show that there is approximately 602.6 kbp more duplicated sequence within the CHM13 haplotype compared to GRCh38. Additionally, there is approximately 9.3 kbp of non-duplicated sequence present in the inverted region in CHM13. Altogether, there is 611.9 kbp of novel sequence in the CHM13 haplotype, and we include this quantitation in the figure legend.

Extended Data Figure 6. Comparison of the CHM13 and GRCh38 β -defensin loci. Miropeats comparison of the CHM13 and GRCh38 β -defensin loci identifies a 4.11 Mbp inverted region (dashed gray line) bracketed by proximal and distal segmental duplication (dup) blocks (black and blue arrows) in CHM13. CHM13 also has an additional segmental duplication block (blue arrow) relative to the GRCh38. In total, the CHM13 haplotype adds 611.9 kbp of new sequence, of which 602.6 kbp is located within segmental duplication blocks and 9.3 kbp is located at the distal edge of the inverted region. Colored segments track blocks of homology between CHM13 and GRCh38.

The data used in the copy number polymorphism analysis shown in **Fig. 1d** was already cited in the Methods, but we now reference this in the **Fig. 1** legend as well.

Line 171 – And Ext Data fig 6a, b – It is highly unusual to put just a single chromosome in a representative image – a full cell should be shown with an inset to each of the two chromosome 8's. How were the probes designed? The phrase “stretched chromosome” is not really very scientific. What exactly do the authors mean? The methods state that a cytofuge was used, but how does this equate to a “stretched chromosome”?

We agree that a full metaphase chromosome spread with both chromosome 8s well-positioned and clearly hybridized with FISH probes would be ideal to include. This is not trivial to obtain with cytopun, or stretched, chromosomes (which are chromosomes that have been mechanically elongated by centrifugal force; see Laan et al., *Genome Res*, 1995; Cerutti et al., *Molecular Cytogenetics*, 2016; and Claussen et al., *Cytogenet Cell Genet*, 1994 for more details) because chromosomes are frequently

lost, occluded, or stretched out of the field of view. Indeed, the original image we showed had one chromosome 8 that was sufficiently stretched and clearly showed the FISH probe location, but the other chromosome 8 was occluded by other chromosomes, which we didn't think would be informative to present as an image.

To address this request, we spent the last several weeks repeating and optimizing the FISH experiment to obtain images of full metaphase chromosome spreads that show the organization of each chromosome 8 centromere clearly. We obtained two such images, shown in **Extended Data Fig. 7b,c** (below). Using probes that target five different regions of the chromosome 8 centromere (**Panel a**), we confirm the long-range order and organization of the chromosome 8 centromere (**Panels b,c**). We modified the text describing this experiment in the Results, Methods, and Figure Legend. Additionally, we provided additional details on how the probes were designed and generated in the Methods. These results are consistent with our other experimental and computational analyses of the chromosome 8 centromere.

**b****c****d**
Extended Data Figure 7. Validation of the CHM13 chromosome 8 centromeric region.

a) Coverage of CHM13 ONT and PacBio HiFi data along the CHM13 chromosome 8 centromeric region (top two panels) is largely uniform, indicating a lack of large structural errors. Analysis with TandemMapper and TandemQUAST⁶, which are tools that assess repeat structure via mapped reads (third panel) and misassembly breakpoints (fourth panel; red), indicates that the chromosome 8 D8Z2 α -satellite HOR array lacks large-scale assembly errors. Five different FISH probes targeting regions in

the chromosome 8 centromeric region (bottom panel) are used to confirm the organization of the α -satellite DNA (**Panels b,c**). **b,c**) Representative images of metaphase chromosome spreads hybridized with FISH probes targeting regions within the chromosome 8 centromere (**Panel a**). Insets show both chromosome 8s (outlined with a dashed line) with the predicted organization of the centromeric region. **d**) Droplet digital PCR (ddPCR) of the chromosome 8 D8Z2 α -satellite array indicates that there are 1344 +/- 142 D8Z2 HORs present on chromosome 8, consistent with the predictions from an *in silico* restriction digest and StringDecomposer¹⁰ analysis (**Methods**). Mean +/- s.d. is shown. Bar = 5 microns. Insets = 2.5x magnification.

Lines 174-5 – The authors state the PFGE “recapitulates” the size predicted, but with what margin of specificity? This should be clearly stated as the PFGE is not as high a resolution as the sequencing. (in other words, “recapitulates” is an overstatement.... “supports” might be a better term.)

We changed the wording in this sentence to “supports” instead of “recapitulates”.

Line 176-178 – This is a red flag for the paper - what is this diploid genome? You have to dig through the supplements to find that it is HG00733, but nowhere are genome assembly statistics, methods, etc provided for this sample. Thus, it makes rendering how statistically meaningful the comparisons to this CHM13 assembly really are.

HG00733 is a human genome of Puerto Rican origin that was originally analyzed as part of the 1000 Genomes Project (Auton et al., *Nature*, 2015) but has now become a reference standard for both structural variation analyses (Chaisson et al., *Nature Comm*, 2019) and phased genome assembly (Porubsky et al., *Nature Biotechnol*, 2020). Although the centromeres were not assembled previously for this sample, the underlying HiFi data and ONT data were generated and publicly released, allowing us to apply our assembly methods for centromere sequence assembly. We successfully assembled both chromosome 8 centromeres and used the data to determine whether the CHM13 assembly was reproducible and representative.

We revised the main text to refer to this genome as “HG00733” (line 188), and we include the sequence and assembly statistics for this genome in the new **Extended Data Table 2** (below). Additionally, we added a description of the data generation, sequencing, assembly, and methylation calling of this genome to the **Methods**.

Extended Data Table 2. Sequence and assembly of the HG00733 genome.

Species	Assembly*			PacBio HiFi data		ONT data	
	Size (Gbp)	No. of contigs	N50 (Mbp)	Sequencing depth [†]	Read N50 (kbp)	Sequencing depth [†]	Read N50 (kbp)
Human (HG00733)	6.08	1,592	34.89	33.48	13.5	94.0	32.7

*Assembled from PacBio HiFi data with hifiasm (Cheng et al., *arXiv*, 2020)

[†]Assumes a 3.1 Gbp genome

Since our original submission, we performed several additional analyses on the HG00733 chromosome 8 centromere assemblies. The results confirm not only the structure and organization of the

chromosome 8 centromere but functional features as well, such as the 77 or 95 kbp pockets of hypomethylation, revealing these characteristics as general properties of the chromosome 8 centromere (**Extended Data Fig. 8**; below). We provide the figure below for convenience.

Extended Data Figure 8. Sequence, structure, and epigenetic map of human diploid HG00733 chromosome 8 centromeres. a,b) Repeat structure, alpha-satellite organization, methylation status, and sequence identity heat map of the **a)** maternal and **b)** paternal chromosome 8 centromeric regions from a diploid human genome (HG00733; **Extended Data Table 2**) shows structural, evolutionary, and epigenetic similarity to the CHM13 chromosome 8 centromeric region (**Fig. 2a**).

Line 190-195 – This is another red flag for the paper – it appears the authors are trying to make a very specific statement about functional annotations of the centromere of chromosome 8 in CHM13 that are only inferential since the ChIP-seq data employed herein is not actually from CHM13 but different data sets from earlier studies. The authors cannot state that they determined the epigenetic centromere for this specific chromosome – they can only infer where it is. This is an important point, particularly given the samples that were used for the ChIP-seq are neither normal diploid lines (contrary to the statement in the supplemental figure legend), but rather are two abnormal cell lines with neocentromeres. This limitation should be made clearer in the language used throughout on this point. Moreover, the recent observations that CENP-A positionally shifts in the genomic space of a centromere among different individual plants of the same species (see Gent et al 2017) make this limitation, and the links to methylation, a crucial point to clarify.

To directly determine the location of centromeric chromatin, we performed two independent replicates of CENP-A ChIP-seq on CHM13 cells and mapped the data to the CHM13 genome using BWA-MEM and a k-mer-based approach that we developed. Both replicates reveal that CENP-A is located within a 632 kbp region centered on the hypomethylated region of the D8Z2 α -satellite HOR array (Fig. 2a, Extended Data Fig. 11; below) and in the same position we previously identified with CENP-A ChIP-seq from two neocentromeric cell lines (Extended Data Fig. 11; below). Coupling these results with the CENP-A chromatin fiber-FISH (Fig. 2c) provides direct support that CENP-A is located in and around the hypomethylated region of the CHM13 centromere.

Extended Data Figure 11. Location of CENP-A chromatin within the CHM13 D8Z2 α -satellite HOR array. a,b) Plot of **a)** the ratio CENP-A ChIP to bulk nucleosome reads mapped via BWA-MEM, or **b)** the number of k-mer-mapped CENP-A ChIP (black) or bulk nucleosome (dark gray) reads (**Methods**). Shown are two independent replicates of CENP-A ChIP-seq performed on CHM13 cells (top two panels), as well as single replicates of CENP-A ChIP-seq performed on human diploid neocentromeric cell lines (bottom two panels; **Methods**). While the neocentromeric cell lines have a neocentromere located on either chromosome 13 (IMS13q) or 8 (MS4221)^{8,9}, they both have at least one karyotypically normal chromosome 8 from which centromeric chromatin can be mapped. We limited our analysis to diploid cell lines rather than aneuploid ones to avoid potentially confounding results stemming from multiple chromosome 8 copies that vary in structure, such as those observed in HeLa cells¹⁰.

Ext data figure 9 – The authors show ChIP-seq peaks for both IMS13q and MS4221, and the scale and heights are virtually identical. It is not clear how this can be the case when IMS13q has two active centromeres at the same location on each homolog of chromosome 8, but MS4221 only has one active centromere at this location. How are the methylation reads and these variable ChIP-seq reads being normalized to afford a realistic comparison among these different centromere haplotypes?

The plots for the two cell lines show similar ratios of CENP-A ChIP to bulk nucleosome reads because we downsampled the two datasets to the same amount of data over the chromosome 8 centromere. This allowed us to more easily discern the site of CENP-A enrichment between the samples. We included this detail in the Methods.

Lines 230-231 and Ext data figure 10 – The rationale behind the comment that fiber FISH allows copy resolution is tenuous. How can the authors distinguish copy numbers versus the signal that occurs on fibers due to chromatin breakage during the FISH technique?

While breakage can occur on large (>2 Mbp) stretched chromatin fibers, it has been consistently shown that chromatin fibers 500 kbp - 2 Mbp in length can be obtained without any breakage, and this has been important for resolving the size and copy number of large repeat regions in diverse genomes over the past few decades. For example, Fransz et al., *Plant J*, 1996 showed that the 5s rDNA region in tomato is ~660 kbp via fiber-FISH; Cheng et al., *Chromosome Res.*, 2002 showed that a region of chromosome 10 spanning 1 Mbp in rice could be detected with fiber FISH; and Jackson et al., *Genome*, 1998 showed that DNA clusters spanning up to 1.7 Mbp in *A. thaliana* can be resolved with fiber FISH. These and other examples in the literature provide support that our chromatin fibers, which contain an estimated 817 +/- 63 kbp VNTR, is well within the range of fiber lengths that have been obtained by others.

We would also like to stress that the chromatin fiber FISH was only one of several techniques and methods that support the estimated size of the 8q21.2 VNTR, which include pulsed-field gel Southern blot as well as long- and short-read mapping. Because of the long-withstanding usage of fiber FISH to physically map and estimate the size of repeat-rich regions and the concordance of our findings with orthogonal techniques we've employed, we believe that our method and analyses are valid.

Ext Data Table 4 – The quality of NHPs assemblies are low that that of the presented CHM13 - this should be explicitly stated in the main text since direct comparisons are being made to a complete assembly of human 8.

We regard both the CHM13 and NHP chromosome 8 centromere assemblies as complete. Indeed, numerous bioinformatic analyses, including ONT and PacBio HiFi long-read data comparisons, short-read data read-depth analysis, and TandemQUAST, support their sequence, organization, and structure. Additionally, Merqury indicates that the NHP centromere assemblies are 99.9988-100% accurate (QV score >49.3). Because of this, we now refer to the NHP assemblies as “**high-quality** draft assemblies” in the abstract and main text.

Lines 297-301, 386-7 and 861-880, Do the authors consider the lower recombination rates at centromeres compared to chromosome arms? This would affect rate models for neutral evolution.

This is an interesting point and one of still considerable debate. For example, it has been reported in multiple species, including in humans, that levels of divergence between species do not correlate with recombination rates (Begun and Aquadro *Nature*, 1992; Nachman, *Trends Genet*, 2001; Takahashi et al., *Mol Biol Evol*, 2004; but also see the review by Smukowski and Noor, *Heredity*, 2011 for conflicting results). Because of the lack of accurate recombination rate estimates for centromeric regions in the human genome, we investigated this by assessing the relationship between divergence and recombination rate using coalescent simulations. We simulated 10 kbp sequences based on the known primate phylogeny with parameters randomly drawn from the 95% confidence intervals reported in recent publications (**Rebuttal Figure 1**). The overall population mutation rate ($\theta=4*Ne*u$, where effective population size $Ne=8345$ and $u=1e-8$ per bp per generation) was set to match the expected divergence levels between human and other primate species based on the literature (human-macaque: 8-9%, human-orangutan: 5-6%, human-gorilla: 3-4%, human-chimpanzee/bonobo: 1-2%). We tested a

range of recombination rates (scaled recombination rate $\rho = [4 * N_e * r]$ per 10 kbp), generated 250 replicates for each recombination rate in order to assess simulation uncertainty, and computed pairwise divergence using the same computation described in the main text. Consistent with the expectation of no correlation between levels of divergence and recombination rates, we find that divergence estimates are stable across different recombination rate bins for all pairs (**Rebuttal Figure 2**), suggesting that the expected divergence estimates around centromeres reported in the main text are unlikely to be affected by the lower recombination rates at centromeres. We note that larger variances in divergence at the bins of lower recombination rates are due to fewer crossing over events among simulated lineages but the mean divergence remains constant.

Rebuttal Figure 1. Schematic representation of the primate phylogeny and parameters used in coalescent simulations. Timing and effective population size estimates are from recent publications (Glazko and Nei, *Mol Biol Evol*, 2003; Prado-Martinez et al., *Nature*, 2013; and Xue et al., *Genome Res*, 2016).

Rebuttal Figure 2. Divergence estimates between human and other primates using coalescent simulations under a range of 21 recombination rates ($\rho/10$ kbp). Simulations were generated based

on models with plausible parameters drawn from literature (Glazko and Nei, *Mol Biol Evol*, 2003; Prado-Martinez et al., *Nature*, 2013; and Xue et al., *Genome Res*, 2016). 250 replicates of 10 kbp segments were simulated with different random seeds to assess the variation of divergence estimates as a function of recombination rate. Divergence is described in the Methods. Lines and dots show the mean divergence at individual recombination rate bins, while the shaded areas represent the 95% confidence intervals.

Lines 320-323. This is another overstatement of the data. The hypomethylated region is not shown herein to be enriched for CENP-A in this genome – rather it is inferred from orthogonal data from different individuals.

As mentioned above (in response to the comment starting with “Line 190-195”), we performed two independent replicates of CENP-A ChIP-seq on CHM13 cells directly. We show that CENP-A is enriched within a 632 kbp region encompassing the hypomethylated region of the D8Z2 HOR array (Fig. 2a), in agreement with orthogonal data from two other individuals that we included in our original submission (Extended Data Fig. 11). We reproduce the new figures here for convenience.

a

Extended Data Figure 11. Location of CENP-A chromatin within the CHM13 D8Z2 α -satellite HOR array. a,b) Plot of a) the ratio CENP-A ChIP to bulk nucleosome reads mapped via BWA-MEM, or b) the number of k-mer-mapped CENP-A ChIP (black) or bulk nucleosome (dark gray) reads (Methods). Shown are two independent replicates of CENP-A ChIP-seq performed on CHM13 cells (top two panels), as well as single replicates of CENP-A ChIP-seq performed on human diploid neocentromeric cell lines (bottom two panels; Methods). While the neocentromeric cell lines have a neocentromere located on either chromosome 13 (IMS13q) or 8 (MS4221)^{8,9}, they both have at least one karyotypically normal chromosome 8 from which centromeric chromatin can be mapped. We limited our analysis to diploid cell lines rather than aneuploid ones to avoid potentially confounding results stemming from multiple chromosome 8 copies that vary in structure, such as those observed in HeLa cells¹⁰.

Lines 395-7 –In the following statement, the authors link multiple statements to a small number of references that are, taken in the context of the entire sentence, incorrect. Molecular drive, including gene conversion, was first described in the early '80s (Dover et al 1982). Refs 48-49 present a model for centromere drive that links observed rapid rates of satellite sequences to rapid rates of centromere binding sequences, but do not present direct evidence for “rapid centromere sequence evolution as a driving force in speciation” nor “hybrid incompatibility”. Other studies have made the link between rapid satellite evolution and hybrid incompatibility that do not necessarily involve centromeres (for example Ferree and Barbash 2009), so this overstatement does not reflect these other satellite observations. Moreover, HOR divergence, specifically, has not been shown to cause speciation nor hybrid incompatibility. Finally, reference 50 is for a study in monkeyflowers that links a meiotic driver to an allele (D) adjacent to a centromere, but the sequence of that allele is not directly defined as an HOR array. The sentence is awkward as stated (“rapid evolution ...due to the accumulation of sequence

differences” is circular). As worded, this sentence makes a grand and defining statement about speciation that both simplifies and overstates the current state of the field and does not consider the many other factors involved, such as inversions, repressed recombination, meiotic drive, etc.

The referee is correct that there are a mix of ideas presented in the Discussion, most of which are fairly speculative in nature. Due to space constraints, we eliminated the sentence in question and revised the paragraph to be more straightforward. The revised section now reads asfollow:

“Satellite repeats rapidly evolve within and between species through mechanisms such as unequal crossing over and gene conversion⁵⁰. It is interesting that sequence comparisons among three human centromere 8 haplotypes predict regions of excess allelic variation and structural divergence (**Extended Data Fig. 9**), although the locations within the HOR differ among haplotypes (**Extended Data Figs. 8, 9**). Now that complex regions such as these can be sequenced and assembled, it will be important to extend these analyses to other centromeres, multiple individuals, and additional species to understand their full impact with respect to genetic variation and evolution.”

Lines 756 – The title of the methods section should be changed to reflect the content; the order is reversed with respect to the order data is presented in the main text.

We revised the title of this section to “Fluorescence *in situ* hybridization (FISH) and immunofluorescence (IF)”. We also reordered this Methods subsection to reflect the ordering in the main text.

Lines 506-508 – What are the mapped kmers being used? This is awkward wording, making the kmers referred to unclear. Why are the authors using Jellyfish instead of Merqury for the validation?

The k-mers used in this analysis are described in the subsequent paragraph, which states: “CHM13 Illumina data (SRR1997411, SRR3189741, SRR3189742, SRR3189743) was used to identify k-mers with $k = 21$. In Merqury, every k-mer in the assembly is evaluated for its presence in the Illumina k-mer database, with any k-mer missing in the Illumina set counted as base-level ‘error’. We detected 1,474 k-mers found only in the assembly out of 146,259,650 bp, resulting in a QV score of 63.19...”. We hope this additional explanation provides clarity about the k-mers and where they come from.

We use Merqury (Rhie et al., *Genome Biol*, 2020) instead of Jellyfish (Marçais and Kingford, *Bioinformatics*, 2011) for the validation because Merqury is designed to evaluate the quality of assemblies while Jellyfish is not. Merqury has two parts: 1) building a k-mer database, and 2) assessing the quality of the assembly with the k-mer database. While Jellyfish is able to build a k-mer database, it is not able to assess the quality of an assembly with it. Therefore, we apply Merqury for assembly quality assessment.

Referee #3 (Remarks to the Author):

The manuscript entitled “The structure, function, and evolution of a complete human chromosome 8” presents first linear assembly of a complete human autosome, including the centromeres and complex repetitive regions, as well as a in depth description of the function and evolution of complex centromeric regions. Overall it is clear that this is a significant achievement that required a large amount of effort and ingenuity. I think the paper could be acceptable for publication as is, but I do have a few comments that might improve the presentation to a broader audience.

We thank the referee for their helpful comments and suggestions. We made all of the requested changes, and we think this has improved the clarity and readability of the manuscript.

1) In the opening sentence it is not completely clear if the authors are referring to complex repeats within segmental duplications or the segmental duplications themselves. I suspect it is actually both, but it may be worth rewording for clarity.

Actually, both are correct, but the emphasis is more on the large blocks of repetitive sequence, so we clarified and revised the wording as follows (changes in **bold**):

“Since the announcement of the sequencing of the human genome 20 years ago^{1,2}, human chromosomes have remained unfinished due to large regions of highly identical repeats **clustered** within centromeres, **regions of** segmental duplication, and the acrocentric short arms of chromosomes. The presence of large swaths (>100 kbp) of highly identical repeats that are themselves copy number polymorphic has meant that such regions have persisted as gaps, limiting our understanding of human genetic variation and evolution^{3,4}.”

2) On line 84 where they mention “neocentromere” for the first time it might be useful to call it a “recurrent polymorphic neocentromere” or add some additional description, as read it sounds like they are saying that the primary human centromere is a neocentromere, although the rest of the manuscript makes it clear that this is not the case.

We added the suggested wording, and it now reads as follows (changes in **bold**):

“The chromosome, however, also contains one of the most structurally dynamic regions in the human genome—the β -defensin gene cluster located at 8p23.1^{18–20}—as well as a **recurrent polymorphic** neocentromere located at 8q21.2, which have been largely unresolved for the last 20 years.”

3) Although this assembly is clearly a significant feat, I think readers may ask why all chromosomes have not been assembled in the same way given that they are all (all but X) present in the ONT and a PacBio data that were generated for this study. Chromosome 8 is a reasonable first target, but why not another or all other chromosomes? A bit more discussion of the challenges that are likely to be met in putting together the other chromosomes could be useful to address this.

Yes, the resources and approaches developed here are, in fact, being used to assemble the entire genome. Certain classes of DNA and repeat elements (acrocentric DNA) still remain particularly problematic, and the ability to generate telomere-to-telomere assemblies from diploid genomes is the

next major challenge. Dedicated efforts for specific chromosomes or regions are still required, for example, for comparative evolutionary analyses such as we have done here. We added some text to the Discussion to highlight these remaining challenges:

“With regards to the completion of the human genome, we note that the resources and approaches developed herein are, in fact, being used to assemble the entire CHM13 genome. Despite efforts over the last year, several gaps still remain, especially with respect to the rDNA clusters, whose high copy number and repeat content on the short arms of human acrocentric chromosomes pose considerable challenges for completion of a telomere-to-telomere assembly. Nevertheless, the first sequence of a complete human genome is imminent, and the next challenge will be applying the methods to fully phase and assemble diploid genomes^{48,49}.”

4) For Pac Bio data it seems a bit strange to use the term “mapped” in relation to methylated cytosines (e.g. line 188, 237). Would it be better to say something like - we inferred the location of methylated cytosines based on PacBio polymerase kinetics – or something to that effect?

We reworded these two sentences to state that methylated bases were detected with Nanopolish (which infers the location of the methylation as described in the Methods). The two sentences are now worded as follows (**changes in bold**):

“To investigate the epigenetic organization, we **detected** methylated cytosines along the centromeric region and found that most of the α -satellite HOR array is methylated, except for a small, 73 kbp hypomethylated region (**Fig. 2a**).”

“**Detection** of methylated cytosines **via Nanopolish**²⁵ shows that each 12.192 kbp repeat is primarily methylated in the 3 kbp region corresponding to *GOR1/REXO1L1*, while the rest of the repeat unit is largely unmethylated (**Fig. 3a**).”

5) “While this is consistent with the VNTR being the potential site of the functional kinetochore of the neocentromere, sequence and assembly of the neocentromere containing cell line will be critically important” - Given that it is stated earlier in the paragraph that there are multiple unrelated instances of neocentromere formation, should this be extended to mention a need to sample other individuals/cell lines?

This is a good suggestion. We revised this sentence to state:

“While this is consistent with the VNTR being the potential site of the functional kinetochore of the neocentromere, sequence and assembly of this **and other** neocentromere-containing cell lines will be critically important.”

6) On Line 344 the “Smith model” is mentioned. This seems to refer to a reference earlier in the paragraph, but it would be useful to tie this to a specific reference in the sentence where it is mentioned.

Thanks for this suggestion. We now provided a citation to Smith, *Science*, 1976 in the relevant sentence.

Sincerely,

Jeramiah Smith

Referee #4 (Remarks to the Author):

Manuscript: 2020-09-16437

Title: The structure, function, and evolution of a complete human chromosome 8

Authors: Logsdon et al.

The authors present the first telomere-to-telomere assembly of a human autosome, chromosome 8. A combination of ONT and HIFI data allowed to bridge three previously remaining gaps, all composed of (multi-)megabase sized near-identical repeat clusters not amenable to sequence assembly before the introduction of ultra-long ONT and Pacbio Sequel II Hifi technology. Besides presenting a blueprint for a method to produce finished human chromosome assemblies the authors demonstrate the importance of gap-free sequences. They reveal the complexity and structural variability of the β -defensin cluster which is important, as SV and CNV at this gene cluster is associated with important diseases. Similarly, the structure of the 8q21.2 VNTR region is fully resolved which is a prerequisite for testing structural features at this locus for their importance in neo-centromere formation which has been reported for this locus.

Sequencing and assembly of entire centromeres of human chromosomes was shown before (Jain et al. 2018, Miga et al. 2020). The exciting key example of this study though for demonstrating the importance of a gap free reference sequence as a basis for understanding basic questions in biology is the presentation of the entire Chr8 centromere including methylation status of the alpha-satellite HOR arrays, postulating the most likely position of the functional kinetochore. Chromosome 8 centromere organization was then analysed in an evolutionary context: orthologous chr8 centromeres were produced for chimpanzee, orangutan and macaque. Shared and distinct repeat organization, overall structural organization and mutation frequencies were studied, providing the basis for new hypothesis to be tested broadly after more autosome centromeres are sequenced in human and apes.

The study is well written and illustrated with intuitive figures. Abstract, introduction and discussion are clear and rather straightforward also for people outside of the human genome research, however, some nomenclature could make reading for non-biologists harder. One example: human chromosome banding nomenclature doesn't need to be explained necessarily, however, 8p23.1 for the β -defensins should probably also be shown in Fig 1a for making tracking of content easier, as 8q21.2 is shown. Major conclusions based on the structural genomic findings were typically validated by independent technical methods. I could not find any indication of flaws.

We thank the referee for their kind comments regarding the manuscript, figures, and study as a whole. We made many of the changes requested by the referee, including the addition of the chromosome 8p23.1 banding annotation in **Fig. 1a** (shown below), a complete list of data generated in this study and the Miga et al., *Nature*, 2020 study (**Extended Data Table 9**), additional analyses regarding the differences between our methods and assemblies relatives to the ones in Miga et al., *Nature*, 2020 (**Rebuttal Figure 3**), and additional discussion on the future of T2T genomes (**Discussion; pp 10**).

Fig. 1a showing the chromosome banding notation for the β -defensin locus.

Initially, I was worried whether we have to expect now gapfree sequence papers for all 22 human autosomes as recently Miga et al. already reported a gapfree x-chromosome as part of a whole genome de novo assembly of the same cell line CHM13. I think the biology and story presented here justifies a chromosome 8 specific publication, however, I think the authors must be more transparent about what datasets were newly developed here and what data was recycled from Miga et al. – a table, listing previously published datasets and newly generated datasets, including all accession codes (not only project IDs) would be very useful and desired, especially as Miga et al. did not fully disclose all their data as datasets submitted to NCBI.

Based on the referee's suggestion, we generated a new table (**Extended Data Table 9**; below) that lists all datasets used in this study. Importantly, this table distinguishes those generated for this study from those that were previously published and/or publicly available. Specifically, as part of this study, we generated new ONT, Strand-seq, Iso-Seq, and CENP-A ChIP-seq data for the CHM13 genome, ultra-long ONT data for the diploid human genome HG00733, and PacBio HiFi and ONT data for the nonhuman primate genomes. We include the corresponding BioProject, accession number(s), and relevant publications for each dataset, and we revised the **Data Availability** section to reference this new table.

Extended Data Table 9. Datasets generated and/or used in this study.

Datasets generated in this study								
Species	Sample	Data type	BioProject	Accession #	URL	Reference		
Human (Homo sapiens)	CHM13	Complete chromosome 8 sequence	PRJNA559484	CP061028	--	This study		
		ONT	PRJNA559484	SRR12618224-SRR12618325	--	This study		
		Strand-Seq	--	--	https://zenodo.org	This study		
		Iso-Seq	PRJNA559484	SRR12519035, SRR12519036	--	This study		
		CENP-A ChIP-Seq	PRJNA559484	SRR13278681-SRR13278684	--	This study		
		HG00733	ONT	PRJNA686388	Pending accession	--	This study	
	Testis (pooled)	PacBio HiFi	PRJEB36100	ERR3822935, ERR3861382-ERR3861387	--	This study		
			PRJNA659539	SRR12544672	--	This study		
		Iso-Seq with Teloprime	PRJNA659539	SRR12544673	--	This study		
			PRJNA659539	SRR12524788	--	This study		
			PRJNA659539	SRR12524789	--	This study		
			PRJNA659539	SRR12552556-SRR12552568	--	This study		
Chimpanzee (Pan troglodytes)	Clint; S006007	ONT	PRJNA659034	SRR12517369-SRR12517374, SRR12517378, SRR12517389-SRR12517390	--	This study		
		PacBio HiFi	PRJNA659034	SRR12551266-SRR12551275	--	This study		
Orangutan (Pongo abelii)	Susie; PR01109	ONT	PRJNA659034	SRR12517385-SRR12517387	--	This study		
		PacBio HiFi	PRJNA659034	SRR12517382-SRR12517385	--	This study		
Macaque (Macaca mulatta)	AG07107	ONT	PRJNA659034	SRR12517375-SRR12517377, SRR12517379-SRR12517381	--	This study		
		PacBio HiFi	PRJNA659034	SRR12517375-SRR12517377, SRR12517379-SRR12517381	--	This study		
Previously published datasets								
Species	Sample	Data type	BioProject	Accession #	URL	Reference		
Human (Homo sapiens)	CHM13	ONT	PRJNA559484	SRR10035573-SRR12564439	--	Miga et al., Nature , 2020		
		PacBio HiFi	PRJNA530776	SRR11292120-SRR11292123	--	Nurk et al., Genome Res. , 2020		
		Illumina	PRJNA269593	SRR1997411, SRR3189741-SRR3189743	--	Schneider et al., Genome Res. , 2017		
		Bionano DLS	PRJNA269593	SUPPPF_000002917	--	Miga et al., Nature , 2020		
	HG00733	ONT	PRJEB37264	ERR3988496-ERR3988498	--	Shafin et al., Nat Biotechnol. , 2020		
			PRJNA191094	SRR766738, SRR766741	--	Hasson et al., Nat Struct Mol Biol. , 2013		
		CENP-A ChIP-Seq	PRJNA191094	SRR766737, SRR766740	--	Hasson et al., Nat Struct Mol Biol. , 2013		
			PRJNA191094	SRR6029680	--	Kronenberg et al., Science , 2018		
Orangutan (Pongo abelii)	Susie; PR01109	Illumina	PRJNA369439	SRR6029680	--	Kronenberg et al., Science , 2018		
Publicly available datasets								
Species	Sample	Data type	BioProject	Accession #	URL	Reference		
Human (Homo sapiens)	K562	Iso-Seq	PRJNA30709	SRR10838645, SRR10838646	--	Unpublished		
		Iso-Seq	PRJNA30709	SRR10838655-SRR10838657	--	Unpublished		
		Iso-Seq	PRJNA30709	SRR10838653, SRR10838654	--	Unpublished		
		Iso-Seq	PRJNA30709	SRR10838647-SRR10838650, ENCLB200YVA, ENCLB735WVC	--	Unpublished		
		Iso-Seq	PRJNA30709	SRR10838641	--	Unpublished		
		Iso-Seq	PRJNA30709	SRR10838651	--	Unpublished		
		Iso-Seq	PRJNA30709	SRR10838643	--	Unpublished		
		Iso-Seq	PRJNA30709	SRR10838652	--	Unpublished		
		Iso-Seq	PRJNA30709	SRR10838644	--	Unpublished		
		Iso-Seq	PRJNA30709	SRR10838642	--	Unpublished		
		Iso-Seq	--	--	https://download.ncbi.nlm.nih.gov	Unpublished		
		Chimpanzee (Pan troglodytes)	Clint; S006007	Illumina	PRJEB18078	ERR1759383-ERR1759395	--	Unpublished
		Macaque (Macaca mulatta)	AG07107	Illumina	PRJNA476474	SRR11467824	--	Unpublished

ONT: Oxford Nanopore Technologies
PacBio: Pacific Biosciences
HiFi: High-Fidelity
DLS: Direct Label and Stain

Miga et al. report already the chromosome 8 centromere and methylation status (Miga et al. extended data fig. 10) – please comment!

While the chromosome 8 centromere assembly and its corresponding methylation profile were briefly mentioned in Miga et al., *Nature*, 2020, the assembly was not complete at that time. It’s important to note that their assembly was constructed based on ONT data only, whereas our assembly was built using a combination of ONT and PacBio HiFi data. As a result, their assembly has several structural errors and a lower per-base accuracy compared to ours (Miga et al.’s QV is 28.19; our QV is 60.46). We also note that our assembly is supported by multiple orthogonal datasets, including mapped native long reads, pulsed-field gel Southern blot, FISH on metaphase chromosomes, and droplet digital PCR.

We generated a new **Rebuttal Figure 3** to highlight the differences between our chromosome 8 centromere assembly and the previous working draft assembly described in Miga et al., *Nature*, 2020. As shown below, the chromosome 8 centromere assembly generated in this study has uniform PacBio HiFi coverage and is free from “spikes” or “dips” that usually indicate a collapse or misassembly. In contrast, the chromosome 8 centromere assembly described in Miga et al. had six such spikes and dips in coverage. Additionally, by color coding the most common base in the HiFi reads aligned at each position, we show that the assembly generated in this study is largely free from misassemblies that can cause read mismapping, in contrast to the Miga et al. assembly. Quantification of the overall size of the

assemblies reveals that the Miga et al. assembly has 27,125 fewer bp relative to our assembly, which are located within the D8Z2 HOR array. Taken together, our analysis reveals that our assembly improves upon the assembly reported in Miga *et al.*, as it resolves six large structural errors and improves the overall base accuracy.

Rebuttal Figure 3. Comparison of the CHM13 chromosome 8 centromere assembly reported in this study and Miga et al., *Nature*, 2020. a,b) Plot showing the PacBio HiFi coverage of the CHM13 chromosome 8 centromeric region reported in **a)** this study and **b)** Miga et al., *Nature*, 2020. While the centromere assembly reported in this study has uniform coverage, indicating a lack of large structural errors, the assembly reported in Miga et al., *Nature*, 2020 has six spikes and dips in coverage that likely indicate a collapse or misassembly in sequence. Additionally, the Miga et al. assembly is missing approximately 27,125 bp of sequence present in our assembly.

Miga et al. used the kmer based ONT ultra-long read barcoding approach for centromere and other repeat-based gap regions. Here Logsdon et al. give the impression of a newly established approach. Since the papers are from the same group of authors, still a transparent referencing should be expected, or – Logsdon should try to explain better what exactly was improved in comparison to the previous Miga et al. paper or clearly state that the same approach was used (maybe by including more de novo generated data).

The approaches for assembly are indeed quite different. While we used a k-mer-based approach, which relies on singly unique k-mers (SUNKs) to assemble ONT reads across select regions, Miga et al. used a structural variant-based approach (not a k-mer-based approach), which uses structural differences in alpha-satellite HORs to assemble ONT reads across a region. These two approaches are

inherently different. To make it clearer that the approach we developed is new, we revised the opening sentence of the Results section as follows (changes in **bold**):

“To resolve the gaps in human chromosome 8 (**Fig. 1a**), we developed a **novel** targeted assembly method (**Methods**) that leverages the complementary strengths of ONT and PacBio long-read sequencing (**Fig. 1b; Methods**).”

All sequencing data was generated generically from whole genomic DNA. The technical advance for centromere assembly was featured here, however for a single autosome similarly as previously done for the human X chromosome. If the authors are not following a strategy for publishing now 20 more chromosome papers, I think it would be very instructive, not only to human genome researchers, to get a better understanding of the challenges remaining for full assembly of the other autosome centromeres. Miga et al. have briefly mentioned this problem and indicated that some autosomal centromeres will be easier assembled than others. I suppose, at least the Logsdon et al datasets and established computational pipelines should be similarly applicable to the whole genome assembly of CHM13. Please give a better understanding of what are the expected limitations for the assembly of other chromosomes? Or are the other centromeres assembled in fact already to similar quality? Why is this not a process for parallelization? For example, I am impressed by the 7 Mbp alpha-satellite repeat assembled for Macaque - this result is of general importance for genome assembly also outside human research. Especially crop genomes can have centromeres much larger than those of human autosomes, so more detail about the true potential and/or limitation of the presented approach beyond human would make this study more attractive for a broader audience.

This is a good suggestion. To be sure, we have no intention of publishing 20 more chromosome papers. The next step will be the complete sequence of the effectively haploid hydatidiform mole genome (T2T-CHM13). The T2T Consortium has been working hard on this for the last year. While most centromeres have now been fully resolved (after considerable effort) and are in the process of experimental validation, other classes of DNA and repeat elements (acrocentric DNA, for example) still remain particularly problematic, and the ability to generate telomere-to-telomere assemblies from diploid genomes is the next major challenge. Dedicated efforts for specific chromosomes or regions are still required, for example, for comparative evolutionary analyses as described in this paper. We added some text to the Discussion to highlight these remaining challenges and to increase appeal:

“With regards to the completion of the human genome, we note that the resources and approaches developed herein are, in fact, being used to assemble the entire CHM13 genome. Despite efforts over the last year, several gaps still remain, especially with respect to the rDNA clusters, whose high copy number and repeat content on the short arms of human acrocentric chromosomes pose considerable challenges for completion of a telomere-to-telomere assembly. Nevertheless, the first sequence of a complete human genome is imminent, and the next challenge will be applying the methods to fully phase and assemble diploid genomes^{48,49}.”

Reviewer Reports on the First Revision:**Ref #1**

This paper reports the finished assembly of human chromosome 8 with a focus on B-defensin gene cluster, the HOR alpha satellite array at the centromere, and the 8q21.2 VNTR (neo-centromere site). The authors have made substantial revisions based on reviewer comments. Among the revisions/additions are: an assessment of mappability, new CENP-A ChIP-seq from CHM13, additional FISH images, and text, tables, and figures to make their methods clearer. These additional data and analyses improved the manuscript. I am satisfied with the authors' responses to my previous comments. I just have a couple of minor comments about the additions.

Minor comments:

This sentence is not easy to read: "Each k-mer from the fragments was placed once at random between all sites in the centromeric region that had a perfect match to that k-mer"

In Extended Data Figure 8 legend: "...shows structural, evolutionary, and epigenetic similarity to the CHM13 chromosome 8 centromeric region". What do you mean by evolutionary similarity (between HG00733 and CHM13) in this context?

Ref #2

In this revised manuscript by Logsdon et al, the authors have added new data and figures to support concerns raised in an earlier round of review. This manuscript has been improved in both language and presentation. New data are included as requested, methods are clarified and figures have been updated to reflect these changes. Points raised in earlier reviews (e.g. details of HG00733, complete FISH images, ChIP-seq datasets (and CENP-A bulk nucleosomes) used, comments on evolutionary rates, etc) have been addressed in full. I recommend that the rebuttal figures 1 and 2 be included in the supplementary material to support the evolutionary rate conclusions and rebuttal figure 3 be included to really drive home the advance in this study over the previous X paper.

This is an important contribution to our understanding of chromosome structure and sequence, particularly across centromeres and worthy of publication in Nature.

Ref #3

I am satisfied with the responses to my previous comments and have no further comments.

Ref #4

Thanks for answering all my questions and comments. Congratulations to a nice study. No additional requests.

Author Rebuttals to First Revision:

Referee comments are in black; our responses are in blue

Referee #1 (Remarks to the Author):

This paper reports the finished assembly of human chromosome 8 with a focus on B-defensin gene cluster, the HOR alpha satellite array at the centromere, and the 8q21.2 VNTR (neo-centromere

site). The authors have made substantial revisions based on reviewer comments. Among the revisions/additions are: an assessment of mappability, new CENP-A ChIP-seq from CHM13, additional FISH images, and text, tables, and figures to make their methods clearer. These additional data and analyses improved the manuscript. I am satisfied with the authors' responses to my previous comments. I just have a couple of minor comments about the additions.

Minor comments:

This sentence is not easy to read: "Each k-mer from the fragments was placed once at random between all sites in the centromeric region that had a perfect match to that k-mer"

We have revised this sentence as follow:

"K-mers with perfect matches to multiple sites within the centromeric region were assigned to one of the sites at random."

In Extended Data Figure 8 legend: "...shows structural, evolutionary, and epigenetic similarity to the CHM13 chromosome 8 centromeric region". What do you mean by evolutionary similarity (between HG00733 and CHM13) in this context?

We mean that the evolutionary layers are similar between the CHM13 and HG00733 centromeres. However, since this is confusing wording, we have removed the word "evolutionary" to state:

"... shows structural and epigenetic similarity to the CHM13 chromosome 8 centromeric region"

Referee #2 (Remarks to the Author):

In this revised manuscript by Logsdon et al, the authors have added new data and figures to support concerns raised in an earlier round of review. This manuscript has been improved in both language and presentation. New data are included as requested, methods are clarified and figures have been updated to reflect these changes. Points raised in earlier reviews (e.g. details of HG00733, complete FISH images, ChIP-seq datasets (and CENP-A bulk nucleosomes) used, comments on evolutionary rates, etc) have been addressed in full. I recommend that the rebuttal figures 1 and 2 be included in the supplementary material to support the evolutionary rate conclusions and rebuttal figure 3 be included to really drive home the advance in this study over the previous X paper.

This is an important contribution to our understanding of chromosome structure and sequence, particularly across centromeres and worthy of publication in Nature.

Due to figure, word, and space constraints, we are currently unable to include the rebuttal figures in the Extended Data. However, we are planning to release the reviews through the transparent review process, so these figures will become available to the public.

Referee #3 (Remarks to the Author):

I am satisfied with the responses to my previous comments and have no further comments.

Thank you.

Referee #4 (Remarks to the Author):

Thanks for answering all my questions and comments. Congratulations to a nice study. No additional requests.

Thank you.